# Erosion rates in a wet, temperate climate derived from rock luminescence techniques

Rachel K. Smedley[1], David Small[2], Richard S. Jones[2, 3], Stephen Brough[1], Jennifer Bradley[1], Geraint T.H. Jenkins[4]

[1] School of Environmental Sciences, University of Liverpool, Liverpool, UK.

[2] Department of Geography, Durham University, South Road, Durham, UK.

[3] School of Earth, Atmosphere and Environment, Monash University, Melbourne, Australia.

[4] Independent researcher: Powys, Wales, UK

*Correspondence to*: Rachel K. Smedley (rachel.smedley@liverpool.ac.uk)

**Abstract**

A new luminescence erosion-meter has huge potential for inferring erosion rates on sub-millennial scales for both steady and transient states of erosion, which is not currently possible with any existing techniques capable of measuring erosion. This study applies new rock luminescence techniques to a well-constrained scenario provided by the Beinn Alligin rock avalanche, NW Scotland. Boulders in this deposit are lithologically consistent, have known cosmogenic nuclide ages, and independently-derived Holocene erosion rates. We find that luminescence-derived exposure ages for the Beinn Alligin rock avalanche were an order of magnitude younger than existing cosmogenic nuclide exposure ages, suggestive of high erosion rates (as supported by field evidence of quartz grain protrusions on the rock surfaces). Erosion rates determined by luminescence were consistent with independently-derived rates measured from boulder-edge roundness. Inversion modelling indicates a transient state of erosion reflecting the stochastic nature of erosional processes over the last ~4.5 ka in the wet, temperate climate of NW Scotland. Erosion was likely modulated by known fluctuations in moisture availability, and to a lesser extent temperature, which controlled the extent of chemical weathering of these highly-lithified rocks prior to erosion. The use of a multi-elevated temperature, post-infra-red, infra-red stimulated luminescence (MET-pIRIR) protocol (50, 150 and 225°C) was advantageous as it identified samples with complexities that would not have been observed using only the standard IRSL signal measured at 50 °C, such as that introduced by within-sample variability (e.g. surficial coatings). This study demonstrates that the luminescence erosion-meter can infer accurate erosion rates on sub-millennial scales and identify transient states of erosion (i.e. stochastic processes) in agreement with independently-derived erosion rates for the same deposit.

## 1. Introduction

Rock erosion is dependent upon a variety of internal (e.g. mineralogy, grainsize, porosity, structures) and external (e.g. temperature, moisture availability, snow cover, wind, aspect) factors. Chemical and/or physical weathering of rocks (or rock

decay; Hall et al. 2012) breaks down the surficial materials making them available for transportation (i.e. erosion), where the rates and processes of degradation is primarily controlled by the rock lithology (e.g. Twidale, 1982; Ford and Williams, 1989). For boulders with similar lithologies, the erosion rate is conditioned by weathering principally caused by moisture availability, but also temperature, and in some cases biological factors (Hall et al. 2012). It is widely reported that warmer temperatures increase most rates of chemical activity, while sub-zero temperatures arrest chemical activity on a seasonal basis. However, cold temperatures alone do not preclude chemical weathering (Thorn et al. 2001). As such, rock erosion rates will be sensitive to changing climate (moisture availability, temperature) such as that experienced throughout the Late Holocene (i.e. last 4 ka) (e.g. Charman, 2010), in addition to that forecast for the future due to anthropogenic climate change (e.g. Stocker et al. 2013). Measuring erosion rates over shorter ($\leq 10^3$ a) and longer ($\geq 10^4$ a) integration times is advantageous as each targets a different phenomenon of erosion. Longer timeframes will inform on how landscapes respond to changing large-scale climatic and tectonic conditions (e.g. Herman et al. 2010), whereas shorter timeframes assess local or regional responses to shorter-lived environmental conditions (e.g. climate fluctuations). A number of techniques can constrain long-term, landscape erosion rates on $\geq 10^4$ a timeframes, such as cosmogenic nuclides (e.g. Lal,1991; Braun et al. 2006; Balco et al. 2008) or thermochronology (Reiners and Brandon, 2006). While observational measurements on very short timeframes $\leq 10^2$ a are performed with both direct contact (e.g. Hanna, 1966; High and Hanna, 1970; Trudgill et al. 1989) and non-contact (e.g. Swantesson, 1989; Swantesson et al. 2006) techniques. However, until now it has been difficult to constrain erosion rates on $10^2$ to $10^3$ a timeframes due to a lack of techniques with the required sensitivity and resolution.

The luminescence signal within mineral grains (quartz and feldspar) is reset when a rock surface is exposed to sunlight for the first time (e.g. Habermann et al. 2000; Polikreti et al. 2002; Vafiadou et al. 2007). With continued exposure the luminescence signal resetting in the mineral grains propagates to increasing depths (i.e. the luminescence depth profile is a function of time). Improved understanding of this fundamental principle has led to the development of new applications of luminescence; constraining the timing of rock exposure events (Laskaris and Liritzis, 2011; Sohbati et al. 2011; Lehmann et al. 2018) and rock surface erosion rates (Sohbati et al. 2018; Lehmann et al. 2019a,b). Brown (2020) combine these phenomena within model simulations to explore different sample histories of exposure and burial, informing geomorphological interpretations of luminescence depth profiles measured in samples collected from the natural environment. Here, we investigate erosion rates, rather than weathering rates as the luminescence technique specifically measures the light penetration into a rock surface after the removal of material (i.e. erosion), occurring after the in-situ rock breakdown (i.e. weathering). Luminescence depth profiles are a product of the competing effects of time (which allows the bleaching front to propagate to greater depths) and erosion (which exhumes the bleaching front closer to the surface). Existing studies have suggested that rock luminescence exposure dating is only feasible for very short timeframes (e.g. <300 a; Sohbati et al. 2018) as light penetrates faster than the material can be removed, and/or in settings where erosion rates are <1 mm/ka (Lehmann et al. 2018). Beyond this, the dominant control on the luminescence depth profile is erosion, rather than time, hence if time can be parameterised, then erosion can be determined (and vice versa). Recent findings from erosion simulations compared with

measured data have shown that the erosion rates derived from luminescence depth profiles can be accurate even where stochastic erosion was experienced in nature (Brown and Moon, 2019).

New luminescence techniques have the potential to derive $10^2$ to $10^3$ a scale erosion rates because of two important characteristics: (1) measurable luminescence depth profiles can develop in a rock surface over extremely short durations of sunlight exposure (e.g. days; Polikreti et al. 2003, or years; Lehmann et al. 2018); and (2) luminescence depth profiles are sensitive to mm-scale erosion. Conversely, cosmogenic nuclides are sensitive to m-scale erosion, depending on the density (e.g. Lal, 1991). Therefore, the new luminescence erosion-meter has the potential to provide a step-change in capabilities of measuring erosion rates on currently impossible $10^2$ to $10^3$ a timeframes. However, its application has been limited to few studies (e.g. Sohbati et al. 2018; Lehmann et al. 2019b) validated against long-term erosion rates of landscape evolution from global or regional datasets rather than local, independently-constrained erosion rates derived from the same rock type.

This study tests the accuracy and applicability of rock erosion rates inferred from luminescence techniques in a new latitudinal (57°N) and climate (wet, temperate) setting with independently-constrained erosion rates. The Beinn Alligin rock avalanche in NW Scotland (Fig. 1) provides a well-constrained test scenario as: (1) the boulders were sourced from a single fault-bounded failure scarp occurring within sandstones of the Torridonian group (i.e. rocks are likely to be lithologically consistent); (2) all boulder samples share an identical exposure history as they were deposited by a single, instantaneous event (Ballantyne and Stone, 2004); (3) independent cosmogenic exposure ages constrain the timing of the rock avalanche (Ballantyne and Stone, 2004); and (4) independently-derived erosion rates over the last ~4 ka for the boulders of the Beinn Alligin rock avalanche uniquely provide constraints on erosion rates (Kirkbride and Bell, 2010).

## 2.    Theoretical background

The propagation of a bleaching front (i.e. the depth at which the luminescence signal has been reduced by 50 %) into a rock surface can be described by a double exponential function (Eq. 1), where $L_x$ is the luminescence measured with depth ($x$) from the rock surface, $L_0$ is the saturation limit for this sample (determined experimentally), $t$ is the exposure time, $\overline{\sigma\varphi_0}$ is the intensity of light of a specific wavelength at the rock surface, and $\mu$ is the light attenuation coefficient. To determine the exposure time (t) of a rock surface (and also erosion rates), it is necessary to parameterise $\mu$ and $\overline{\sigma\varphi_0}$, which are likely unique to any specific rock lithology and natural sunlight conditions (e.g. latitude, cloudiness) of the sample being dated, respectively. Therefore, to provide accurate luminescence exposure ages (and also erosion rates), $\mu$ and $\overline{\sigma\varphi_0}$ must be calibrated using samples of known-age with the same lithology and natural sunlight conditions (e.g. a nearby road-cutting).

$$L_x = L_0 e^{\overline{-\sigma\varphi_0}t} e^{-\mu x} \tag{1}$$

Studies have applied rock luminescence techniques (mostly exposure dating) to a variety of lithologies including granites, gneisses (Lehmann et al. 2018, 2019a,b; Meyer et al. 2019), sandstones (Sohbati et al. 2012; Chapot et al. 2012; Pederson et al. 2014), quartzites (Gliganic et al. 2019) and carbonate limestone (Brill et al. 2021). These studies showed that $\mu$ is highly dependent upon the rock lithology, where mineralogy has a strong control on the rock transparency. This is supported by direct measurements of $\mu$ for a variety of lithologies (greywacke, sandstone, granite, and quartzite) using a spectrometer (Ou et al.

2018). In addition to mineralogy, it has also been shown that the precipitation of dark Fe-hydroxides (Meyer et al. 2018) and
rock varnishing (or weathering crusts) (e.g. Luo et al. 2019) can influence $\mu$ by changing the rock transparency principally at
the rock surface. Mineralogy is broadly a constant variable over time. However, the formation of precipitates or rock varnishing
can be time-variable due to changing environmental factors external to the rock; thus, we should consider the possibility that
$\mu$ may be time-variable. Consequently, investigating the rock opacity of each sample is important to assess whether the known-
age samples used to parameterise $\mu$ and $\overline{\sigma\varphi_0}$ were consistent with the unknown-age samples used for exposure dating or
erosion rates.

104         Since the introduction of the new rock luminescence techniques, most studies on K-feldspar (except Luo et al. 2019)

have only utilised the $IR_{50}$ signal as it bleaches more efficiently with depth into rock surfaces compared to higher temperature
post-IR IRSL signals (e.g. Luo et al. 2019; Ou et al. 2018). However, electron multiplying charged coupled device (EMCCD)
measurements of four rock types (quartzite, orthoclase and two different granites) have shown that the post-IR IRSL signals
of rock slices were dominated by K-feldspars, while Na-rich feldspars can contribute towards the $IR_{50}$ signal (Thomsen et al.
2018). It is possible that the different IRSL signals will have different luminescence characteristics (e.g. bleaching rates, fading
rates, saturation levels, light attenuation, internal mineral composition) that could be exploited during measurements. Luo et
al. (2019) used the post-IR IRSL signals with a multiple elevated temperature (MET) protocol (50, 110, 170, 225 °C) to
demonstrate that all the IRSL signals provide luminescence depth profiles, but the lower temperature signals penetrated further
into the rock with depth. The authors fit the four IRSL signals to improve the accuracy of their parameterisation of $\mu$ and $\overline{\sigma\varphi_0}$.
However, no study has yet used the MET-post IR IRSL protocol to exploit the differing luminescence characteristics of the
successively-measured IRSL signals to provide an internal quality control check on the reliability of the measured data, i.e.
the luminescence depth profile will penetrate deeper in to the rock for the $IR_{50}$ signal than the $pIRIR_{150}$ signal, which in turn
will penetrate deeper than the $pIRIR_{225}$ signal. However, all three signals should determine the same erosion rates if the model
parameterisation (i.e. $\mu$ and $\overline{\sigma\varphi_0}$) is accurate. To maximise the potential information that could be derived from the samples,
this study applied a MET-post IR IRSL protocol (50, 150 and 225 °C).

120         For determining erosion rates for rock surfaces of known exposure age, Sohbati et al. (2018) used a confluent

hypergeometric function to provide an analytical solution, but assuming only steady-state erosion. Lehmann et al. (2019a)
provide a numerical approach that exploits the differential sensitivities to erosion of the luminescence (short-term) and
cosmogenic nuclide (longer-term) techniques to erosion to infer erosion histories (steady state and transient over time) for rock
surfaces. This approach uses the experimental data from the luminescence depth profiles and the $^{10}Be$ concentrations for each
sample. Modelling of the luminescence depth profiles accounts for the electron trapping dependent upon the environmental
dose-rate and $D_0$ but does not consider athermal loss of the signal (i.e. anomalous fading) as it has been demonstrated to have
a negligible impact upon the luminescence depth profiles (Lehmann et al. 2019a). Modelling of the $^{10}Be$ concentrations
assumes no inheritance of cosmogenic nuclides from prior exposure, and that the $^{10}Be$ concentrations have been corrected for
sample depth, density and topographical shielding. The luminescence depth profiles and cosmogenic nuclide concentrations
are solved simultaneously for two unknowns: the exposure duration and the erosion history as defined by a step function (e.g.
zero erosion for an initial period of time followed by an instant increase to a constant erosion rate). Forward modelling is used
to simply simulate a projected outcome and here it is used to calculate all of the possible luminescence depth profiles for these
synthetic erosion and exposure histories. Inversion modelling matches measured data with the outcome of simulations to
determine best fit of the raw data. Here, inversion modelling was used to validate the luminescence depth profile and
cosmogenic nuclide concentration data against the synthetic erosion and exposure histories to determine the combinations with
the highest likelihood. Throughout these modelling experiments, a forbidden zone is defined by combinations of erosion rate
and duration that are not possible given the measured $^{10}$Be concentrations; these solutions are excluded from the parameter
ranges used for the inversion model. For example, the forbidden zone identified in the inversion model profile shown in Fig.
7A is restricted to ranges from ca. $10^4$ mm/ka for durations of ca. 100 a to ca. $10^3$ mm/ka for ca. >3000 a.
The approach of Lehmann et al. (2019a) can model synthetic erosion histories in both steady and transient states.
Steady state erosion is defined as a constant erosion rate over a portion of the total duration of surface exposure. Transient
erosion is typical of shorter exposure histories where a steady state of erosion has not yet been reached and the erosion rate
varies over time. In the approach of Lehmann et al. (2019a), transient erosion is defined by erosion rates that decrease linearly
with increased timing of erosion onset within the parameter space, ultimately reaching steady state (i.e. a constant erosion
rate). An illustration of this is provided by Fig. 7A where transient erosion rates of ca. $10^4$ mm/ka were inferred for a minimum
duration of ca. ≤1 a, and extending up to ca. $10^3$ mm/ka for durations up to ca. 50 a. Beyond ca. 50 a, a steady state of erosion
was reached at a constant erosion rate of ca. $10^3$ mm/ka, represented by the flattening of the profile with the highest likelihood.
Alternatively, a profile indicative of a transient state of erosion where no steady state has been established is illustrated by Fig.
7D where transient erosion rates of between ca. $10^2$ mm/ka were inferred for a minimum duration of ca. ≤1 a, and extending
up to ca. $10^1$ mm/ka for durations beyond ca. 200 a. This numerical approach (Lehmann et al. 2019a) allows erosion history
to be considered as non-constant in time (i.e. transient), in addition to steady-state, and so it is more indicative of the stochastic
erosional processes (driven by temperature, precipitation, snow cover, wind) in nature.

### 3. The Beinn Alligin rock avalanche

Today, average winter and summer temperatures in NW Scotland are 7°C and 18°C, respectively, while average annual
precipitation (mostly rainfall) is high (ca. 2,300 mm/a) (Met Office, 2021). The Beinn Alligin rock avalanche (57°35'N,
05°34'W) is a distinct, lobate deposit of large boulders that is 1.25 km long and covers an area of 0.38 km$^2$ (Fig. 1). It has
previously been ascribed various origins including a rockslide onto a former corrie glacier (e.g. Ballantyne, 1987; Gordon,
1993) and a former rock glacier (Sissons, 1975; 1976). However, on the basis of cosmogenic exposure dates that constrain its
deposition to the Late Holocene it is now widely accepted to have been deposited by a rock-slope failure that experienced
excess run-out (e.g. a rock avalanche). The source is a distinct, fault-bounded failure scar on the southern flank of Sgurr Mor,
the highest peak of Beinn Alligin (Ballantyne, 2003; Ballantyne and Stone, 2004). The rock avalanche is comprised of large,
poorly-sorted boulders and is calculated to comprise a total volume of 3.3 – 3.8 x $10^6$ m$^3$, equivalent to a mass of 8.3 – 9.5 Mt
(Ballantyne and Stone, 2004). The source lithology is Late Precambrian Torridonian sandstone strata. The Torridonian
sandstones are reddish or reddish brown terrestrial sedimentary rocks deposited under fluvial or shallow lake conditions
(Stewart, 1982). The sandstones maintained a common origin throughout deposition (Stewart, 1982) and are thus largely
consistent in mineralogy (dominated by quartz, and alkali and plagioclase feldspar) although there are some local variations
in grain size (Stewart and Donnellan, 1992).

168       The $^{10}$Be concentrations of three boulders used for cosmogenic nuclide exposure dating were internally consistent

evidencing a single, catastrophic mass movement event which occurred $4.54 \pm 0.27$ ka (re-calculated from Ballantyne and
Stone, 2004). Consequently, the boulders were very unlikely to have previously been exposed to cosmic rays or sunlight prior
to transport and deposition. Moreover, the large size of the flat-topped boulders (>2 x 2 x 2 m) and lack of finer sediment
matrix within the rock avalanche deposit, suggested that post-depositional movement or exhumation is unlikely. The
Torridonian sandstones are hard, cemented rocks (Stewart, 1984; Stewart and Donnellson, 1992) susceptible to granular
disintegration (e.g. Ballantyne and Whittington, 1987). Given its inland location, salt weathering is likely negligible. Kirkbride
and Bell (2010) estimated edge-rounding rates of ~3.3 mm/ka for a suite of Torridonian sandstone boulder samples from a
range of sites in NW Scotland under the warmer, wetter climates of the Holocene. A notably higher erosion rate of 12 mm/ka
was specifically determined for the Beinn Alligin rock avalanche. Kirkbride and Bell (2010) suggest that this higher erosion
rate, in comparison to the other sites, is likely due to inherited rock roundness caused by abrasion during the high-magnitude
depositional event. Additionally, minor differences in lithology cannot be ruled out (e.g. Twidale, 1982; Ford and Williams,
1989). Consequently, we consider the range ~3.3 to 12 mm/ka as a reasonable estimation of the Holocene erosion rate of the
Torridonian sandstone boulders that comprise the Beinn Alligin rock avalanche.
**4.      Methods**
A total of six rock samples were taken from the Torridonian sandstones in NW Scotland (Fig. 1). Three samples were taken
from three different road-cuttings of known age to calibrate the values of $\mu$ and $\overline{\sigma\varphi_0}$: ROAD01 (0.01 a), ROAD02 (57 a; Fig.
S1a), ROAD03 (44 a; Fig. S1b). Three further samples were taken from flat-topped, angular boulders that were part of the
Beinn Alligin rock avalanche deposit: BALL01, BALL02 and BALL03 (Fig. 1D). Portions of the original boulder or bedrock
sample were collected in the field in daylight and immediately placed into opaque, black sample bags. All samples were taken
from surfaces perpendicular to incoming sunlight to ensure that the daylight irradiation geometry was similar between
calibration and dating samples (cf. Gliganic et al. 2019).
*4.1     Luminescence measurements*
To calculate the environmental dose-rate throughout burial for each sample (Table 1), U, Th and K concentrations were
measured for ca. 80 g of crushed bulk sample using high-resolution gamma spectrometry. Internal dose-rates were calculated
assuming an internal K-content of $10 \pm 2$ % (Smedley et al. 2012) and internal U and Th concentrations of $0.3 \pm 0.1$ ppm and
$1.7 \pm 0.4$ ppm (Smedley and Pearce, 2016), in addition to the measured average grain sizes for each sample. Cosmic dose-rates
were calculated after Prescott and Hutton (1994). For measuring the luminescence depth profiles, sample preparation was

performed under subdued-red lighting conditions to prevent contamination of the luminescence signal. Rock cores ~7 mm in diameter and up to 20 mm long were drilled into the rock surface using an Axminster bench-top, pillar drill equipped with a water-cooled, diamond-tipped drillbit (~9 mm diameter). Each core was sliced at a thickness of ~0.7 mm using a Buehler IsoMet low-speed saw equipped with a water-cooled, 0.3 mm diameter diamond-tipped wafer blade. All slices were then mounted in stainless steel cups for luminescence measurements.

Luminescence measurements were performed on a Risø TL/OSL reader (TL-DA-15) with a $^{90}Sr/^{90}Y$ beta irradiation source. Heating was performed at 1°C/s and the rock slices were held at the stimulation temperature (i.e. 50, 150 and 225°C) for 60 s prior to IR stimulation to ensure all of the disc was at temperature before stimulating (cf. Jenkins et al. 2018). IRSL signals were detected in blue wavelengths using a photo-multiplier tube fitted with Schott BG-39 (2 mm thickness) and Corning 7-59 (2 mm thickness) filters. A MET-post-IR IRSL sequence (Table S1) was used to determine IRSL signals at three different temperatures (50, 150 and 225°C) successively, hereafter termed the $IR_{50}$, $pIRIR_{150}$ and $pIRIR_{225}$ signals. Luminescence depth profiles were determined for each core by measuring the natural signal ($L_n$) normalised using the signal measured in response to a 53 Gy test-dose ($T_n$), hereafter termed the $L_n/T_n$ signal. The IRSL signal was determined by subtracting the background signal (final 20 s, 40 channels) from the initial signal (0 – 3.5 s, 7 channels). The large test-dose (53 Gy) was used to reduce the impact of thermal transfer/incomplete resetting of the IRSL signal between measurements (after Liu et al. 2016).

$D_e$ values were determined for the shallowest disc and the deepest disc from one core of each sample to quantify the natural residual dose and saturation limit ($L_0$, Eq. 1), respectively. Fading rates (g-values, Aitken 1985) were determined for three discs of each sample and normalised to a $t_c$ of two days (Huntley and Lamothe 2001). The weighted mean and standard error of the g-values for all discs were 3.7 ± 0.4 %/dec. ($IR_{50}$), 1.0 ± 0.5 %/dec. ($pIRIR_{150}$) and 1.0 ± 0.5 %/dec. ($pIRIR_{225}$). The large uncertainties on the individual g-values measured were derived from uncertainty in the fit of the data, which is typical of fading measurements (e.g. Smedley et al. 2016). The fading rates were in line with previous measurements of IRSL signals (e.g. Roberts 2012; Trauerstein et al. 2014; Kolb and Fuchs 2018). Lehmann et al. (2019a) performed sensitivity tests of the shape of the luminescence depth profiles ($IR_{50}$) with a high and low g-value end-members and these simulations demonstrated that athermal loss of signal has a minimal impact upon the IRSL depth profile shape; thus, athermal loss (i.e. fading rates) was not considered in calculations.

Previous studies have shown that the $IR_{50}$ signal bleached faster than the pIRIR signals (Smedley et al., 2015). To test the inherent bleaching rates of the feldspars in our samples, artificial bleaching experiments were performed on seven discs from all six samples (n.b. these experiments do not test for variations in light attenuation with depth). All previously-analysed discs were given a 105 Gy dose, then subjected to different exposure times in a solar simulator (0 m, 1 m, 10 m, 30 m, 1 h, 4 h and 8 h) and the normalised luminescence signals ($IR_{50}$, $pIRIR_{150}$ and $pIRIR_{225}$) were measured (Fig. S2). The results show some variations after 1 m of solar simulator exposure. However, luminescence signals reduced to 2 – 6 % ($IR_{50}$), 6 – 11 % ($pIRIR_{150}$) and 14 – 22 % ($pIRIR_{225}$) of the unexposed light levels after 1 h and 1 – 2 % ($IR_{50}$), 2 – 3 % ($pIRIR_{150}$) and 4 – 7 %

(pIRIR$_{225}$) after 8 h. This indicates that within our samples the minerals emitting the IRSL signals (i.e. K-feldspar) have similar
inherent bleaching rates when exposed to longer durations of time (i.e. > 8 h in the solar simulator).
*4.2      Rock composition*
After luminescence measurements were performed, each rock slice (e.g. Fig. 2) was analysed to investigate potential changes
in rock composition with depth (inferred by opacity and grainsize). The average down-core grainsize of each sample was
measured under an optical microscope using *Infinity Analyze*. For each rock slice of an example core per sample, ten randomly-
selected grains were measured and the mean and standard deviation grainsize were calculated per core and plotted against the
core depths (Fig. 3B). Down-core red-green-blue (RGB) values were determined for each sample to investigate whether there
was any colour variation within the sample, and externally between samples; thus, providing a semi-quantitative tool to detect
variability in rock opacity (Meyer et al. 2018). Raster images of RGB were obtained for each rock slice using an EPSON
Expression 11000XL flatbed scanner at 1200 dpi resolution (e.g. Fig. S3). Mean and standard deviations of the RGB values
(e.g. Fig. 3A) for each rock slice were calculated using the *raster* package in R (version 2.9-23; Hijmans, 2019).
**5.      Results**
**5.1      Luminescence depth profiles**
The luminescence depth profiles (IR$_{50}$, pIRIR$_{150}$ and pIRIR$_{225}$) (Fig. 4) record bleaching fronts caused by sunlight exposure
for all of the known-age samples. The luminescence depth profile measured for core 3 of sample ROAD02 (Fig. 4 G,H,J) was
inconsistent with cores 1 and 2, giving high standard deviation values for the IR$_{50}$ (1.2), pIRIR$_{150}$ (1.1) and pIRIR$_{225}$ (0.9)
signals; thus, core 3 was removed from subsequent analysis (likely sample preparation issues related to drilling preservation
of the weathered surface). The luminescence depth profiles for the remaining replicate cores for all three samples were broadly
consistent within each rock sample with mean standard deviations ranging from 0.2 – 0.8.
The luminescence depth profiles (Fig. 4) for the IR$_{50}$ signal were consistent with the increasing sunlight exposure
ages for ROAD01 (0.01 a), ROAD03 (44 a) and ROAD02 (57 a), with bleaching fronts at 0.75 mm, 4.00 mm and 4.75 mm,
respectively (Fig. S5a). This indicated that the depth of the IR$_{50}$ bleaching front was dominated by exposure duration for the
known-age samples as expected. Similarly, the pIRIR$_{150}$ and pIRIR$_{225}$ bleaching fronts were shallower in sample ROAD01
(0.75 mm) compared to ROAD02 and ROAD03 (2.00 – 3.00 mm), reflecting the younger exposure duration of ROAD01.
However, the pIRIR$_{150}$ and pIRIR$_{225}$ bleaching fronts were at similar depths (2.75 and 3.00 mm and 2.00 and 2.50 mm
respectively) for both ROAD02 (57 a) and ROAD03 (44 a). This suggests that either another factor is influencing light
penetration with depth in these rocks (e.g. small differences in the orientation of the sampled rock faces; Fig. S1) or that the
pIRIR signals cannot resolve between a 57 a and 44 a exposure history (difference of only 13 a). Note that the inferred models
shown in Fig. 4 were fitted using the $\overline{\sigma\varphi_0}$ and μ values included in each figure. See Section 5.2 for further explanation of the
estimation of the model parameters.
The luminescence depth profiles measured for the unknown-age samples BALL02 and BALL03 using the IR$_{50}$,
pIRIR$_{150}$ and pIRIR$_{225}$ signals (Fig. 5) recorded bleaching fronts caused by sunlight exposure. Conversely, the luminescence

depth profile for sample BALL01 had saturated IRSL signals throughout the core and did not display any evidence of IRSL signal resetting with depth (Fig. 5A-C). A luminescence depth profile measured for a core drilled into the bottom surface (Bottom C1; Fig. 5A-C) confirmed that the bottom surface of BALL01 was also saturated. The lack of a bleaching front in sample BALL01 is difficult to explain as the sample was taken in daylight and had seemingly identical characteristics to samples BALL02 and BALL03 (i.e. no lichen-cover or coatings preventing light penetration in the rock). Although all the samples were similar in colour/opacity (Fig. 3A), the surface of sample BALL01 was coarser grained than BALL02 and BALL03 (Fig. 2; Fig. 3B). Studies have shown that coarser grain sizes are more susceptible to mechanical weathering via grain detachment induced by chemical weathering (Israelli and Emmanuel, 2018). Thus, although care was taken when sampling to mark the surface of the rock and to measure the length of the rock cores before and after slicing, it is possible that the luminescence depth profile (likely <10 mm based on BALL02 and BALL03) was lost during sampling and/or sample preparation due to the presence of a fragile weathering crust, potentially with a sub-surface zone of weakness (e.g. Robinson and Williams, 1987). Furthermore, field observations showed the presence of a rock pool on the surface of the boulder sampled for BALL01, which is not present on BALL02 and BALL03 (Fig. 1D); thus, there is also potential that the surface sampled for BALL01 had experienced enhanced chemical weathering via trickle paths draining the rock pool. These are commonly linked to a greater density of micro-cracks in the uppermost millimetres of the rock (Swantesson, 1989, 1992). Consequently, we did not derive exposure ages or erosion rates from BALL01. Where rock pools are likely on boulders, the highest rock surface should be sampled for luminescence techniques to avoid the potential for pooling or trickle paths.

## 5.2    Estimation of model parameters

To determine an apparent exposure age or erosion rate from the measured luminescence depth profiles, the variables that control the evolution of a luminescence depth profile in a rock surface must be parameterised; specifically, the dose-rate ($\dot{D}$) (see Section 4.1), saturation level ($D_0$), $\overline{\sigma\varphi_0}$ and $\mu$. $D_0$ was determined experimentally from saturated dose-response curves measured for the deepest rock slices of each sample. $\overline{\sigma\varphi_0}$ and $\mu$ were calibrated using Eq. (1) and the known-age samples (ROAD01, ROAD02 and ROAD03) of similar, suitable rock composition as determined by the down-core profiles of RGB and grainsize (Section 4.2). Note that ($\dot{D}$) is not considered in Eq. (1) but is used to determine an apparent exposure age or erosion rate and so needs to be measured for each sample (see Section 2). Down-core RGB values for all samples were internally consistent (Fig. 3A) as indicated by the relative standard deviation (RSD) range between 8 and 12 %. The down-core RGB values were also externally consistent between all samples (Fig. 3A), with the exception of the slightly darker-coloured sample ROAD01. However, measurements of grainsize (Fig. 3B) showed that the known-age sample ROAD02 (90 ± 23 µm) had a similar grainsize to the unknown-age samples BALL02 (73 ± 18 µm) and BALL03 (98 ± 19 µm), whereas ROAD01 (42 ± 9 µm) and ROAD03 (168 ± 56 µm) were finer and coarser grained, respectively. Given the similarity in colour and grainsize, it was considered most appropriate to calibrate $\overline{\sigma\varphi_0}$ and $\mu$ for the unknown age samples (BALL02 and BALL03) using known-age sample ROAD02.

293        The values of $\overline{\sigma\varphi_0}$ and μ were determined by fitting Eqn. (1) using the approach of Lehmann et al. (2019a). The

inferred model (Eq. 1) had a good fit to the measured data for all samples and signals (Fig. 4) and μ and $\overline{\sigma\varphi_0}$ were calculated
(Table 2; Fig. 6). For ROAD01, the parameters determined using the $IR_{50}$ (μ = 3.2 mm$^{-1}$, $\overline{\sigma\varphi_0}$ = 2.80e$^{-4}$ s$^{-1}$), pIRIR$_{150}$ (μ = 3.1
mm$^{-1}$, $\overline{\sigma\varphi_0}$ = 3.27e$^{-5}$ s$^{-1}$) and pIRIR$_{225}$ (μ = 3.0 mm$^{-1}$, $\overline{\sigma\varphi_0}$ = 2.88e$^{-5}$ s$^{-1}$) signals were broadly consistent. For ROAD02, the
parameters differed between the $IR_{50}$ (μ = 2.1 mm$^{-1}$, $\overline{\sigma\varphi_0}$ = 6.67e$^{-6}$ s$^{-1}$), pIRIR$_{150}$ (μ = 1.5 mm$^{-1}$, $\overline{\sigma\varphi_0}$ = 1.73e$^{-8}$ s$^{-1}$) and pIRIR$_{225}$
(μ = 2.8 mm$^{-1}$, $\overline{\sigma\varphi_0}$ = 9.01e$^{-8}$ s$^{-1}$) signals, but the values for each signal were broadly similar to the equivalent values
determined for ROAD03 using the $IR_{50}$ (μ = 2.7 mm$^{-1}$, $\overline{\sigma\varphi_0}$ = 1.56e$^{-5}$ s$^{-1}$), pIRIR$_{150}$ (μ = 1.5 mm$^{-1}$, $\overline{\sigma\varphi_0}$ = 3.80e$^{-8}$ s$^{-1}$) and
pIRIR$_{225}$ (μ = 1.4 mm$^{-1}$, $\overline{\sigma\varphi_0}$ = 1.70e$^{-8}$ s$^{-1}$) signals. Given the similarity of $\overline{\sigma\varphi_0}$ and μ determined using all three IRSL signals
for ROAD02 and ROAD03 and the difference in grainsizes (Fig. 3B), it suggests that grainsize has a minimal impact upon the
attenuation of light into a rock surface in comparison to other factors (e.g. mineralogy, surficial coatings). The μ values for
samples ROAD01, ROAD02 and ROAD03 determined using the $IR_{50}$ signal in this study were comparable to μ values in
existing literature for sandstones using K-feldspar e.g. 3.06 mm$^{-1}$ (Ou et al. 2018). For sample ROAD01, μ and $\overline{\sigma\varphi_0}$ were
similar for all three IRSL signals with large uncertainties (Fig. 6A-C) which is likely related to the shorter exposure age of this
sample (0.01 a). The finer grain size and darker rock opacity of sample ROAD01 in comparison to ROAD02 and ROAD03
likely explained the larger values of μ (i.e. greater light attenuation with depth into the rock surface).
**5.3**       **Apparent exposure ages and erosion rates**
Luminescence exposure ages were determined from the luminescence depth profiles using μ and $\overline{\sigma\varphi_0}$ derived from sample
ROAD02 for each of the IRSL signals (Table 3). For BALL03, the $IR_{50}$ (387 ± 103 a), pIRIR$_{150}$ (296 ± 54 a) and pIRIR$_{225}$ (362
± 49 a) signals all gave luminescence exposure ages in agreement within uncertainties. For BALL02, the three signals were
inconsistent with one another. The pIRIR$_{225}$ signal (263 ± 30 a) was consistent with BALL03, but the $IR_{50}$ (8 ± 2 a) and
pIRIR$_{150}$ (66 ± 16 a) signals for BALL02 were younger than BALL03. All apparent exposure ages based on the different
luminescence signals were at least one order of magnitude younger than the apparent exposure age based cosmogenic nuclide
dating (4.54 ± 0.27 ka; Ballantyne and Stone, 2004). This was likely because erosion over time in this wet, temperate climate
has removed material from the surface of the rock and created shallower luminescence depth profiles in comparison to a non-
eroding profile; thus, the luminescence depth profile is dependent upon both exposure age and the erosion rate (Sohbati et al.
2018; Lehmann et al. 2019a).

319        To test whether erosion rates could be determined for the Beinn Alligin boulders from the luminescence depth

profiles, we performed erosion rate modelling following the inversion approach of Lehmann et al. (2019) and constrained by
the re-calculated cosmogenic nuclide age (Ballantyne and Stone, 2004). This approach defines an erosion history that follows
a step function with an initial period of zero erosion, followed by an immediate increase to a constant erosion rate at a defined
time. It attempts to recover parameter combinations (erosion rate and timing of erosion initiation) that are both consistent with
the cosmogenic nuclide concentration and produce modelled luminescence profiles that match observations. For BALL02,
both the $IR_{50}$ and pIRIR$_{150}$ signals suggested that the system had approached a steady-state with erosion rates of 66 mm/ka
(IR$_{50}$) and 9 mm/ka (pIRIR$_{150}$) applied over time periods >73 a and 593 a, respectively. However, the pIRIR$_{225}$ signal suggested
a transient erosion state, where the luminescence signal could be derived from numerous pairs of erosion rates and initiation
times from a maximum erosion rate of 310 mm/ka over a minimum time interval of 4 a to a minimum erosion rate of 12 mm/ka
over a minimum time interval of 90 a. All three IRSL signals from sample BALL03 consistently suggested a system undergoing
a transient response to erosion, which was consistent with the pIRIR$_{225}$ signal of BALL02 (Fig. 7, Table 3). The IR$_{50}$ signal
for BALL03 derived a maximum erosion rate of 460 mm/ka over a minimum time interval of 3 a and a minimum erosion rate
of 6 mm/ka over a minimum time interval of 231 a. The pIRIR$_{150}$ signal for BALL03 derived a maximum erosion rate of 100
mm/ka over minimum time interval of 19 a and a minimum erosion rate of 14 mm/ka over a minimum time interval of 137 a.
The pIRIR$_{225}$ signal for BALL03 derived a maximum erosion rate of 180 mm/ka over a minimum time interval of 4 a and a
minimum erosion rate of 11 mm/ka over a minimum time interval of 73 a.
At face value, the fit of the inferred erosion model to the experimental data for BALL02 using the IR$_{50}$ (Fig. 5D) and
pIRIR$_{150}$ (Fig. 5E) signals is better than the equivalent fits for BALL02 using the pIRIR$_{225}$ signal (Fig. 5F) and BALL03 using
the IR$_{50}$ (Fig. 5G), pIRIR$_{150}$ (Fig. 5H) and pIRIR$_{225}$ (Fig. 5I) signals. In the latter cases, the inferred erosion model is shallower
than the experimental data. This could suggest that the $\overline{\sigma\varphi_0}$ and µ values were inaccurate, i.e. the attenuation of light with
depth into the rock surface is lower in BALL02 (pIRIR$_{225}$ signal) and BALL03 (IR$_{50}$, pIRIR$_{150}$ and pIRIR$_{225}$ signals) than
estimated by ROAD02. A possible explanation for this is that the surface of the roadcut sampled by ROAD02 (Fig. S1a) was
orientated slightly differently to the Beinn Alligin rock avalanche boulders sampled by BALL02 and BALL03 (Fig. 1D),
relative to the incoming sunlight (e.g. Gliganic et al. 2019). However, if the orientation of the known-age roadcut samples was
even slightly inconsistent with the unknown samples, we would expect these inconsistencies to manifest similarly in all three
MET signals for BALL02 and BALL03, which was not observed here. A factor that is common to the less well fitting profiles
is that they define transient erosion states. This suggests that these surfaces experienced complex erosional histories over time
whereby the erosion rate was time-varying. Consequently, it is possible that surficial weathering products may have changed
in thickness and composition over time, which in turn could slightly vary the attenuation of light (Meyer et al. 2018; Luo et al.
2018), meaning that the calibration of $\overline{\sigma\varphi_0}$ and µ from ROAD02 here introduced uncertainty into the inferred erosion model
as it was not time-varying. It is also possible that sample-specific measurements of $\overline{\sigma\varphi_0}$ and µ (e.g. Ou et al. 2018), rather than
calibration from known-age samples, could reduce the uncertainty introduced by time-varying light attenuation. However,
further investigation is required into the physical mechanisms of time-varying light attenuation in the context of surficial
weathering and subsequent erosion, and the impacts upon inferred transient erosion rates.
**6. Discussion**
**6.1 Luminescence depth profiles for the Beinn Alligin rock avalanche**
Despite the similarity in rock opacity, grainsize, aspect and exposure history, the luminescence depth profiles for samples
BALL02 and BALL03 from the Beinn Alligin rock avalanche were inconsistent (Fig. 5). We consider it unlikely that this lack
of consistency was caused by local variations in erosion rates (e.g. due to microclimate, aspect etc; Hall et al. 2005, 2008) as

there were discrepancies between all three IRSL signals of BALL02. We would expect local erosion rate variations between samples to be consistently recorded across each of the IRSL signals, assuming the model parameterisation ($\mu$ and $\overline{\sigma\varphi_0}$) were accurate. Specifically, and with all other things being equal, a locally-variable erosion rate would translate the bleaching front(s) closer to the rock surface by a proportionally consistent amount for each signal of a given sample.

Analysis of the rock opacity with depth (Section 4.2; Meyer et al. 2018) showed that sample BALL02 was more positively skewed towards darker colours than ROAD02 and BALL03 (Fig. S3, S4), with higher surficial values caused by Fe-staining. Fe-staining can occur on rock surfaces with seasonal rock pools and trickle paths (Swantesson, 1989, 1992). The presence of a thin Fe-coating (<1 mm) on the rock surface would have changed the intensity and wavelength of the net daylight flux received by individual grains (e.g. Singhvi et al.,1986; Parish, 1994) and likely increased light attenuation with depth (e.g. Meyer et al. 2018; Luo et al. 2018). Consequently, the parameterisation of $\mu$ and $\overline{\sigma\varphi_0}$ derived from sample ROAD02 would be inaccurate for BALL02. Interestingly, the similarity between BALL02 and BALL03 for the pIRIR$_{225}$ signal suggests that the presence of an Fe-coating altered the attenuation of the IR$_{50}$ and pIRIR$_{150}$ signals to a lesser extent than the pIRIR$_{225}$ signal, but the reasons for this requires further investigation. The application of the MET-pIRIR rather than just the stand-alone IR$_{50}$ signal protocol provided a major advantage as it identified samples where the parameterisation of $\mu$ and $\overline{\sigma\varphi_0}$ from known-age samples was complicated by factors such as surficial weathering coatings. Beyond this, it is possible that the MET-pIRIR protocol may be useful in identifying complex burial or exposure histories of rocks, similar to those that have been reported in previous studies but solely using the IR$_{50}$ signal (e.g. Freiesleben et al. 2015; Brill et al. 2021). There is also potential to explore whether the different temperature IRSL signals of the MET protocol record different states of erosion (i.e. steady or transient states) within the same rock surface, whereby the post-IR IRSL signals that are attenuated greater would be more susceptible to transient states of erosion in comparison to the lower temperature signals, which measure luminescence depth profiles to greater depths within the rock surface.

The boulders from the Beinn Alligin rock avalanche have been subject to a temperate climate for the last ~4 ka. The luminescence depth profiles from the boulders demonstrated that on these timeframes and under these climatic conditions the technique was an erosion-meter, rather than a chronometer, as expected (Sohbati et al. 2018; Lehmann et al. 2019a). Lehmann et al. (2019a) noted that two of their samples, uncorrected for erosion, gave apparent luminescence exposure ages of ca. 640 a and <1 a compared to apparent cosmogenic nuclide ages of ca. 16.5 ka and 6.5 ka, respectively. It has thus been inferred that erosion rates >1 mm/ka can make interpretation of luminescence depth profiles in terms of an exposure age difficult without accurately constraining the erosion rate (Sohbati et al., 2018; Lehmann et al., 2018). This is consistent with the underestimation of luminescence exposure ages measured here for the Beinn Alligin rock avalanche (Table 3), which have been independently-dated to 4.54 ± 0.27 ka using cosmogenic nuclides (Ballantyne and Stone, 2004). Consequently, luminescence depth profiles for the Beinn Alligin rock avalanche can only be inferred in terms of erosion rates.

**6.2**     **Luminescence as an erosion-meter**

The numerical approach of Lehmann et al. (2019a) exploits the different sensitivities of the luminescence (short-term) and cosmogenic nuclide (longer-term) techniques to erosion to infer erosion histories (steady state and transient over time) for rock surfaces. Their modelling shows that the higher erosion rates (>100 mm/ka) can only be sustained over shorter time durations (up to decadal) while at the same time being consistent with cosmogenic nuclide measurements. For BALL03, transient erosion rates were derived using the $IR_{50}$ (6 - 460 mm/ka), $pIRIR_{150}$ (14 - 100 mm/ka) and $pIRIR_{225}$ (11 - 180 mm/ka) signals. These modelled transient erosion rates were broadly comparable to erosion rates inferred from luminescence depth profiles over comparable timeframes in previous studies: (i) rates between <0.038 ± 0.002 and 1.72 ± 0.04 mm/ka for glacial boulders and landslides (granite gneiss, granodorite and quartzite) in the Eastern Pamirs, China (Sohbati et al. 2018); and (ii) between 3.5 ± 1.2 mm/ka and 4,300 ± 600 mm/ka for glacially-modified, granitic bedrock in the French Alps (Lehmann et al., 2019b). This latter study modelled higher erosion rates (>100 mm/ka) over timescales from $10^1$ to $10^3$ a and lower erosion rates (<100 mm/ka) over longer time scales of $10^3$ to $10^4$ a. However, this comparison between modelled erosion rates does not account for the primary role that lithology has on weathering (e.g. Twidale, 1982; Ford and Williams, 1989). The sampled boulders in our study were composed of Torridonian sandstone, which has been reported to undergo granular disintegration (e.g. Ballantyne and Whittington, 1987), particularly around edges, and thus may have experienced higher erosion rates than the crystalline rocks (e.g. gneiss, granite) used in the studies of Sohbati et al. (2018) and Lehmann et al., 2019b.

A major advantage of applying this new erosion-meter technique to boulders of the Beinn Alligin rock avalanche was the existing constraints on Holocene erosion rates (~3.3 to 12 mm/ka) for Torridonian sandstones in NW Scotland inferred from boulder edge roundness measurements (Kirkbride and Bell, 2009). The long-term erosion rates inferred from luminescence depth profiles were consistent with the estimates provided by measuring the boulder-edge roundness, when considering the differing approaches and assumptions of each method. Firstly, the sampling approach for the luminescence depth profiles targeted the flat-top surface of the boulders where granular disintegration would have been reduced relative to the boulder edges and corners. Thus, the boulder-edge roundness based erosion rates provided an upper constraint on the long-term erosion rate experienced by the boulders. Finally, the boulder-edge roundness measurements assumed steady-state erosion and could not identify the potential for a transient state of erosion, whereas the approach of Lehmann et al. (2019a,b) inferred some transient state of erosion (Table 3). Consequently, it is notable that the lower range of the transient erosion rates derived here using the $IR_{50}$ (6 - 460 mm/ka), $pIRIR_{150}$ (14 - 100 mm/ka) and $pIRIR_{225}$ (11 - 180 mm/ka) signals were broadly consistent with the steady-state erosion rate derived from boulder edge roundness measurements for the Torridonian sandstones (in the range of ca. 3.3 to 12.0 mm/ka). Lehmann et al. (2019b) noted that their modelled steady-state erosion rates were one to two orders of magnitude higher than suggested by a global compilation of bedrock surface erosion rates based on [10]Be (Portenga and Bierman, 2011), and measurements of upstanding, resistant lithic components (ca. 0.2 – 5.0 mm/ka) in crystalline rock surfaces in Arctic Norway (André, 2002). The authors inferred that shorter-term erosion rates derived from luminescence measurements were higher than the longer-term averages due to the stochastic nature of weathering impacting upon shorter-term erosion rates, this is also suggested by the data presented here. These stochastic processes (i.e. varying over time) will be

controlled by the in-situ weathering rates, which provided the material for erosion. For bare rock surfaces in wet, temperate climates, weathering rates are primarily driven by rock-type and moisture availability (i.e. precipitation) (Hall et al. 2012; Swantesson, 1992). The Torridonian sandstones are hard, cemented rocks (Stewart, 1984; Stewart and Donnellson, 1992) susceptible to granular disintegration (e.g. Ballantyne and Whittington, 1987), which may have been stochastic in nature due to changing moisture availability for chemical weathering over time (Hall et al. 2012; Swantesson, 1992). Although Torridonian sandstones are unlikely to be prone to frost shattering due to their low permeability and porosity (Lautridou, 1985; Hudec 1973 in Hall et al. 2012), cracks, faults and joints in the rock may have facilitated stochastic physical weathering (Swantesson 1992; Whalley et al. 1982), but little field evidence of this was preserved.

The modelled erosion histories that we have calculated here using the luminescence erosion-meter for samples BALL02 and BALL03 would have had a minimal effect upon the cosmogenic nuclide exposure age ($4.54 \pm 0.27$ ka; Ballantyne and Stone, 2004). Only the steady-state erosion rate of 66 mm/ka inferred for BALL02 using the $IR_{50}$ signal, when applied for durations exceeding 1 ka, would increase the exposure age to any great degree. For example, when the steady-state erosion rate of 66 mm/ka was applied for 0.1 ka, the corrected cosmogenic nuclide exposure age would have been 4.58 ka and, when the same erosion rate was applied for 1 ka it would have been 4.99 ka; these corrected ages were consistent within $\pm 2 \sigma$ uncertainties of the uncorrected age of $4.54 \pm 0.27$ ka (reported at 1σ: Ballantyne and Stone, 2004). The higher, transient erosion rates inferred for BALL03 were all applied for such a short period of time (e.g. Table 3) that they had a minimal effect on the cosmogenic nuclide exposure age.

Based on the long-term erosion rates derived here, the boulder sampled for BALL02 would have lost a total of 300 mm ($IR_{50}$), 41 mm ($pIRIR_{150}$) and 54 mm ($pIRIR_{225}$) from the surface over 4.54 ka, while the long-term erosion rates determined for BALL03 suggested that the boulder surface would have lost 27 mm ($IR_{50}$), 64 mm ($pIRIR_{150}$) and 50 mm ($pIRIR_{225}$). All of these values (except for the $IR_{50}$ signal of BALL02) were broadly consistent with field observations of quartz protrusions on the surface of boulders >2 x 2 x 2 m that were densely distributed within the rock avalanche feature (Fig. 1). Alternatively, the maximum (shorter-term) erosion rate end members of the transient erosion histories would have removed 1407 mm (BALL02, $pIRIR_{225}$), 2088 mm (BALL03, $IR_{50}$), 454 mm (BALL03, $pIRIR_{150}$) and 817 mm (BALL03, $pIRIR_{225}$) from the boulder surface over the 4.54 ka. These large values were inconsistent with field evidence and so indicative of the transient state of erosion where high erosion rates were only sustained over short periods of time.

### 6.3     Late Holocene erosion history

The transient state of erosion inferred by the rock luminescence measurements reflected the stochastic nature of erosion over the last 4 ka, where a lower time-averaged erosion rate was interrupted by discrete intervals of higher time-averaged erosion rates. Rock weathering would have been dependent upon a variety of factors, primarily rock type and climate (Merrill 1906). The main constituents of the Torridonian sandstones are quartz, alkali and plagioclase feldspar (mostly albite), with precipitated quartz cementing the rock being resistant to chemical weathering (Stewart and Donnellan, 1992). However, the

red colouring of the sandstones represents the presence of Fe within the rock (Stewart and Donnellsan, 1992), which is prone to chemical weathering via oxidation and reduction. Field evidence of quartz grain protrusions on the rock surfaces (Fig. 1) indicated that granular disintegration, rather than flaking or shattering, was the likely weathering process that produced material for erosion on these hard boulders (e.g. Swantesson, 1992). This is also supported by a lack of shattered material surrounding the large sampled boulders (and in fact on much of the Beinn Alligin rock avalanche deposit), despite the presence of dense, low-level vegetation surrounding the boulders (e.g. Fig. S6). Granular disintegration has been reported as responsible for much of the general microweathering in the temperate climate of Southern and Central Sweden during the Holocene (e.g. Swantesson, 1992).

Given the coupling between precipitation, temperature and erosion (e.g. Reiners et al., 2003; Portenga and Bierman, 2011), the stochastic processes producing transient erosion can relate to varying environmental conditions (Hall et al. 2012; Swantesson, 1992; Whalley et al. 1982). In an environment where moisture is abundant due to high precipitation rates (e.g. for NW Scotland, annual precipitation rates between 1981 and 2010 were ca. 2,300 mm/a; Met Office, 2021), chemical weathering dominates; this is also reported for Holocene weathering processes in Sweden (Swantesson, 1989, 1992). Moisture availability, rather than temperature, is the limiting factor as studies have reported the presence of chemical weathering in natural settings subject to sub-zero temperatures (e.g. northern Canada, Hall, 2007; Antarctica, Balke et al. 1991). Proxy evidence from across the British Isles records variability in temperature and precipitation rates over the last 4.5 ka, where key increases in precipitation occurred at 2,750, 1,650 and 550 cal. years BP correlated to Bond cycles (Charman, 2010). Thus, the transient erosion rates measured from boulders of the Beinn Alligin avalanche were potentially a representation of the fluctuations in moisture availability experienced over the last 4.5 ka. Such processes can only be inferred from luminescence depth profiles as they are sensitive to changing erosion on shorter timeframes than all other techniques.

**7.    Conclusion**

This study applies new rock luminescence techniques to a well-constrained test scenario provided by flat-topped boulders from the Beinn Alligin rock avalanche in NW Scotland (a wet, temperate climate), which are lithologically consistent (Torridonian sandstones), have known-age road-cuts for parameterisation of $\mu$ and $\overline{\sigma\varphi_0}$, have known cosmogenic nuclide exposure ages (4.54 ± 0.27 ka) and independently-derived Holocene erosion rates (ca. 3.3 to 12.0 mm/ka). Applying the rock luminescence techniques for exposure dating underestimated the cosmogenic nuclide ages for the Beinn Alligin rock avalanche expected due to high erosion rates (as supported by field evidence of quartz grain protrusions on the rock surfaces). Alternatively, the erosion rates determined were consistent with expected rates that were independently measured in the field from boulder-edge roundness when considering the relative timescales of the time-averaged erosion rates. The findings show that the luminescence erosion-meter has the resolution and sensitivity required to detect transient erosion of boulders over the last 4.5 ka. The transient erosion rates reflect the stochastic nature of erosional processes in the wet, temperate region of NW Scotland, likely in response to the known fluctuations in moisture availability (and to a lesser extent temperature), which control the extent of chemical weathering. This study demonstrates that the luminescence erosion-meter has huge potential for inferring

erosion rates on sub-millennial scales for both steady-state and transient states of erosion (i.e. stochastic processes), which is currently impossible with other techniques. Larger sample populations and careful sampling of rock surfaces (avoiding the potential for rock pools and trickle paths) will likely be key for accurate measurements of landscape-scale erosion, and the use of a MET-pIRIR protocol (50, 150 and 225 °C) is advantageous as it can identify samples suffering from complexities that would not have been observed using only the standard IRSL signal measured at 50 °C, such as that introduced by within-sample variability (e.g. surficial coatings).

**Author contributions**

RS, DS and RSJ were involved in project conception. RS, DS, RSJ and SB performed the field sampling. RS, DS, JB and GJ performed the measurements, analysis and interpretations. All authors contributed to the writing of the manuscript, including the preparation of figures.

**Acknowledgments**

Field and laboratory work was funded by Durham University Department of Geography Research Development Fund to DS. The rock luminescence equipment in the Liverpool Luminescence Laboratory was funded by a Royal Society Research Grant (RG170194) to RKS. DS is supported by a NERC Independent Research Fellowship NE/T011963/1. We thank Benjamin Lehmann, an anonymous reviewer and the Associate Editor Jim Feathers for their constructive comments which improved this manuscript.

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

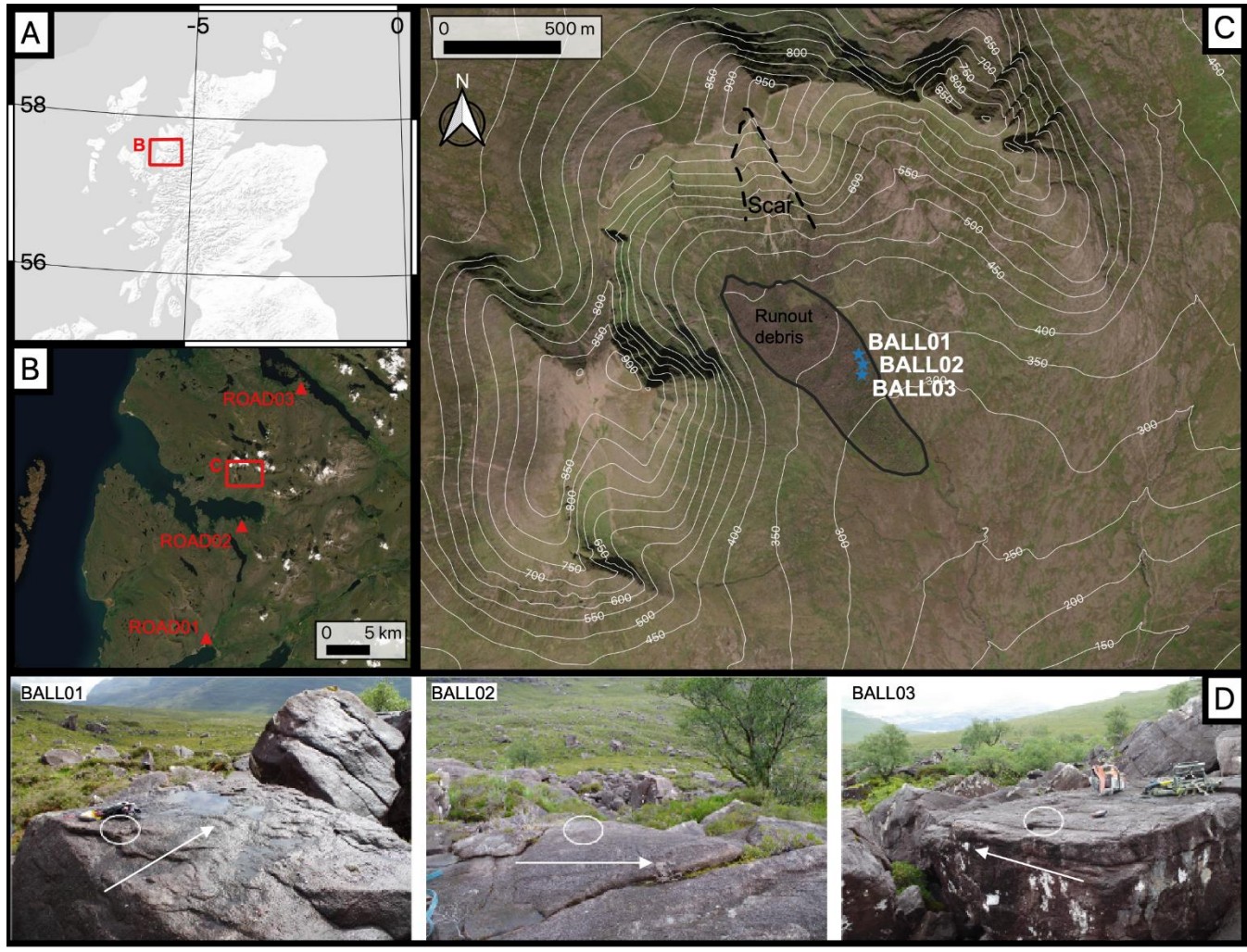

Figure 1. Location of the Beinn Alligin rock avalanche (57°35'N, 05°34'W) and roadcut sections in NW Scotland (A,B). Sample sites
on the rock avalanche deposit (C). Photographs of flat-topped boulders sampled and the general rock avalanche flow direction (white
arrow) for BALL01, BALL02 and BALL03 (D). The backgrounds used are ESRI World Terrain Base (A) and ESRI World Imagery
(B,C). Contains OS data © Crown copyright and database right (2021). Scar and runout debris locations mapped in (C) follow
Ballantyne and Stone (2004).


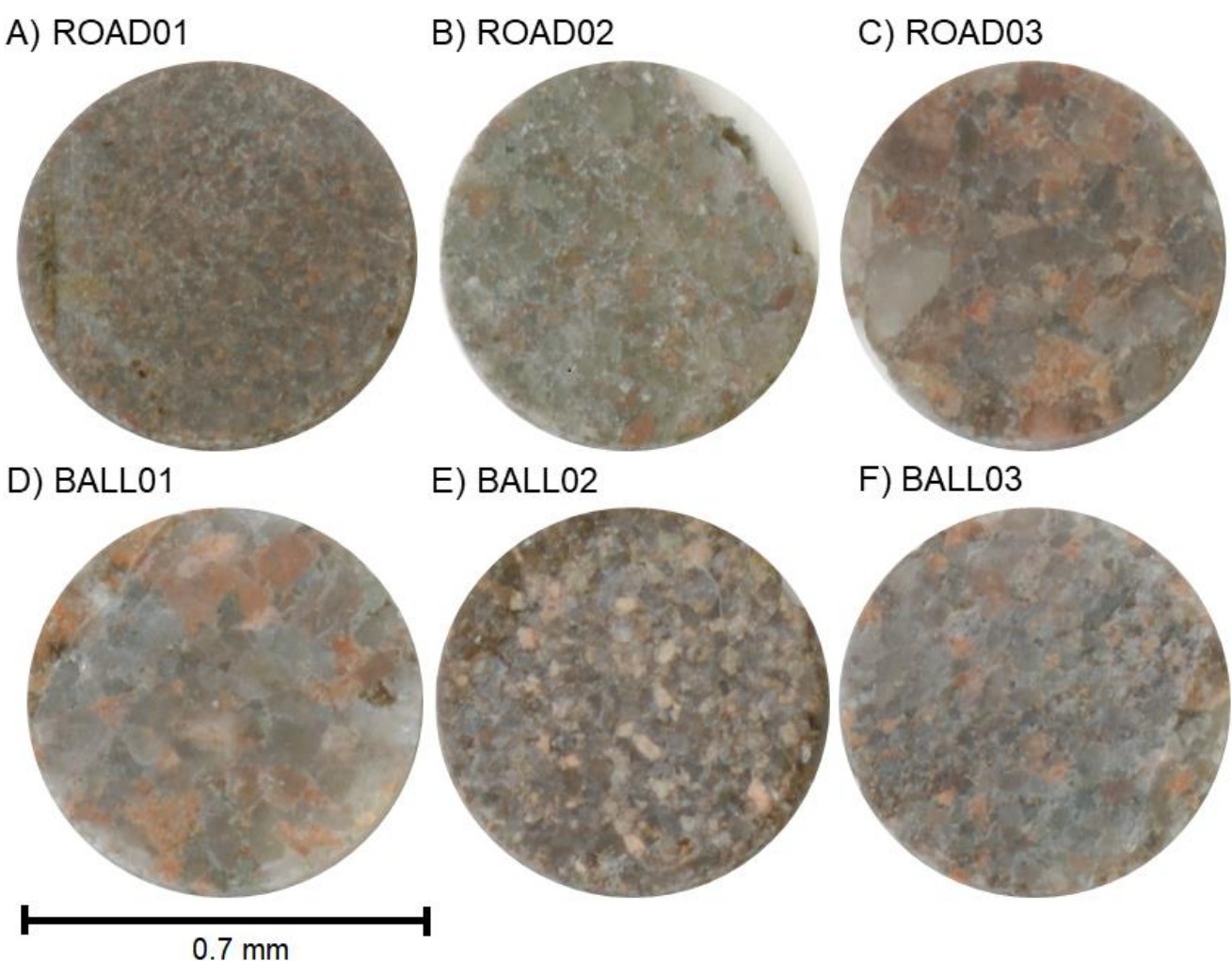

**Figure 2. Images of example rock slices (0.7 mm diameter) for each sample taken using the EPSON Expression 11000XL flatbed**
**scanner.**

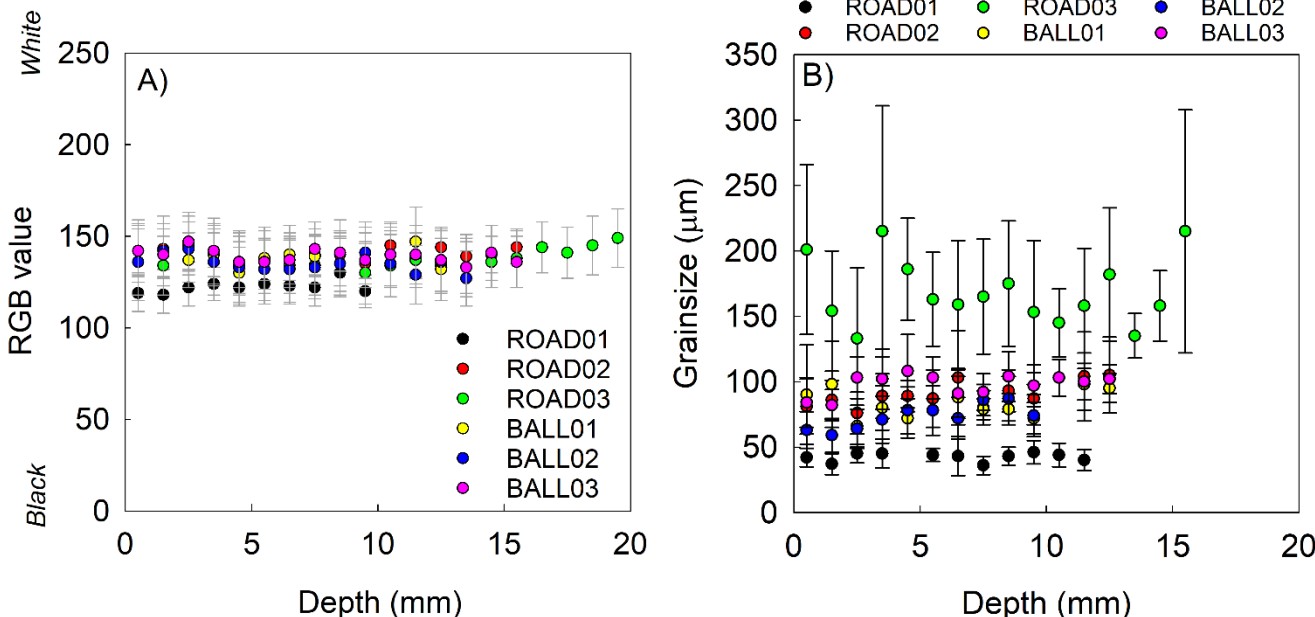

**Figure 3. (A) RGB values (0 = black and 255 = white) and (B) grainsize for each sample, calculated as the mean (± standard deviation)**
**of the slices at each depth in all of the replicate cores analysed. Note that the RGB values and grainsize measurements were not**
**derived from exactly the same cores, but example cores for each sample.**

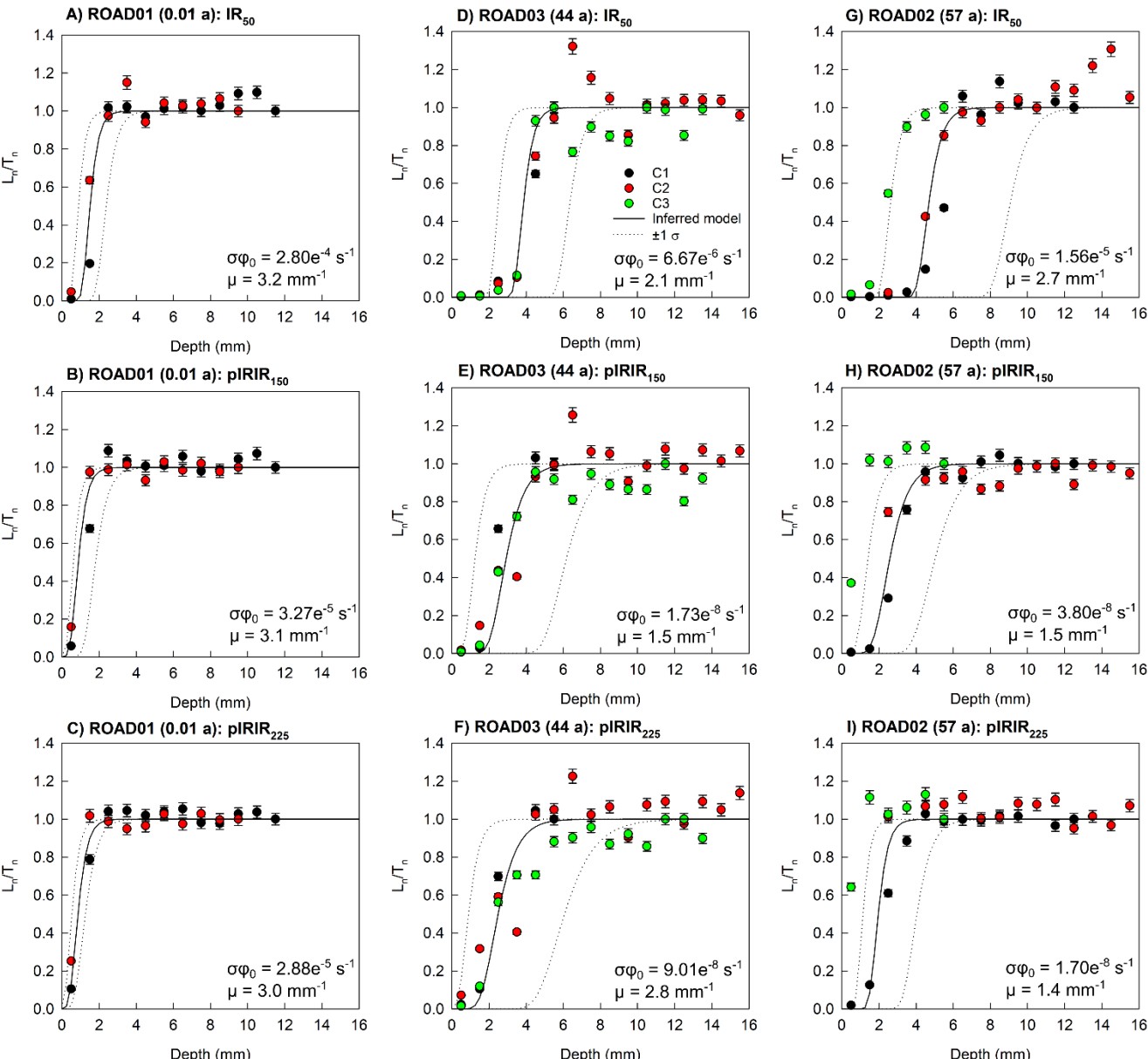

Figure 4. Presented in age-order are the IRSL-depth profiles for each of the three replicate cores analysed per sample using the $IR_{50}$ (A,D,G), $pIRIR_{150}$ (B,E,H) and $pIRIR_{225}$ (C,F,I) signals for samples ROAD01 (0.01 a; A-C), ROAD03 (44 a; D-F) and ROAD02 (57 a; G-I). All of the raw $L_n/T_n$ data presented in this figure (Table S2-S4) were normalised individually for each core, and subsequent analysis uses the data in this format. The black line shown is the inferred model that was fitted to derive the corresponding $\overline{\sigma\varphi_0}$ and $\mu$ values included in each figure. The dotted lines show the corresponding fits modelled using the $\pm 1\ \sigma\ \overline{\sigma\varphi_0}$ and $\mu$ values (Table 2). Note that core 3 of ROAD02 was not considered for fitting.

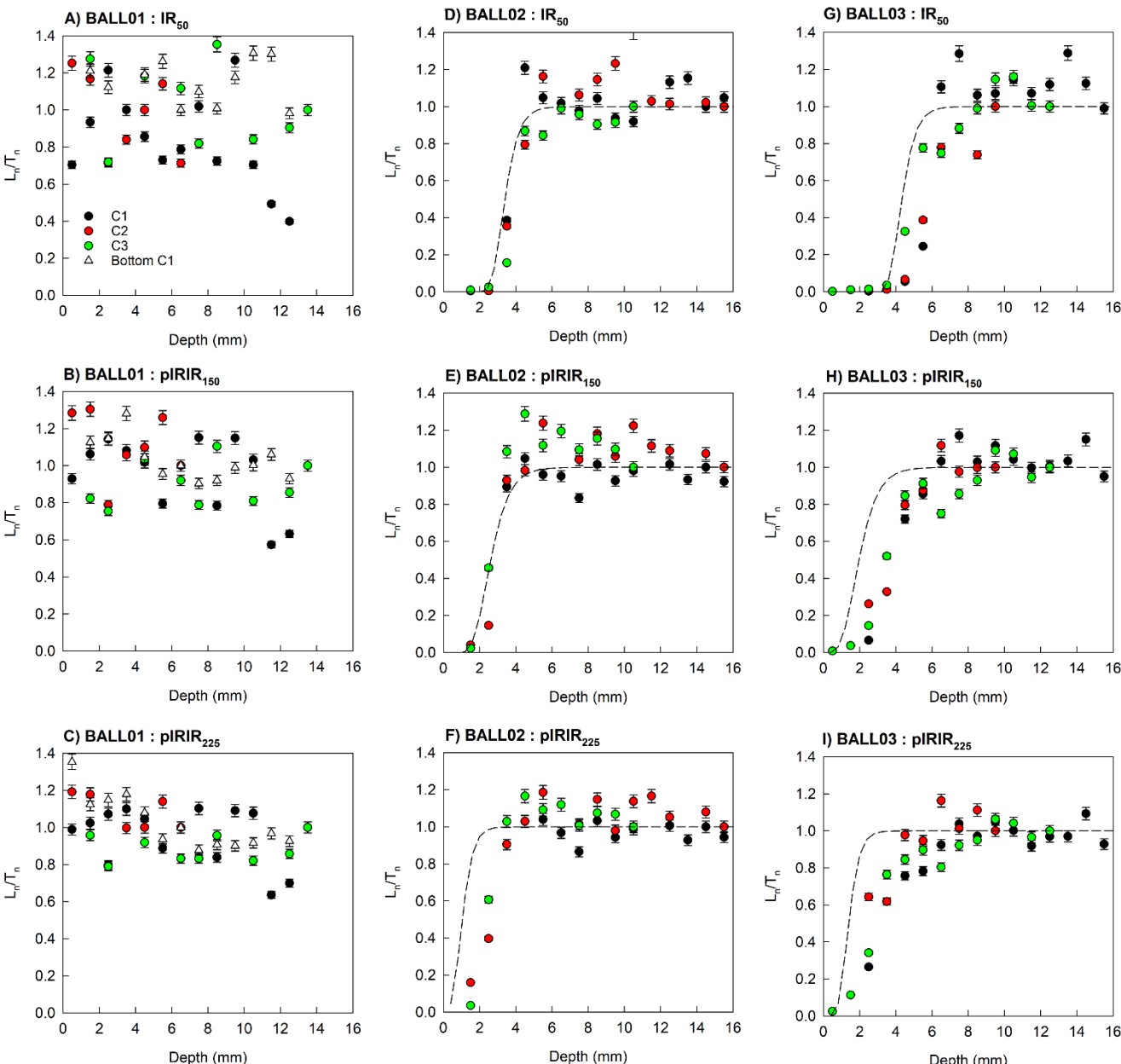

Figure 5. IRSL-depth profiles for each replicate cores analysed using the $IR_{50}$ (A,D,G), $pIRIR_{150}$ (B,E,H) and $pIRIR_{225}$ (C,F,I) signals for samples BALL01 (A-C), BALL02 (D-F) and BALL03 (G-I). All of the raw $L_n/T_n$ data (Table S5-S7) were normalised individually for each core, and subsequent analysis uses the data in this format. The dashed line is the inferred erosion model for each luminescence depth profile derived from the probability distributions shown in Fig. 7, where erosion rates are included in Table 3.

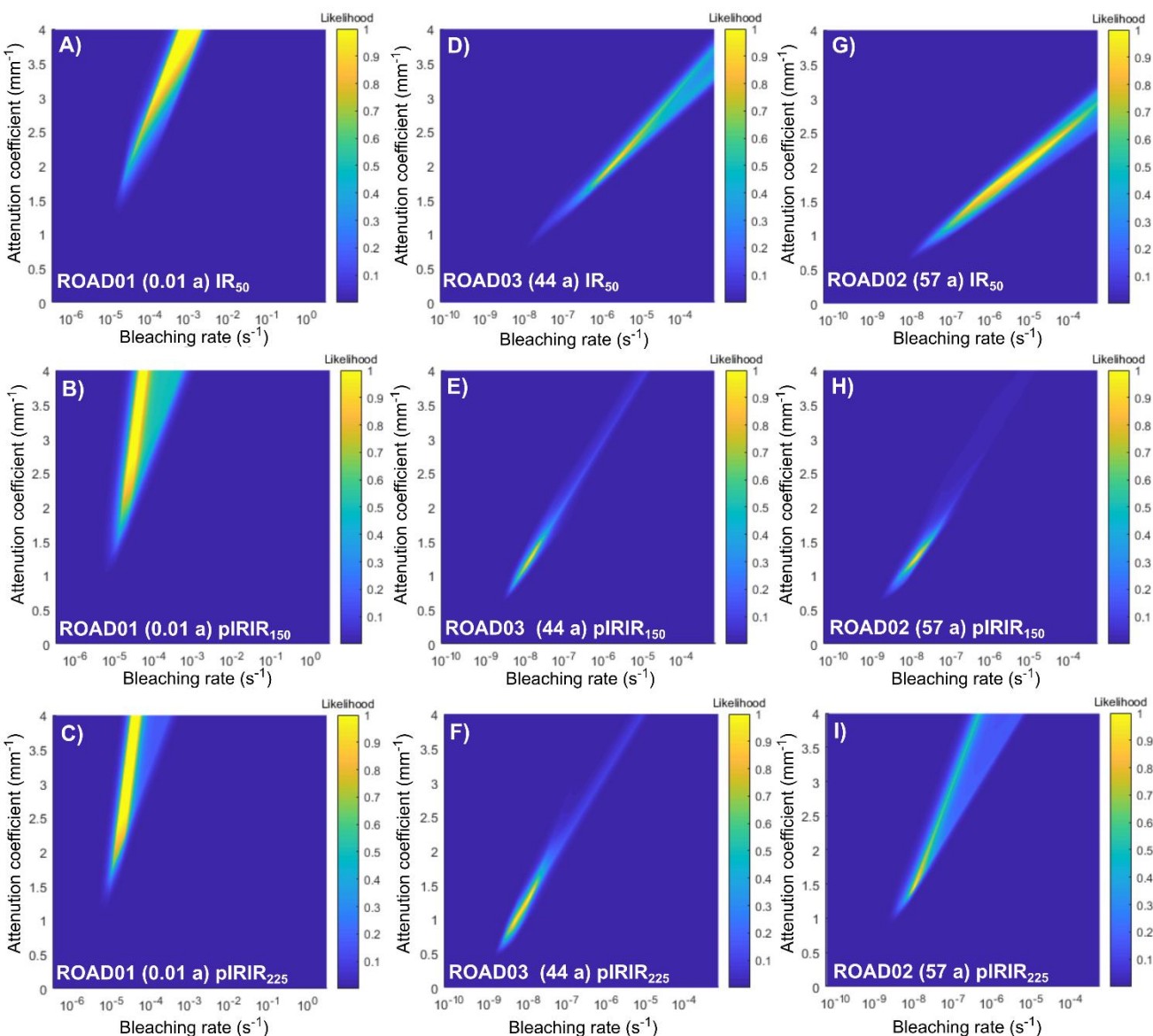

**Figure 6. Presented in age-order is the relationship between $\overline{\sigma\varphi_0}$ and µ parameters for ROAD01 (A-C), ROAD03 (D-F) and**
**ROAD02 (G-I) using the IR$_{50}$ (A,D,G), pIRIR$_{150}$ (B,E,H) and pIRIR$_{225}$ (C,F,I) signals using the approach of Lehmann et al. (2018).**

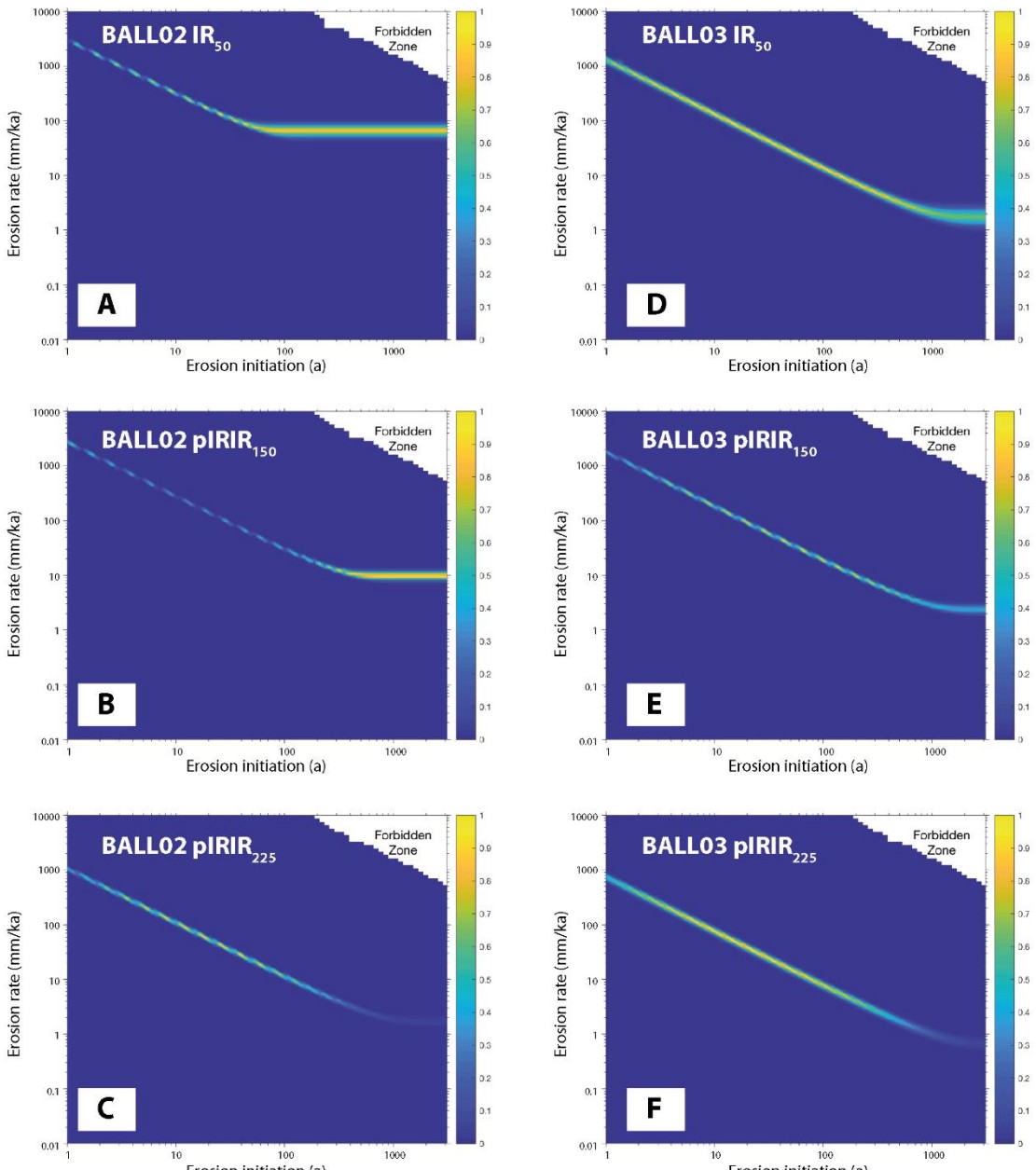

**Figure 7. Probability distributions inverted from the respective plots of luminescence depth profiles derived from the inversion**
**results (using the approach of Lehmann et al. 2019a) for samples BALL02 (A-C) and BALL03 (D-F) using the IR$_{50}$, (A,D), pIRIR$_{150}$**
**(B,E) and pIRIR$_{225}$ (C,F) signals. The x-axis plots the time interval of the erosion rate initiation. Forbidden zones define the range**
**of solutions with high erosion rates and durations that are not feasible within the bounds of the experimental [10]Be and luminescence**
**data.**

720

Table 1. Luminescence results for the rock slices analysed in this study. Environmental dose-rates were determined using high-resolution gamma spectrometry. The dose-rates were calculated using the conversion factors of Guerin et al. (2011) and alpha (Bell, 1980) and beta (Guerin et al. 2012) dose-rate attenuation factors. An internal K-content of 10 ± 2 % (Smedley et al. 2012) and internal U and Th concentrations of 0.3 ± 0.1 ppm and 1.7 ± 0.4 ppm (Smedley and Pearce, 2016) were used to determine the internal alpha and beta dose-rates. An a-value of 0.10 ± 0.02 (Balescu and Lamothe, 1993) was used to calculate the alpha dose-rates. Cosmic dose-rates were determined after Prescott and Hutton (1994). Dose-rates were calculated using the Dose Rate and Age Calculator (DRAC; Durcan et al. 2015). Grain size was measured by randomly selecting grains in the rock slices for each sample and calculating ±1 standard deviation around the mean grain size.

| Sample | Grain size (μm) | U (ppm) | Th (ppm) | K (%) | Internal alpha dose-rate (Gy/ka) | Internal beta dose-rate (Gy/ka) | External alpha dose-rate (Gy/ka) | External beta dose-rate (Gy/ka) | External gamma dose-rate (Gy/ka) | External cosmic dose-rate (Gy/ka) | Total dose-rate (Gy/ka) |
|---|---|---|---|---|---|---|---|---|---|---|---|
| BALL02 | 56-91 | 1.02±0.15 | 4.85±0.28 | 1.73±0.29 | 0.14±0.04 | 0.27±0.06 | 0.21±0.05 | 1.62±0.00 | 0.78±0.08 | 0.31±0.03 | 3.32±0.12 |
| BALL03 | 79-117 | 1.02±0.14 | 5.21±0.28 | 1.86±0.29 | 0.16±0.04 | 0.35±0.08 | 0.17±0.04 | 1.71±0.00 | 0.83±0.08 | 0.31±0.03 | 3.52±0.12 |
| ROAD01 | 33-51 | 2.07±0.27 | 7.80±0.42 | 2.45±0.43 | 0.10±0.03 | 0.16±0.03 | 0.61±0.12 | 2.43±0.00 | 1.22±0.11 | 0.30±0.03 | 4.81±0.18 |
| ROAD02 | 67-113 | 1.55±0.18 | 5.67±0.38 | 2.88±0.40 | 0.15±0.04 | 0.32±0.08 | 0.23±0.05 | 2.59±0.00 | 1.16±0.10 | 0.30±0.03 | 4.76±0.15 |
| ROAD03 | 112-225 | 1.93±0.21 | 5.30±0.30 | 1.96±0.31 | 0.18±0.04 | 0.58±0.20 | 0.14±0.04 | 1.85±0.00 | 0.96±0.08 | 0.29±0.03 | 4.00±0.22 |

**Table 2. Calibration factors determined by fitting depth profiles. Note that values presented are medians.**

| Sample | IRSL signal | $\overline{\sigma\varphi_0}$ (s$^{-1}$) | Range ±1 σ (s$^{-1}$) | μ (mm$^{-1}$) | Range ±1 σ (mm$^{-1}$) |
|---|---|---|---|---|---|
| ROAD01 | IR$_{50}$ | $2.80\mathrm{e}^{-4}$ | $8.41\mathrm{e}^{-4} - 6.43\mathrm{e}^{-5}$ | 3.2 | 2.5 – 3.8 |
| | pIRIR$_{150}$ | $3.27\mathrm{e}^{-5}$ | $1.16\mathrm{e}^{-4} - 2.14\mathrm{e}^{-5}$ | 3.1 | 2.2 – 3.7 |
| | pIRIR$_{225}$ | $2.88\mathrm{e}^{-5}$ | $3.99\mathrm{e}^{-5} - 1.51\mathrm{e}^{-5}$ | 3.0 | 2.3 – 3.6 |
| ROAD02 | IR$_{50}$ | $6.67\mathrm{e}^{-6}$ | $1.27\mathrm{e}^{-4} - 3.50\mathrm{e}^{-7}$ | 2.1 | 1.4 – 2.6 |
| | pIRIR$_{150}$ | $1.73\mathrm{e}^{-8}$ | $9.64\mathrm{e}^{-8} - 9.75\mathrm{e}^{-9}$ | 1.5 | 1.1 – 2.3 |
| | pIRIR$_{225}$ | $9.01\mathrm{e}^{-8}$ | $5.53\mathrm{e}^{-7} - 2.31\mathrm{e}^{-8}$ | 2.8 | 1.8 – 3.6 |
| ROAD03 | IR$_{50}$ | $1.56\mathrm{e}^{-5}$ | $1.64\mathrm{e}^{-4} - 1.48\mathrm{e}^{-6}$ | 2.7 | 2.0 – 3.2 |
| | pIRIR$_{150}$ | $3.80\mathrm{e}^{-8}$ | $4.40\mathrm{e}^{-7} - 1.12\mathrm{e}^{-8}$ | 1.5 | 1.1 – 2.5 |
| | pIRIR$_{225}$ | $1.70\mathrm{e}^{-8}$ | $1.17\mathrm{e}^{-7} - 4.70\mathrm{e}^{-9}$ | 1.4 | 0.9 – 2.5 |

Table 3. Luminescence exposure ages and erosion rates determined using the approach of Lehmann et al. (2018) and Lehmann et al. (2019a), respectively. The values of $\overline{\sigma\varphi_0}$ and $\mu$ were determined from known-age sample ROAD02 (57 a).

| Sample | Signal | $\overline{\sigma\varphi_0}$ (s⁻¹) | $\mu$ (mm⁻¹) | $\dot{D}$ (Gy/ka) | $D_0$ (Gy) | Exposure age (a) | Steady-state erosion rate (mm/ka) | Min. initiation time (a) | Max. transient erosion rate (mm/ka) | Initiation time (a) | Min. transient erosion rate (mm/ka) | Initiation time (a) |
|---|---|---|---|---|---|---|---|---|---|---|---|---|
| BALL02 | $IR_{50}$ | $6.67e^{-6}$ | 2.1 | $3.32 \pm 0.12$ | 500 | $8 \pm 2$ | 66 | 73 | - | - | - | - |
| | $pIRIR_{150}$ | $1.73e^{-8}$ | 1.5 | $3.32 \pm 0.12$ | 350 | $66 \pm 16$ | 9 | 593 | - | - | - | - |
| | $pIRIR_{225}$ | $9.01e^{-8}$ | 2.8 | $3.32 \pm 0.12$ | 350 | $263 \pm 30$ | - | - | 310 | 4 | 12 | 90 |
| BALL03 | $IR_{50}$ | $6.67e^{-6}$ | 2.1 | $3.52 \pm 0.12$ | 500 | $387 \pm 103$ | - | - | 460 | 3 | 6 | 231 |
| | $pIRIR_{150}$ | $1.73e^{-8}$ | 1.5 | $3.52 \pm 0.12$ | 350 | $296 \pm 54$ | - | - | 100 | 19 | 14 | 137 |
| | $pIRIR_{225}$ | $9.01e^{-8}$ | 2.8 | $3.52 \pm 0.12$ | 350 | $362 \pm 49$ | - | - | 180 | 4 | 11 | 73 |