# Peer review of "Erosion rates in a wet, temperate climate derived from rock luminescence techniques"

_Geochronology, 2021_

## Author Comment (AC1)

**Responses to compiled reviewer and editor comments for manuscript titled: *Erosion* rates in a wet, temperate climate derived from rock luminescence techniques.**

In this document, we propose changes to the manuscript in response to each of the reviewer and editor comments. We have collated all of the comments into one document as some of the reviewer and editor comments interact with one another, and so it provides more clarity. We use the following abbreviations listed below. Please note that the ACs are our responses to the reviewer and editor comments.

Reviewer 1 (Benjamin Lehmann) comments: RC1

Reviewer 2 (Anonymous) comments: RC2

Editor comments: EC

Author comments: AC

**Reviewer 1: Benjamin Lehmann (RC1)**

**General comments**

**RC1:** This paper presents a study on the erosion rate history of rock avalanche deposit using surface exposure datings from optically stimulated luminescence signals. The authors present luminescence signals of calibration sites and from rock boulder surfaces. The setting of the rock avalanche deposits is framed by terrestrial cosmogenic nuclide (TCN) dating from the literature. The study is well organized and takes advantage of the previous work on this OSL application by using rock color and texture observations in order to choose the most appropriated calibration samples. The main innovation of this study is the use of a multi-elevated temperature, post-infra-red, infra-red stimulated luminescence (MET-pIRIR) protocol (50, 150 and 225°C) allowing the identification of samples complexities and bringing more constrain to calculate exposure datings and erosion rate histories.

Overall the paper is well written, easy to follow but the figures lake in clarity and bring confusion to the reader. Indeed, Figure 7 is supposed to convince of the good quality of the inversion of the erosion history (erosion rate and time at which the erosion is switch on) from the experimental luminescence signals, but in its current form, the figure does not allow any visual inspection and validation of the results of erosion rate history.

Also, the authors use an approach developed by Lehmann et al., 2019a, in which the use of OSL signals from bedrock surface allows to calculate an erosion correction over TCN dating. Here, this approach in not fully exploited. Erosion rate histories are determined but are not used to discuss the possible erosion correction of the TCN exposure dating.

**AC**: This is a very good suggestion that we propose to include in the paper, please see a full explanation in response to your comment on Lines 271-281, which is directly related to this comment.

**RC1:** Finally, this study brings important observations on the differences of bleaching depth according of the energy traps targeted by the OSL stimulation. The  $IR_{50}$  signals are bleached deeper than  $pIR_{150}$  which is bleached deeper than  $pIR_{225}$ , in a way that the higher the temperature of stimulation is, the longer it takes to the light exposure to affect the OSL signal in depth. However, the discussion on the difference of bleaching rate of the different signal could have be brought further. Does the difference in bleaching depth of the different stimulations of a same sample could give us information about complex burial/exposure histories? Do the signals from different stimulations would have the same bleaching difference in a steady state or with a transient state with erosion?

**AC**: Although, these are very interesting ideas to explore in the future and we thank the reviewer for sharing them, we do not currently consider that our data provides any evidence to explore them to any depth within the discussion of this paper as our samples have very simple burial/exposure histories. As such, we propose to include a statement to highlight these future research avenues where we discuss the value of using the MET protocol in Section 6.1 as we consider the ideas to be valuable for future research avenues in light of our findings. See below:

Beyond this, it is possible that the MET-pIRIR protocol may be useful in identifying complex burial or exposure histories of rocks, similar to those that have been reported in previous studies but solely using the IR50 signal (e.g. Freiesleben et al. 2015; Brill et al. 2021). There is also potential to explore whether the different temperature IRSL signals of the MET protocol record different states of erosion (i.e. steady or transient states) within the same rock surface, whereby the pIRIR signals that are attenuated greater are more susceptible to transient states of erosion in comparison to the lower temperature signals, which measure luminescence depth profiles to greater depths within the rock surface.

**Specific comments and technical corrections**

**RC1: Line 53:** The authors should mentioned the work of Brown and Moon, 2019\* and Brown, 2020\*\*.

**AC**: We propose adding citations for these works in Section 1. It includes the following sentences at the appropriate positions within the paragraph:

Brown (2020) even combine these phenomena within model simulations to explore different sample histories of exposure and burial to inform geomorphological interpretations of luminescence depth profiles measured for samples collected from the natural environment.

Recent findings from erosion simulations compared with measured data have shown that the erosion rates derived from luminescence depth profiles can be accurate even with stochastic erosion as experience in nature (Brown and Moon, 2019).

RC1: Line 90: The authors should mentioned the work of Brill et al., 2021\*\*\*.

**AC**: We agree and propose adding the work of Brill et al. (2021) on carbonate limestones into our text:

Studies have applied rock luminescence techniques (mostly exposure dating) to a variety of lithologies including granites, gneisses (Lehmann et al. 2018 2019a,b; Meyer et al. 2019), sandstones (Sohbati et al. 2012; Chapot et al. 2012; Pederson et al. 2014), quartzites (Gliganic et al. 2019) and carbonate limestone (Brill et al. 2021).

**RC1: Lines 208-213:** How do the raw data  $L_x/T_x$  were normalised ( $L_0$  determination)? Is it done for each core individually or for each sample (considering the same  $L_0$  for every core of a same sample)? Core 3 for IR50 signal of sample ROAD3 (Fig. 4D) for example, seems to be normalised too low. I would recommend to normalise each independently. The normalisation approach should also be mentioned the Fig. 4 caption.

**AC**: The raw data  $L_x/T_x$  for each core was normalised individually as you suggest. There is a large amount of scatter in the saturated plateau of the cores (as is typical in rock luminescence depth)

profiles) which makes Core 3 for  $IR_{50}$  signal of sample ROAD03 (Fig. 4D) look low, but it is not inconsistent with the rest of the samples. Based upon your suggestion, we propose adding the following explanation of the normalisation approach in the caption of Fig. 4:

All of the raw  $L_n/T_n$  data presented in this figure were normalised individually for each core and subsequent analysis uses the data in this format.

**RC1: Lines 219-221:** The difference in depth of the bleaching front regarding the difference of ages of sample ROAD02 and ROAD03 could be explain also with the noise of the signal, the orientation of the sampled faces, difference at the rock surface.

**AC**: We propose adding this as an example within the text to illustrate factors that may influence the light penetration in our calibration samples as suggested below:

This suggests that either another factor is influencing light penetration with depth in these rocks (e.g. small differences in the orientation of the sampled rock faces)...

**RC1: Lines 214-221:** How does the fit (black lines) in Fig. 4 are produced? This should be mentioned in the main text and in the caption of Fig. 4.

AC: In the text we propose adding the following sentence in response to this comment:

Note that the inferred models shown in Fig. 4 were fit using the  $\overline{\sigma\varphi_0}$  and  $\mu$  values included in each figure. See Section 5.2 for further explanation of the estimation of the model parameters.

While in the caption of Fig. 4 we propose adding an explanation of how the fit was produced and also provided the  $\overline{\sigma \varphi_0}$  and  $\mu$  values in each figure to provide further clarification (this is in response to RC2 comments):

The black line shown is the inferred model that was fitted to derive the corresponding  $\overline{\sigma \varphi_0}$  and  $\mu$  values included in each figure.

**RC1: Line 229:** "sample BALL01 was coarser grained than BALL02 and BALL03" this affirmation is not supported by the results shown in Fig. 3B where only the two first discs of sample BALL01 are coarser than the other samples BALLs but deeper into the signal BALL03 seems to has the coarsest grained texture.

**AC**: We propose editing the text so that it specifies that the surface is coarser:

... the surface of sample BALL01 was coarser grained than BALL02 and BALL03 (Fig. 2; Fig. 3b).

**RC1: Lines 232-233:** "[...] lost during sample and/or sample preparation [...]". Are there any ways to invalidate this experimental biais? During the sampling, did you mark the exposed surface with ink? Did you measure the core before and after slicing? Does the surface of the first disc look alike the surface of other 1st disc of other cores?

**AC**: Yes, we did all those suggestions during sampling and preparation as we were very careful. Prior to writing the paper, we revisited all information on this sample, visually inspecting all of the discs, cores and core holes in the rocks, and could not find any other explanation. We propose adding further clarification of this into the text, but cannot invalidate the statement definitively so retain the narrative. See below:

Thus, although care was taken when sampling to mark the surface of rock and to measure the rock cores before and after slicing, it is possible that the luminescence depth profile (likely <10 mm based on BALL02 and BALL03) was lost during sampling and/or sample preparation due to the presence of a fragile weathering crust, potentially with a sub-surface zone of weakness (e.g. Robinson and Williams, 1987).

**RC1:** Lines 244-245: " $\overline{\sigma\varphi_0}$  and  $\mu$  were calibrated using the known-age sample [...]" Reading this sentence I was confused that ROADs samples are the calibration samples. "ROAD samples" could be explicitly mentioned in this sentence to improve clarity.

**AC**: We propose adding this into the sentence:

 $\overline{\sigma \varphi_0}$  and  $\mu$  were calibrated using the known-age samples (ROAD01, ROAD02 and ROAD03) of similar, suitable rock composition as determined by the down-core profiles of RGB and grainsize (Section 4.2).

**RC1:** Lines 253-258: There is no mention of the results for the calibration of sigmaphi0 and  $\mu$  parameters. They should be explicitly written in the main text.

**AC**: We had originally omitted this because it turns into a long list in the text that is much more easily read from the table. However, to respond to the reviewers comment here, we propose adding a few sentences listing the parameters in the text.

For ROAD01, the parameters determined using the IR50 ( $\mu$  = 3.2 mm-1,  $\overline{\sigma\varphi_0}$  = 2.80e-4 s-1), pIRIR150 ( $\mu$  = 3.1 mm-1,  $\overline{\sigma\varphi_0}$  = 3.27e-5 s-1) and pIRIR225 ( $\mu$  = 3.0 mm-1,  $\overline{\sigma\varphi_0}$  = 2.88e-5 s-1) signals were broadly consistent. For ROAD02, the parameters differed between the IR50 ( $\mu$  = 2.1 mm-1,  $\overline{\sigma\varphi_0}$  = 6.67e-6 s-1), pIRIR150 ( $\mu$  = 1.5 mm-1,  $\overline{\sigma\varphi_0}$  = 1.73e-8 s-1) and pIRIR225 ( $\mu$  = 2.8 mm-1,  $\overline{\sigma\varphi_0}$  = 9.01e-8 s-1) signals, but the values for each signal were broadly similar to the equivalent values determined for ROAD03 using the IR50 ( $\mu$  = 2.7 mm-1,  $\overline{\sigma\varphi_0}$  = 1.56e-5 s-1), pIRIR150 ( $\mu$  = 1.5 mm-1,  $\overline{\sigma\varphi_0}$  = 3.80e-8 s-1) and pIRIR225 ( $\mu$  = 1.4 mm-1,  $\overline{\sigma\varphi_0}$  = 1.70e-8 s-1) signals.

**RC1:** Lines 258-260: Note that the  $\mu$  value for ROAD03 are not so different than the ones for ROAD02 even if the grain size are very different. Could you comment on that?

**AC**: Yes, we propose adding a comment as suggested:

Given the similarity of  $\overline{\sigma \varphi_0}$  and  $\mu$  determined using all three IRSL signals for ROAD02 and ROAD03 and the difference in grainsizes (Fig. 3B), it suggests that grainsize has a minimal impact upon the attenuation of light into a rock surface in comparison to other factors (e.g. mineralogy, surficial coatings).

**RC1: Lines 262-266:** All the values mentioned in this section are different than the values in Table 3!

AC: We will correct these mistakes in text.

RC1: Line 305: The "(2018)" reference is wrong and it should "(2019a)"

AC: We will correct this in the text.

**RC1: Lines 271-281:** Now that you inferred erosion history is determined for the two boulders, what are the consequences on the cosmo age of the deposit? Does the sampled boulders are the same sampled for TCN dating? If not, it would be interested to discuss the potential exposure age correction. Lehmann et al., 2019a approach does correct the TCN age with the inferred erosion history. It seems that the approach is not fully exploited here.

**AC:** This is a very good suggestion that we propose incorporating into the paper as a paragraph within Section 6.2, which is included below:

The modelled erosion histories that we have calculated here using the luminescence erosion-meter for samples BALL02 and BALL03 would have had a minimal effect upon the TCN exposure age (4.54 ± 0.27 ka; Ballantyne and Stone, 2004). Only the steady-state erosion rate of 66 mm/ka inferred for BALL02 using the IR50 signal, when applied for durations exceeding 1 ka, would have increased the exposure age to any great degree. For example, when the steady-state erosion rate of 66 mm/ka was applied for 0.1 ka, the corrected TCN exposure age would have been 4.58 ka and, when the same erosion rate was applied for 1 ka it would have been 4.99 ka; these corrected ages are consistent within  $\pm 2 \sigma$  uncertainties of the uncorrected age of 4.54  $\pm$  0.27 ka (Ballantvne and Stone. 2004). The higher, transient erosion rates inferred for BALL03 were all applied for such a short period of time (e.g. Table 3) that they had a minimal effect of the TCN exposure age. Based on the erosion rates derived here, at steady state, the boulder sampled for BALL02 would have lost 300 mm (IR50), 41 mm (pIRIR150) and 54 mm (pIRIR225) from the surface in total over 4.54 ka; these values for the pIRIR150 and pIRIR225 signal are broadly consistent with field observations of quartz protrusions on the surface of boulders >2 x 2 x 2 m that were densely distributed within the rock avalanche feature (Fig. 1). The erosion for the IR50 signal is high and inconsistent with these field observations. Alternatively, the maximum erosion rate end member suggested by the pIRIR225 signal would have removed 1407 mm, which is also inconsistent with field evidence and so likely indicative of the transient state of erosion where high erosion rates were only sustained over short periods of time. For the boulder samples for BALL03, steady state erosion would have lost 27 mm (IR50), 64 mm (pIRIR150) and 50 mm (pIRIR225) from the surface over the 4.54 ka, which is also consistent with field observations. Similar to BALL02, the maximum end members of the derived transient erosion rates for the IR50 (2088 mm), pIRIR150 (454 mm) and pIRIR225 (817 mm) signals would have removed more material from the boulder surface over the 4.54 ka than is consistent with field observations; thus, also indicative that these high erosion rates were only sustained over a short period of time.

**RC1: Lines 355-365:** Do field observations of the deposit of weathered material on/in the ground/soil below the blocks have been done and would validate the hypothesis raised in this paragraph?

**AC**: We propose including the sentence below in Section 6.3 to discuss field observations of weathered material and also include a photograph of the wider area in Fig. S5, newly included in the supplementary material to illustrate our observations.

This is also supported by a lack of shattered material surrounding the large sampled boulders (and in fact on much of the Beinn Alligin rock avalanche deposit), despite the presence of dense vegetation surrounding the boulders (e.g. Fig. S5).

**RC1: Figure 1:** This figure could be highly improved by adding in Panel B, the outline of the rock avalanche deposit, the elevation isoline or two elevation points and the coordinates. In Panel C, the north or flow direction of the rock avalanche deposit.

**AC**: We propose adding the outline of the rock avalanche into Fig. 1B in addition to the elevations. We will also add the rock avalanche flow direction into each image of Fig. 1C.

RC1: Figure 2: The scale could be directly placed on the figure.

**AC**: The scale will be added to the figure.

**RC1: Figure 3:** The direction of the y-axis label should be turned 180° to be consistent with other figures.

**AC**: The axes will be turned by 180° so that the plot is consistent with other figures.

**RC1: Figure 4:** The uncertainties of the inversion could be plot as an envelope using  $\pm 1$  sigma of the  $\mu$  and sigmaphi0.

**AC**: Thank you for this comment, it was very enlightening when we produced inferred models for each of the ROAD samples using the  $\pm 1 \sigma \overline{\sigma \varphi_0}$  and  $\mu$  values as suggested here. These lines will be included in Fig. 4 and we will add a sentence into the figure caption to describe how they were produced.

**RC1: Figure 6:** In every sub figure, an age is written in white, for example  $(0.01 \text{ a}^{-1})$  in Fig. 6A. Does unit  $[a^{-1}]$  is the correct unit? Also, the units are mentioned with "[]" but should be "()" for consistency with the rest of the paper.

**AC**: You are absolutely correct, sorry for the mistake. We will correct both the units and the brackets in Fig. 6.

**RC1:** Finally, the inversions for each stimulation of the ROAD01 sample appear to explore a truncated range of  $\mu$  values, that is, the probabilities of 1 (yellow) reach the side of the inspection box. The  $\mu$  values obtained will surely be much higher if the inversion will allow to explore the values of  $\mu$  up to 5 or 6 mm-1.

**AC**: We have not extended the axes of these inversions as ROAD01 had such a short exposure history of only 0.01 a. Thus, there is large uncertainty in these inversions. We use it as a means of demonstrating that even very short durations of exposure can lead to the development of a luminescence depth profile, but do not use them for calibrating any of our samples and so retain the

use of the original parameter space. If we were using them for calibration, we would of course provide a wider parameter space; however, it is unlikely that the data would then have such large uncertainties and so would likely not need this.

**RC1: Figure 7:** The formatting of this figure should be considerably improved. The panels A, B, C, G, H, I do not allow any visual inspection of the data and inversion qualities (for ex: the x-axis boundaries should be set between 0 and 20 mm). The inversions in C, G, H and I do not seem to fit to the experimental value. This figure should be THE figure of the paper, but in the current formatting, it removes persuasive force of the results on the erosion rate history and confuses the conveyed message by the study.

**AC**: To improve the data visualisation, we propose removing the luminescence depth profiles from Fig. 7 and including the inferred erosion models in the luminescence depth profiles of Fig. 5. Fig. 7 will then only include the inversion model profiles of likelihoods. We propose providing additional comments on the fit of the inferred erosion models to the experimental data in "Section 5.3 Apparent exposure ages and erosion rates" (see RC2 general comments below where we address this in detail).

RC1: Figures S1, S4: These figures are too pixelised and should be improved.

**AC**: Both figures have been re-exported, which has improved the resolution.

**RC1: Supplementary material:** Raw data of  $L_x/T_x$  luminescence with depth for every core/sample could be shared in the supplementary material.

**AC**: We will provide the raw data of the luminescence depth profiles in the supplementary material from Table S2-S7. We will also include a note explaining the availability of this data in the supplementary material in the figure captions.

**RC1: Formatting:** In general, there is a lack in consistency between the labelling of figures (i.e. Fig. 4 A, B) in uppercase letters and its mention in the main text (i.e. Fig 4 a, b) in lowercase letters.

**AC**: Apologies, I think some confusion arose around journal formatting regulations! We have corrected all of these formatting issues already.

**RC1:** \*Brown, N. D., & Moon, S. (2019). Revisiting erosion rate estimates from luminescence profiles in exposed bedrock surfaces using stochastic erosion simulations. Earth and Planetary Science Letters, 528, 115842.

\*\*Brown, N.D. (2020) Which geomorphic processes can be informed by luminescence measurements?, Geomorphology, https://doi.org/10.1016/j.geomorph.2020.107296

\*\*\*Brill, D., May, S. M., Mhammdi, N., King, G., Lehmann, B., Burow, C., ... & Brückner, H. (2021). Evaluating optically stimulated luminescence rock surface exposure dating as a novel approach for reconstructing coastal boulder movement on decadal to centennial timescales. Earth Surface Dynamics, 9(2), 205-234.

**AC**: Thank you for the reference suggestions here, we have included them within the text and reference list.

**Reviewer 2: Anonymous (RC2)**

**General remarks**

**RC2:** The paper "Erosion rates in a wet, temperate climate derived from rock luminescence techniques" presents new data for the application of luminescence rock surface techniques. The approach is applied to rock avalanche deposits from Scotland that have previously been dated by terrestrial cosmogenic nuclides to infer regional erosion rates for the last millennia. The study is well structured, well written and generally easy to follow. It presents valuable new results for the emerging topic of luminescence rock surface dating and erosion rate modelling that are highly needed to better understand the limitations and the potential of the technique. Innovative methodological aspects of the study are the use of a MET-post-IR-IRSL protocol to provide internal quality criteria for the selection of samples with appropriate lithology to adequately record light penetration into the rock surface.

There are several inconsistencies in the paper with regard to the presentation of the data. In particular numbers presented in the main text, the tables and figures do not always match (I will provide details below). This must be revised prior to publication.

**AC**: We will corrected all of the inconsistencies and formatting issues within the paper. They will be presented as tracked changes within the document.

**RC2**: Another irritating aspect that needs clarification is the model fit for the unknown age samples in Figure 7. It seems that the inferred model does not really fit the measured data for most of the measured signals. This might indicate inadequate values for  $\mu$  and/or  $\overline{\sigma\varphi_0}$  and the authors should comment on that.

**AC**: This is a sound observation and so we have further explored this. Firstly, we propose moving the inferred erosion model from Fig. 7 to Fig. 5, so that Fig. 7 would only include the inversion profiles. We also propose including further discussion into "Section 5.3 Apparent exposure ages and erosion rates" of the fits in response to the RCs. See below:

At face value, the fit of the inferred erosion model to the experimental data for BALL02 using the  $IR_{50}$ (Fig. 5D) and pIRIR150 (Fig. 5E) signals is better than the equivalent fits for BALL02 using the pIRIR225 signal (Fig. 5F) and BALL03 using the  $IR_{50}$  (Fig. 5G), pIRIR150 (Fig. 5H) and pIRIR225 (Fig. 5I) signals. In the latter cases, the inferred erosion model is shallower than the experimental data. This could suggest that the  $\overline{\sigma \varphi_0}$  and  $\mu$  values were inaccurate, i.e. the attenuation of light with depth into the rock surface is lower in BALL02 and BALL03 than estimated by ROAD02. A possible explanation for this is that the surface of the roadcut sampled by ROAD02 (Fig. S1a) was orientated differently to the Beinn Alligin rock avalanche boulders sampled by BALL02 and BALL03 (Fig. 1), relative to the incoming sunlight (e.g. Gliganic et al. 2019). If the orientation of the known-age roadcut samples was inconsistent with the unknown samples, we would expect these inconsistencies to manifest similarly in all three MET signals for BALL02 and BALL03, which was not observed here. A factor that is common to all the profiles that are less well fit by the inferred erosion model is that they determined transient erosion rates. This suggests that these surfaces experienced complex erosional histories over time whereby the erosion was time-varying. Consequently, it is possible that surficial weathering products may have changed in thickness and composition over time, which in turn could slightly vary the attenuation of light (Meyer et al. 2018; Luo et al. 2018), meaning that the calibration of  $\overline{\sigma \varphi_0}$  and µ from ROAD02 would introduce more uncertainty into the inferred erosion model. It is possible that sample-specific measurements of  $\overline{\sigma \varphi_0}$  and  $\mu$  (e.g. Ou et al. 2018), rather than calibration from known-age samples, could reduce the uncertainty introduced by time-varying light attenuation. However, further investigation is required into the physical mechanisms of time-varying light attenuation in the context of surficial weathering and subsequent erosion, and the impacts upon inferred transient erosion rates.

RC2: Apart from that I have only minor comments.

**AC**: All of the minor comments are address below.

**Specific comments**

RC2: Lines 79-80: Please add a reference for the insights on the exposure history.

AC: We will add the citation for the exposure history from Ballantyne and Stone, (2004).

**RC2:** Lines 85-93: The shape and position of the bleaching front is also influenced by dose accumulation during exposure. Although this term is irrelevant for most samples, the authors should include a short explanation why they think it is not necessary to address dosing in their setting. It is also confusing that dose rates are considered in the methods section, while they are not part of equation (1). This is confusing and needs clarification.

**AC**: We propose adding worked examples into "Section 2 Theoretical background" in response to the ECs comments, which provide clarification on how the dose accumulated over the exposure time is incorporated into the modelling. Please see comments below where we respond to the ECs comments for further information.

**RC2:** Lines 104-107: How do you consider temporal variability of  $\mu$ ? With your approach you rather account for spatial variability of light attenuation between different surfaces and in different depth levels of a surface.

**AC**: It is not possible to determine the temporal variability of  $\mu$  as we cannot monitor this in real-time over thousands of years. As such, none of the existing studies consider the temporal variability of  $\mu$ . Similarly, we cannot measure it here, however, we do propose including some consideration of the possibility that  $\mu$  is time-dependent when exploring the potential explanations for the fit of the inferred erosion models to the experimental data in Section 5.2 in response to a number of comments from RC1 and RC2.

**RC2: Lines 176-177**: Please provide details regarding internal dose rate assessment here. What internal potassium contents were used? How exactly has sample grain size been determined?

**AC**: This information was already provided in the caption for Table 1, however, we propose also including the information in the text in response to the reviewers comment for clarity:

Internal dose-rates were calculated assuming an internal K-content of  $10 \pm 2$  % (Smedley et al. 2012) and internal U and Th concentrations of  $0.3 \pm 0.1$  ppm and  $1.7 \pm 0.4$  ppm (Smedley and Pearce, 2016), in addition to measured average grain sizes for each sample.

RC2: Line 190: This should be "successively" instead of "simultaneously".

**AC**: We will exchange these words in the text.

RC2: Line 196: Please provide the number of cores that were used per sample.

**AC**: We had already included this information in the original manuscript. See below:

De values were determined for the shallowest disc and the deepest disc from one core of each sample to quantify the natural residual dose and saturation limit (L0, Eq. 1), respectively.

RC2: Line 202: "...were in line with previous measurements of IRSL signals"

AC: We will correct this in the text and it will be shown as tracked changes in the manuscript.

**RC2: Lines 219-220**: How exactly was grain size determined with the microscope? Did you use a software or were grains measured manually? For the latter, how many grains were measured per slice?

**AC**: We will include additional information into the text to explain the methods of measuring grainsize. See below:

The average down-core grainsize of each sample was measured under an optical microscope using Infinity Analyze. For each rock slice of an example core per sample, ten randomly-selected grains were measured and the mean and standard deviation grainsize were calculated per core and plotted against the core depths (Fig. 3B).

**RC2:** Line 235: Which values for  $\mu$  and  $\overline{\sigma\varphi_0}$  have been used for the fits in Figure 4? I assume you used the parameters presented in section 5.2? If so, I would suggest to change the order of sections 5.1 and 5.2.

**AC**: In response to RC1 comments, the parameters used to fit the data shown in Fig. 4 will be included in each plot, which will clarify what parameters were used to fit each dataset. We have not changed the order of Sections 5.1 and 5.2 as Section 5.1 presents the data and describe the luminescence depth profiles in their raw form, and then Section 5.2 then fits this raw data to determine the parameters, so it would not make sense to us to switch the two sections around.

**RC2:** Lines 236-237: The numbers given for the depths of the  $IR_{50}$  bleaching front do not agree with the modelled fits in figure 4. Please clarify.

**AC**: This is a fair comment. We originally presented the depth of the first rock slice that measured an  $L_n/T_n$  value that was >50 % of the saturation plateau, which meant that it was much deeper than the model fit, and as highlighted here, misleading. We will rectify this by referring to the position of

the bleaching front according to the fits show in Fig. 4 and correct the text using tracked changes. We will also correct the values presented in Fig. S4a.

**RC2:** Lines 265-268: Here dose rates are considered, although the term for dose accumulation is not part of equation (1). So either it was not equation (1) that was used for fitting, or the dose rate information is not needed. Please clarify.

**AC**: This is a fair comment as it is not clear exactly how the dose-rate information is used in the parameterisation of  $\overline{\sigma \varphi_0}$  and  $\mu$ , and age/erosion rate determination. We will provide more clarity in this section and refer the reader to our new worked example of the inversion model (requested in the Editors Comments - ECs) for further explanation.

**RC2:** Line 282: The values of Sohbati et al. (2012) are for quartz signals, thus for a different wave length spectrum. Since different wave lengths are attenuated differently, I would suggest to compare with other feldspar studies.

AC: A very good point, thank you. We will remove reference to Sohbati et al. (2012).

**RC2:** Lines 282-286: Please provide the numbers for  $\mu$  and  $\overline{\sigma \varphi_0}$  directly in the text.

**AC**: We had originally omitted this because it turns into a long list in the text that is much more easily read from the table. However, to respond to the RC2 comment here, we will add a few sentences listing the parameters in the text.

For ROAD01, the parameters determined using the IR50 ( $\mu$  = 3.2 mm-1,  $\overline{\sigma\varphi_0}$  = 2.80e-4 s-1), pIRIR150 ( $\mu$  = 3.1 mm-1,  $\overline{\sigma\varphi_0}$  = 3.27e-5 s-1) and pIRIR225 ( $\mu$  = 3.0 mm-1,  $\overline{\sigma\varphi_0}$  = 2.88e-5 s-1) signals were broadly consistent. For ROAD02, the parameters differed between the IR50 ( $\mu$  = 2.1 mm-1,  $\overline{\sigma\varphi_0}$  = 6.67e-6 s-1), pIRIR150 ( $\mu$  = 1.5 mm-1,  $\overline{\sigma\varphi_0}$  = 1.73e-8 s-1) and pIRIR225 ( $\mu$  = 2.8 mm-1,  $\overline{\sigma\varphi_0}$  = 9.01e-8 s-1) signals, but the values for each signal were broadly similar to the equivalent values determined for ROAD03 using the IR50 ( $\mu$  = 2.7 mm-1,  $\overline{\sigma\varphi_0}$  = 1.56e-5 s-1), pIRIR150 ( $\mu$  = 1.5 mm-1,  $\overline{\sigma\varphi_0}$  = 3.80e-8 s-1) and pIRIR225 ( $\mu$  = 1.4 mm-1,  $\overline{\sigma\varphi_0}$  = 1.70e-8 s-1) signals.

**RC2: Lines 288-292**: The unit of the ages should be a instead of a-1. Also the numbers do not match those in Table 3.

**AC**: We will correct these mistakes in the text.

RC2: Lines 306-309: Please provide numbers for erosion rates in the text.

**AC**: Similar to above, we had originally omitted this because it turns into a long list in the text that is much more easily read from the table. However, to respond to the RC2 comment here, we propose adding the following to list the parameters in the text.

However, the pIRIR225 signal suggested a transient erosion state, where the luminescence signal could be derived from numerous pairs of erosion rates and initiation times from a maximum erosion

rate of 310 mm/ka over a minimum time interval of 4 a to a minimum erosion rate of 12 mm/ka over time a minimum time interval of 90 a. All three IRSL signals from sample BALL03 consistently suggested a system undergoing a transient response to erosion, which was consistent with the pIRIR225 signal of BALL02 (Fig. 7, Table 3). The IR50 signal for BALL03 derived a maximum erosion rate of 460 mm/ka over a minimum time interval of 3 a and a minimum erosion rate of 6 mm/ka over a minimum time interval of 13 a and a minimum erosion rate of 100 mm/ka over minimum time interval of 19 a and a minimum erosion rate of 14 mm/ka over a minimum time interval of 19 a and a minimum erosion rate of 14 mm/ka over a minimum time interval of 19 a and a minimum erosion rate of 14 mm/ka over a minimum time interval of 19 a and a minimum erosion rate of 14 mm/ka over a minimum time interval of 19 a and a minimum erosion rate of 14 mm/ka over a minimum time interval of 19 a and a minimum erosion rate of 14 mm/ka over a minimum time interval of 19 a and a minimum erosion rate of 14 mm/ka over a minimum time interval of 137 a. The pIRIR225 signal for BALL03 derived a maximum erosion rate of 180 mm/ka over a minimum time interval of 4 a and a minimum erosion rate of 11 mm/ka over a minimum time interval of 73 a.

**RC2:** Line 317: Based on the modelled fits shown in figure 7 (red lines), the fit seems not to match the measured data for most of the signals, which indeed indicates that the parameters might be inaccurate. This should be explored in more detail.

**AC**: This was also identified by RC2 in their more general comments above and so we have provided a more detailed response there of the further discussion that we have included exploring the fit of the inferred erosion models to the experimental data. The same response applies to this comment.

**RC2:** Lines 327-329: This assumption does not really make sense in my opinion. Post-IR225 signals need shorter wave lengths than  $IR_{50}$  and post-IR150 signals to be reset. But the attenuation of shorter wave lengths in rocks tends to be stronger than that of longer wave lengths (cf. Ou et al., 2018).

**AC**: In this statement we are suggesting that it is the Fe-coating that forms as a weathering product on the surface of the rock that may have preferentially attenuated the longer wavelengths that would reset the  $IR_{50}$  and  $pIRIR_{150}$  signals, rather than the referring to the rock itself. An Fe-coating and crystalline rock would have very different compositions. However, we appreciate that this is a speculative statement and to respond to the reviewer's comment here, we propose re-phrasing this sentence to better reflect the evidence we present in this manuscript. See below:

Interestingly, the similarity between BALL02 and BALL03 for the pIRIR225 signal suggests that the presence of an Fe-coating altered the attenuation of the IR50 and pIRIR150 signals to a lesser extent than the pIRIR225 signal, but the reasons for this requires further investigation.

RC2: Line 343: I think it should be "exploited" or similar instead of "inferred"?

**AC**: We have considered this comment at length and currently do not understand how the word exploited would replace inferred in this context. If the authors were exploiting the fact that shorter-term erosion rates derived from luminescence measurements were higher than the long-term averages, then they would needed to have used them for some specific purpose after calculating them. From our understanding, the authors essentially reported that shorter-term erosion rates derived from luminescence measurements were higher than the long-term averages; thus, the word "inferred" is appropriate in this case. We apologise if we have misunderstanding if this response is not suitable.

**RC2:** Figure 1: Please also mark the locations of the road cuts and provide photographs of the road cut sampling sites.

**AC**: To allow the roadcut sample locations to be shown, we will change the scale of Fig. 1A to include only Scotland, and add the locations into the map. We will include photographs of the roadcut sections in the Supplementary material.

**RC2: Figure 3**: Why are the datasets for grain size and RGB different? Were the analysis performed on different slices?

**AC**: The measurements were not performed on exactly the same cores, but example cores for each sample. We will add the text below into the figure caption to clarify this.

Note that the RGB values and grainsize measurements were not derived from exactly the same cores, but example cores for each sample.

**RC2:** Figure 4: I would suggest to mention in the figure caption that core 3 of Road 2 was not considered for fitting. Also, information regarding the inferred model (which  $\mu$ , which sigmaphi, reference) should be provided.

**AC**: We will add the following text into the figure caption for Fig. 4 "Note that core 3 of ROAD02 was not considered for fitting". We will also include the  $\mu$  and  $\overline{\sigma \varphi_0}$  values in each plot as it was an excellent suggestion.

**RC2:** Figure 5: I recommend to add the inferred model fits that are shown in figure 7 also here. In figure 7 it seems that the model fits (red lines) do not match the measured data for most signals, but it is hard to judge since the panels are rather small. Figure 5 would allow for a much better evaluation.

**AC**: We will move the inferred erosion models from Fig. 7 to Fig. 5 to better represent the data and fit. Fig. 7 will then be limited to only the inversion model profiles of erosion rates.

**RC2:** Figure 6: The unit for the ages should be a instead of  $a^{-1}$ .

**AC**: You are absolutely correct, sorry for the mistakes. We will correct this.

**RC2:** Figure 7: It seems that the model (red lines) in C, G, h and I does not really fit the data. Could you please comment why this might be the case? Please also provide more information on the likelihoods (where do these come from) and the forbidden zone (how do I recognize it in the figure).

**AC**: This was also identified by RC2 in their more general comments above and so we have provided a more detailed response there of the further discussion that we have included exploring the fit of the inferred erosion models to the experimental data. The same response applies to this comment. Also, the new worked examples in "Section 2 Theoretical background" also includes further details on the calculations of likelihoods and the forbidden zone. These additions were included in response to the ECs and so we refer you to our response to the ECs for full details.

RC2: Figure S4a: The signals are only identical for sample ROAD 01 (1 a reference sample).

**AC**: Correct, we will add the missing text "for ROAD01 (0.01 a known-age sample)" into the caption to clarify this.

**Associate Editor comments: Jim Feathers**

EC: Associate editor here.

The two reviews of Smedley et al. on erosion rates in NW Scotland have now been posted. I read them over and think they made many good points. The authors should address all their comments, but two in particular need attention. First, both reviewers found Figure 7 difficult to understand. I found the graphs with mostly blue shading to be fully incomprehensible, yet as one of the reviewers pointed out it is the main figure of the paper. Second, the authors rely heavily on Lehmann et al.'s approach for determining erosion rates that are punctuated rather than steady state. The authors need to better explain that approach, so that the reader does not have to consult Lehmann et al. to understand what the authors are doing. I would suggest giving a working example of how this approach works, using one of their samples as the example.

**AC**: We thank the EC for this suggestion and agree that this will add clarity to the approach used here. As such, we propose adding two paragraphs into "Section 2 Theoretical background" that provide worked examples of steady state and transient state erosion based on our data presented in Fig. 7 (revised). See the paragraphs below:

For determining erosion rates for rock surfaces of known exposure age, Sohbati et al. (2018) use a confluent hypergeometric function to provide an analytical solution, but assuming only steady-state erosion. Lehmann et al. (2019a) provide a numerical approach that exploits the differential sensitivities to erosion of the luminescence (shorter-term) and cosmogenic nuclide (longer-term) techniques to erosion to infer erosion histories (steady state and transient over time) for rock surfaces. This approach uses the experimental data from the luminescence depth profiles and the 10Be concentrations for the same sample. Modelling of the luminescence depth profiles accounts for the electron trapping dependent upon the environmental dose-rate and  $D_0$  but does not consider athermal loss of the signal (i.e. anomalous fading) as it has been demonstrated to have a negligible impact upon the luminescence depth profiles (Lehmann et al. 2019a). Modelling of the 10Be concentrations assumes that no inheritance of the cosmogenic nuclides from prior exposure has occurred, and that the 10Be concentrations have been corrected for the sample depth and density, topographical shielding, local production rates, and the sample location (longitude, latitude and elevation). The combined experimental data for the luminescence depth profiles and cosmogenic nuclides are solved simultaneously for two unknowns: the exposure age and the erosion rate. Forward modelling is used to calculate all of the possible combinations of luminescence depth profiles and 10Be concentrations for synthetic erosion and exposure histories, which are then validated using inversion models against the experimental data to determine the combinations with the highest likelihood. A forbidden zone is determined where the range of possible solutions of erosion rates and durations are in excess of those that are feasible for the experimental 10Be concentrations provided for the sample; these solutions are excluded from the parameter ranges used for the inversion model. For example, the forbidden zone identified in the inversion model profile shown in Fig. 7A (formerly Fig. 7D) is restricted to ranges from ca. 104 mm/ka for durations of ca. 100 a to ca.  $10^3$  mm/ka for ca. >3000 a.

The approach of Lehmann et al. (2019a) models synthetic erosion histories in both steady and transient states. Steady state erosion assumes a constant erosion rate throughout the duration of surface exposure. Transient erosion is typical of shorter exposure histories where a steady state of erosion has not yet been reached. Transient state erosion varies with time and is simulated here by assuming that the evolution of erosion in time follows a stepped function of fixed increases in erosion rates from zero for varying durations throughout the exposure history. An illustration of this is provided by Fig. 7A (formerly Fig. 7D) where transient erosion rates of between ca. 104 mm/ka are

inferred for a minimum duration of ca.  $\leq 1$  a, extending up to ca.  $10^3$  mm/ka for durations up to ca. 50 a. Beyond ca. 50 a, a steady state of erosion is reached at a constant erosion rate of ca.  $10^3$  mm/ka, represented by the flattening of the profile with the highest likelihood. Alternatively, a profile indicative of a transient state of erosion where no steady state has been established is illustrated by Fig. 7D (formerly Fig. 7J) where transient erosion rates of between ca.  $10^2$  mm/ka are inferred for a minimum duration of ca.  $\leq 1$  a, extending up to ca.  $10^1$  mm/ka for durations beyond ca. 200 a. This numerical approach (Lehmann et al. 2019a) allows erosion history to be considered as non-constant in time (i.e. transient), in addition to steady-state, and so it is more indicative of the stochastic erosional processes (driven by temperature, precipitation, snow cover, wind) in nature.

**EC**: So I think the paper needs major changes, but I do concur with the reviewers that the paper has merit and deserves publication once the problems are corrected.

A few other minor comments that I have:

**EC**: Lines 153-158 – Were whole rocks or just portions collected in the field? Cores were drilled in the laboratory, but it is not clear how the rocks were collected.

**AC**: We will clarify this in the text in the Methods section. See below:

Portions of the original boulder or bedrock sample were collected in the field in daylight and immediately placed into opaque, black sample bags.

EC: Line 196 – What do you mean by "similar". You just mentioned fairly wide ranges in reduction.

**AC**: We will add further clarification to the text in response to this comment. See below:

This indicates that within our samples the minerals emitting the IRSL signals (i.e. K-feldspar) have similar inherent bleaching rates when exposed to longer durations of time (i.e. > 8 h in the solar simulator).

EC: Line 578. Figure 4 caption. What do you mean by "replicate" core? Replicates of what?

**AC**: We will add further clarification into the figure caption to help explain this.

Presented in age-order are the IRSL-depth profiles for each of the three replicate cores analysed per sample.

**EC**: Lines 229-239 – I have seen surfaces of cores show complete saturation when nearby cores from the same rock did not. I am not sure we understand fully why this should be.

**AC**: Interesting, it is certainly puzzling. We will add further clarification in this section in response to the RC1 comments, and hopefully our discussion here helps to promote future research to address these issues.

---

## Author Comment (AC4)

**Responses to compiled reviewer and editor comments for manuscript titled:** *Erosion rates in a wet, temperate climate derived from rock luminescence techniques*.

In this document, we detail changes to the manuscript in response to each of the reviewer and editor comments. We use the following abbreviations. Please note that the ACs are our responses to the (unedited) reviewer and editor comments.

Reviewer 1 (Benjamin Lehmann) comments: RC1

Reviewer 2 (Anonymous) comments: RC2

Editor comments: EC

Author comments: AC

**Reviewer 1: Benjamin Lehmann (RC1)**

**General comments**

**RC1:** This paper presents a study on the erosion rate history of rock avalanche deposit using surface exposure datings from optically stimulated luminescence signals. The authors present luminescence signals of calibration sites and from rock boulder surfaces. The setting of the rock avalanche deposits is framed by terrestrial cosmogenic nuclide (TCN) dating from the literature. The study is well organized and takes advantage of the previous work on this OSL application by using rock color and texture observations in order to choose the most appropriated calibration samples. The main innovation of this study is the use of a multi-elevated temperature, post-infra-red, infra-red stimulated luminescence (MET-pIRIR) protocol (50, 150 and 225°C) allowing the identification of samples complexities and bringing more constrain to calculate exposure datings and erosion rate histories.

Overall the paper is well written, easy to follow but the figures lake in clarity and bring confusion to the reader. Indeed, Figure 7 is supposed to convince of the good quality of the inversion of the erosion history (erosion rate and time at which the erosion is switch on) from the experimental luminescence signals, but in its current form, the figure does not allow any visual inspection and validation of the results of erosion rate history.

Also, the authors use an approach developed by Lehmann et al., 2019a, in which the use of OSL signals from bedrock surface allows to calculate an erosion correction over TCN dating. Here, this approach in not fully exploited. Erosion rate histories are determined but are not used to discuss the possible erosion correction of the TCN exposure dating.

**AC**: This is a very good suggestion that we have included in the paper, please see a full explanation in response to your comment on Lines 271-281, which is directly related to this comment.

**RC1:** Finally, this study brings important observations on the differences of bleaching depth according of the energy traps targeted by the OSL stimulation. The IR50 signals are bleached deeper than pIR150 which is bleached deeper than pIR225, in a way that the higher the temperature of stimulation is, the longer it takes to the light exposure to affect the OSL signal in depth. However, the discussion on the difference of bleaching rate of the different signal could have be brought further. Does the difference in bleaching depth of the different stimulations of a same sample could give us information about complex burial/exposure histories? Do the signals from different stimulations would have the same bleaching difference in a steady state or with a transient state with erosion?

**AC**: Although, these are very interesting ideas to explore in the future and we thank the reviewer for sharing them, we do not currently consider that our data provides any evidence to explore them to any depth within the discussion of this paper as our samples have very simple burial/exposure histories. As such, we have included a statement to highlight these future research avenues where we discuss the value of using the MET protocol in Section 6.1 as we consider the ideas to be valuable for future research avenues in light of our findings. See below:

Beyond this, it is possible that the MET-pIRIR protocol may be useful in identifying complex burial or exposure histories of rocks, similar to those that have been reported in previous studies but solely using the $IR_{50}$ signal (e.g. Freiesleben et al. 2015; Brill et al. 2021). There is also potential to explore whether the different temperature IRSL signals of the MET protocol record different states of erosion (i.e. steady or transient states) within the same rock surface, whereby the pIRIR signals that are attenuated greater are more susceptible to transient states of erosion in comparison to the lower temperature signals, which measure luminescence depth profiles to greater depths within the rock surface.

**Specific comments and technical corrections**

**RC1: Line 53:** The authors should mentioned the work of Brown and Moon, 2019* and Brown, 2020**.

**AC**: We have added mention of these works in Section 1. It includes the following sentences at the appropriate positions within the paragraph:

Brown (2020) even combine these phenomena within model simulations to explore different sample histories of exposure and burial to inform geomorphological interpretations of luminescence depth profiles measured for samples collected from the natural environment.

Recent findings from erosion simulations compared with measured data have shown that the erosion rates derived from luminescence depth profiles can be accurate even with stochastic erosion as experience in nature (Brown and Moon, 2019).

**RC1: Line 90:** The authors should mentioned the work of Brill et al., 2021***.

**AC**: We agree and have added the work of Brill et al. (2021) on carbonate limestones into our text:

Studies have applied rock luminescence techniques (mostly exposure dating) to a variety of lithologies including granites, gneisses (Lehmann et al. 2018 2019a,b; Meyer et al. 2019), sandstones (Sohbati et al. 2012; Chapot et al. 2012; Pederson et al. 2014), quartzites (Gliganic et al. 2019) and carbonate limestone (Brill et al. 2021).

**RC1: Lines 208-213:** How do the raw data Lx/Tx were normalised (L0 determination)? Is it done for each core individually or for each sample (considering the same L0 for every core of a same sample)? Core 3 for IR50 signal of sample ROAD3 (Fig. 4D) for example, seems to be normalised too low. I would recommend to normalise each independently. The normalisation approach should also be mentioned the Fig. 4 caption.

**AC**: The raw data $L_x/T_x$ for each core was normalised individually as you suggest. There is a large amount of scatter in the saturated plateau of the cores (as is typical in rock luminescence depth profiles) which makes Core 3 for $IR_{50}$ signal of sample ROAD03 (Fig. 4D) look low, but it is not inconsistent with the rest of the samples. Based upon your suggestion, we have added an explanation of the normalisation approach in the caption of Fig. 4:

All of the raw $L_n/T_n$ data presented in this figure were normalised individually for each core and subsequent analysis uses the data in this format.

**RC1: Lines 219-221:** The difference in depth of the bleaching front regarding the difference of ages of sample ROAD02 and ROAD03 could be explain also with the noise of the signal, the orientation of the sampled faces, difference at the rock surface.

**AC**: We have added this as an example within the text to illustrate factors that may influence the light penetration in our calibration samples as suggested:

This suggests that either another factor is influencing light penetration with depth in these rocks (e.g. small differences in the orientation of the sampled rock faces)…

**RC1: Lines 214-221:** How does the fit (black lines) in Fig. 4 are produced? This should be mentionned in the main text and in the caption of Fig. 4.

**AC**: In the text we have added the following sentence in response to this comment:

Note that the inferred models shown in Fig. 4 were fit using the $\overline{\sigma\varphi_0}$ and µ values included in each figure. See Section 5.2 for further explanation of the estimation of the model parameters.

While in the caption of Fig. 4 we have included an explanation of how the fit was produced and also provided the sigmaphi and mu values in each figure to provide further clarification (this is in response to other reviewer comments):

The black line shown is the inferred model that was fitted to derive the corresponding $\overline{\sigma\varphi_0}$ and µ values included in each figure.

**RC1: Line 229:** "sample BALL01 was coarser grained than BALL02 and BALL03" this affirmation is not supported by the results shown in Fig. 3B where only the two first discs of sample BALL01 are coarser than the other samples BALLs but deeper into the signal BALL03 seems to has the coarsest grained texture.

**AC**: We have edited the text so that it specifies that the surface is coarser:

… the surface of sample BALL01 was coarser grained than BALL02 and BALL03 (Fig. 2; Fig. 3b).

**RC1: Lines 232-233:** "[…] lost during sample and/or sample preparation […]". Are there any ways to invalidate this experimental bias? During the sampling, did you mark the exposed surface with ink? Did you measure the core before and after slicing? Does the surface of the first disc look alike the surface of other 1st disc of other cores?

**AC**: Yes, we did all those suggestions during sampling and preparation as we were very careful. Prior to writing the paper, we revisited all information on this sample, visually inspecting all of the discs, cores and core holes in the rocks, and could not find any other explanation. We have added a clarification of this into the text, but cannot invalidate the statement definitively so retain the narrative. See below:

Thus, although care was taken when sampling to mark the surface of rock and to measure the rock cores before and after slicing, it is possible that the luminescence depth profile (likely <10 mm based on BALL02 and BALL03) was lost during sampling and/or sample preparation due to the presence of a fragile weathering crust, potentially with a sub-surface zone of weakness (e.g. Robinson and Williams, 1987).

**RC1: Lines 244-245:** "sigmaphi0 and µ were calibrated using the known-age sample […]" Reading this sentence I was confused that ROADs samples are the calibration samples. "ROAD samples" could be explicitly mentioned in this sentence to improve clarity.

**AC**: We have added this to the sentence:

$\overline{\sigma\varphi_0}$ and µ were calibrated using Eq (1) and the known-age samples (ROAD01, ROAD02 and ROAD03) of similar, suitable rock composition as determined by the down-core profiles of RGB and grainsize (Section 4.2).

**RC1: Lines 253-258:** There is no mention of the results for the calibration of sigmaphi0 and µ parameters. They should be explicitly written in the main text.

**AC**: We had original omitted this because it turns into a long list in the text that is much more easily read from the table. However, to respond to the reviewers comment here, we have added a few sentences listing the parameters in the text:

For ROAD01, the parameters determined using the $IR_{50}$ (µ = 3.2 mm$^{-1}$, $\overline{\sigma\varphi_0}$ = 2.80e$^{-4}$ s$^{-1}$), $pIRIR_{150}$ (µ = 3.1 mm$^{-1}$, $\overline{\sigma\varphi_0}$ = 3.27e$^{-5}$ s$^{-1}$) and $pIRIR_{225}$ (µ = 3.0 mm$^{-1}$, $\overline{\sigma\varphi_0}$ = 2.88e$^{-5}$ s$^{-1}$) signals were broadly consistent. For ROAD02, the parameters differed between the $IR_{50}$ (µ = 2.1 mm$^{-1}$, $\overline{\sigma\varphi_0}$ = 6.67e$^{-6}$ s$^{-1}$), $pIRIR_{150}$ (µ = 1.5 mm$^{-1}$, $\overline{\sigma\varphi_0}$ = 1.73e$^{-8}$ s$^{-1}$) and $pIRIR_{225}$ (µ = 2.8 mm$^{-1}$, $\overline{\sigma\varphi_0}$ = 9.01e$^{-8}$ s$^{-1}$) signals, but the values for each signal were broadly similar to the equivalent values determined for ROAD03 using the $IR_{50}$ (µ = 2.7 mm$^{-1}$, $\overline{\sigma\varphi_0}$ = 1.56e$^{-5}$ s$^{-1}$), $pIRIR_{150}$ (µ = 1.5 mm$^{-1}$, $\overline{\sigma\varphi_0}$ = 3.80e$^{-8}$ s$^{-1}$) and $pIRIR_{225}$ (µ = 1.4 mm$^{-1}$, $\overline{\sigma\varphi_0}$ = 1.70e$^{-8}$ s$^{-1}$) signals.

**RC1: Lines 258-260:** Note that the µ value for ROAD03 are not so different than the ones for ROAD02 even if the grain size are very different. Could you comment on that?

**AC**: Yes, we have added a comment as suggested:

Given the similarity of $\overline{\sigma\varphi_0}$ and µ determined using all three IRSL signals for ROAD02 and ROAD03 and the difference in grainsizes (Fig. 3B), it suggests that grainsize has a minimal impact upon the attenuation of light into a rock surface in comparison to other factors (e.g. mineralogy, surficial coatings).

**RC1: Lines 262-266:** All the values mentioned in this section are different than the values in Table 3!

**AC**: We thank the reviewer for noticing these mistakes and have corrected the text accordingly.

**RC1: Line 305:** The "(2018)" reference is wrong and it should "(2019a)"

**AC**: We have corrected this in text.

**RC1: Lines 271-281:** Now that you inferred erosion history is determined for the two boulders, what are the consequences on the cosmo age of the deposit? Does the sampled boulders are the same sampled for TCN dating? If not, it would be interested to discuss the potential exposure age correction. Lehmann et al., 2019a approach does correct the TCN age with the inferred erosion history. It seems that the approach is not fully exploited here.

**AC:** This is a very good suggestion that we have incorporated into the paper as a paragraph within Section 6.2, which is included below:

The modelled erosion histories that we have calculated here using the luminescence erosion-meter for samples BALL02 and BALL03 would have had a minimal effect upon the cosmogenic nuclide exposure age (4.54 ± 0.27 ka; Ballantyne and Stone, 2004). Only the steady-state erosion rate of 66 mm/ka inferred for BALL02 using the $IR_{50}$ signal, when applied for durations exceeding 1 ka, would have increased the exposure age to any great degree. For example, when the steady-state erosion rate of 66 mm/ka was applied for 0.1 ka, the corrected cosmogenic nuclide exposure age would have been 4.58 ka and, when the same erosion rate was applied for 1 ka it would have been 4.99 ka; these corrected ages are consistent within ± 2 σ uncertainties of the uncorrected age of 4.54 ± 0.27 ka (Ballantyne and Stone, 2004). The higher, transient erosion rates inferred for BALL03 were all applied for such a short period of time (e.g. Table 3) that they had a minimal effect on the cosmogenic nuclide exposure age.

Based on the long-term erosion rates derived here, the boulder sampled for BALL02 would have lost a total of 300 mm ($IR_{50}$), 41 mm ($pIRIR_{150}$) and 54 mm ($pIRIR_{225}$) from the surface over 4.54 ka, while the long-term erosion rates determined for BALL03 suggested that the boulder surface would have lost 27 mm ($IR_{50}$), 64 mm ($pIRIR_{150}$) and 50 mm ($pIRIR_{225}$). All of these values (except for the $IR_{50}$ signal of BALL02) were broadly consistent with field observations of quartz protrusions on the surface of boulders >2 x 2 x 2 m that were densely distributed within the rock avalanche feature (Fig. 1). Alternatively, the maximum (shorter-term) erosion rate end members of the transient erosion histories would have removed 1407 mm (BALL02, $pIRIR_{225}$), 2088 mm (BALL03, $IR_{50}$), 454 mm (BALL03, $pIRIR_{150}$) and 817 mm (BALL03, $pIRIR_{225}$) from the boulder surface over the 4.54 ka. These large values were inconsistent with field evidence and so indicative of the transient state of erosion where high erosion rates were only sustained over short periods of time.

**RC1: Lines 355-365:** Do field observations of the deposit of weathered material on/in the ground/soil below the blocks have been done and would validate the hypothesis raised in this paragraph?

**AC**: We have included the sentence below in Section 6.3 to discuss field observations of weathered material and also included a photograph of the wider area in Fig. S6 newly included in the supplementary material to illustrate our observations:

This is also supported by a lack of shattered material surrounding the large sampled boulders (and in fact on much of the Beinn Alligin rock avalanche deposit), despite the presence of dense vegetation surrounding the boulders (e.g. Fig. S6.

**RC1: Figure 1:** This figure could be highly improved by adding in Panel B, the outline of the rock avalanche deposit, the elevation isoline or two elevation points and the coordinates. In Panel C, the north or flow direction of the rock avalanche deposit.

**AC**: We have added the outline of the rock avalanche into Fig. 1C in addition to the elevations. We have also added the rock avalanche flow direction into each image of Fig. 1D.

**RC1: Figure 2:** The scale could be directly placed on the figure.

**AC**: The scale has now been added to the figure.

**RC1: Figure 3:** The direction of the y-axis label should be turned 180° to be consistent with other figures.

**AC**: The axes have been turned by 180° so that the plot is consistent with other figures.

**RC1: Figure 4:** The uncertainties of the inversion could be plot as an envelope using ±1sigma of the µ and sigmaphi0.

**AC**: Thank you for this comment, it was very enlightening. We have produced inferred models for each of the ROAD samples using the $\pm$ 1 $\sigma$ $\overline{\sigma\varphi_0}$ and µ values as suggested here. These lines are included in Fig. 4 and we have added a sentence into the figure caption to describe how they were produced.

**RC1: Figure 6:** In every sub figure, an age is written in white, for example (0.01 $a^{-1}$) in Fig. 6A. Does unit [$a^{-1}$] is the correct unit? Also, the units are mentioned with "[ ]" but should be "( )" for consistency with the rest of the paper.

**AC**: You are absolutely correct, sorry for the mistake. We have corrected both the units and the brackets in Fig. 6.

**RC1:** Finally, the inversions for each stimulation of the ROAD01 sample appear to explore a truncated range of µ values, that is, the probabilities of 1 (yellow) reach the side of the inspection box. The µ values obtained will surely be much higher if the inversion will allow to explore the values of µ up to 5 or 6 $mm^{-1}$.

**AC**: We have not extended the axes of these inversions as ROAD01 had such a short exposure history of only 0.01 a. Thus, there is large uncertainty in these inversions. We use it as a means of demonstrating that even very short durations of exposure can lead to the development of a luminescence depth profile, but do not use them for calibrating any of our samples and so retain the use of the original parameter space. If we were using them for calibration, we would of course provide a wider parameter space; however, it is unlikely that the data would then have such large uncertainties and so would likely not need this.

**RC1: Figure 7:** The formatting of this figure should be considerably improved. The panels A, B, C, G, H, I do not allow any visual inspection of the data and inversion qualities (for ex: the x-axis boundaries should be set between 0 and 20 mm). The inversions in C, G, H and I do not seem to fit to the experimental value. This figure should be THE figure of the paper, but in the current formatting, it removes persuasive force of the results on the erosion rate history and confuses the conveyed message by the study.

**AC**: To improve the data visualisation, we have removed the luminescence depth profiles from Fig. 7 and included the inferred erosion models in the luminescence depth profiles of Fig. 5. Fig. 7 now only includes the inversion model profiles of likelihoods. We have provided additional comments on the fit of the inferred erosion models to the experimental data in "Section 5.3 Apparent exposure ages and erosion rates".

**RC1: Figures S1, S4:** These figures are too pixelised and should be improved.

**AC**: Both figures have been re-exported, which has improved the resolution.

**RC1: Supplementary material:** Raw data of Lx/Tx luminescence with depth for every core/sample could be shared in the supplementary material.

**AC**: We have provided the raw data of the luminescence depth profiles in the supplementary material from Table S2-S7. We have also included a note explaining the availability of this data in the supplementary material in the figure captions.

**RC1: Formatting:** In general, there is a lack in consistency between the labelling of figures (i.e. Fig. 4 A, B) in uppercase letters and its mention in the main text (i.e. Fig 4 a, b) in lowercase letters.

**AC**: Apologies, I think some confusion arose around journal formatting regulations. We have corrected all of these formatting issues.

**RC1:** *Brown, N. D., & Moon, S. (2019). Revisiting erosion rate estimates from luminescence profiles in exposed bedrock surfaces using stochastic erosion simulations. Earth and Planetary Science Letters, 528, 115842.

**Brown, N.D. (2020) Which geomorphic processes can be informed by luminescence measurements?, Geomorphology, https://doi.org/10.1016/ j.geomorph.2020.107296

***Brill, D., May, S. M., Mhammdi, N., King, G., Lehmann, B., Burow, C., ... & Brückner, H. (2021). Evaluating optically stimulated luminescence rock surface exposure dating as a novel approach for reconstructing coastal boulder movement on decadal to centennial timescales. Earth Surface Dynamics, 9(2), 205-234.

**AC**: Thank you for the reference suggestions here, we have added them into the text and reference list.

**General remarks**

**RC2:** The paper "Erosion rates in a wet, temperate climate derived from rock luminescence techniques" presents new data for the application of luminescence rock surface techniques. The approach is applied to rock avalanche deposits from Scotland that have previously been dated by terrestrial cosmogenic nuclides to infer regional erosion rates for the last millennia. The study is well structured, well written and generally easy to follow. It presents valuable new results for the emerging topic of luminescence rock surface dating and erosion rate modelling that are highly needed to better understand the limitations and the potential of the technique. Innovative methodological aspects of the study are the use of a MET-post-IR-IRSL protocol to provide internal quality criteria for the selection of samples with appropriate lithology to adequately record light penetration into the rock surface.

There are several inconsistencies in the paper with regard to the presentation of the data. In particular numbers presented in the main text, the tables and figures do not always match (I will provide details below). This must be revised prior to publication.

**AC**: We thank the reviewer for bringing this to our attention and have corrected all of the inconsistencies and formatting issues within the paper. Please see the tracked changes document for details.

**RC2:** Another irritating aspect that needs clarification is the model fit for the unknown age samples in Figure 7. It seems that the inferred model does not really fit the measured data for most of the measured signals. This might indicate inadequate values for μ and/or sigmaphi and the authors should comment on that.

**AC**: This is a sound observation and so we have further explored this. Firstly, we have moved the inferred erosion model from Fig. 7 to Fig. 5, so that Fig. 7 now only includes the inversion profiles. We have also included further discussion into "Section 5.3 Apparent exposure ages and erosion rates" of the fits in response to the RCs:

At face value, the fit of the inferred erosion model to the experimental data for BALL02 using the $IR_{50}$ (Fig. 5D) and $pIRIR_{150}$ (Fig. 5E) signals is better than the equivalent fits for BALL02 using the $pIRIR_{225}$ signal (Fig. 5F) and BALL03 using the $IR_{50}$ (Fig. 5G), $pIRIR_{150}$ (Fig. 5H) and $pIRIR_{225}$ (Fig. 5I) signals. In the latter cases, the inferred erosion model is shallower than the experimental data. This could suggest that the $\overline{\sigma\varphi_0}$ and μ values were inaccurate, i.e. the attenuation of light with depth into the rock surface is lower in BALL02 and BALL03 than estimated by ROAD02. A possible explanation for this is that the surface of the roadcut sampled by ROAD02 (Fig. S1a) was orientated differently to the Beinn Alligin rock avalanche boulders sampled by BALL02 and BALL03 (Fig. 1), relative to the incoming sunlight (e.g. Gliganic et al. 2019). If the orientation of the known-age roadcut samples was inconsistent with the unknown samples, we would expect these inconsistencies to manifest similarly in all three MET signals for BALL02 and BALL03, which was not observed here. A factor that is common to all the profiles that are less well fit by the inferred erosion model is that they determined transient erosion rates. This suggests that these surfaces experienced complex erosional histories over time whereby the erosion was time-varying. Consequently, it is possible that surficial weathering products may have changed in thickness and composition over time, which in turn could slightly vary the attenuation of light (Meyer et al. 2018; Luo et al. 2018), meaning that the calibration of $\overline{\sigma\varphi_0}$ and μ from ROAD02 would introduce more uncertainty into the inferred erosion model. It is possible that sample-specific measurements of $\overline{\sigma\varphi_0}$ and μ (e.g. Ou et al. 2018), rather than calibration from known-age samples, could reduce the uncertainty introduced by time-varying light attenuation. However, further investigation is required into the physical mechanisms of time-varying light attenuation in the context of surficial weathering and subsequent erosion, and the impacts upon inferred transient erosion rates.

**Specific comments**

**RC2: Lines 79-80**: Please add a reference for the insights on the exposure history.

**AC**: We have added the citation for the exposure history from Ballantyne and Stone (2004).

**RC2:** Lines 85-93: The shape and position of the bleaching front is also influenced by dose accumulation during exposure. Although this term is irrelevant for most samples, the authors should include a short explanation why they think it is not necessary to address dosing in their setting. It is also confusing that dose rates are considered in the methods section, while they are not part of equation (1). This is confusing and needs clarification.

**AC**: We have now added worked examples into "Section 2 Theoretical background" in response to the ECs comments, which provides improved clarification on how the dose accumulated over the exposure time is incorporated into the modelling. Please see comments below where we respond to the ECs comments for further information.

**RC2: Lines 104-107**: How do you consider temporal variability of μ? With your approach you rather account for spatial variability of light attenuation between different surfaces and in different depth levels of a surface.

**AC**: It is not possible to determine the temporal variability of μ as we cannot monitor this in real-time over thousands of years. As such, none of the existing studies consider the temporal variability of μ. Similarly, we cannot measure it here. However, we do include some consideration of the possibility that μ is time-dependent when exploring the potential explanations for the fit of the inferred erosion models to the experimental data in Section 5.2 in response to a number of comments from RC1 and RC2.

**RC2: Lines 176-177**: Please provide details regarding internal dose rate assessment here. What internal potassium contents were used? How exactly has sample grain size been determined?

**AC**: This information was provided in the caption for Table 1, however, we have also included the information in the text in response to the reviewers comment:

Internal dose-rates were calculated assuming an internal K-content of 10 ± 2 % (Smedley et al. 2012) and internal U and Th concentrations of 0.3 ± 0.1 ppm and 1.7 ± 0.4 ppm (Smedley and Pearce, 2016), in addition to measured average grain sizes for each sample.

**RC2: Line 190**: This should be "successively" instead of "simultaneously".

**AC**: We have exchanged these words in the text where suggested and in Section 4.1 where it also applies.

**RC2: Line 196**: Please provide the number of cores that were used per sample.

**AC**: We had already included this information in the original manuscript. See below:

De values were determined for the shallowest disc and the deepest disc from one core of each sample to quantify the natural residual dose and saturation limit (L0, Eq. 1), respectively.

**RC2: Line 202**: "…were in line with previous measurements of IRSL signals"

**AC**: We have corrected this in the text. See the tracked changes document.

**RC2: Lines 219-220**: How exactly was grain size determined with the microscope? Did you use a software or were grains measured manually? For the latter, how many grains were measured per slice?

**AC**: We have added additional information into the text to explain the methods of measuring grainsize. See below:

The average down-core grainsize of each sample was measured under an optical microscope using Infinity Analyze. For each rock slice of an example core per sample, ten randomly-selected grains were measured and the mean and standard deviation grainsize were calculated per core and plotted against the core depths (Fig. 3B).

**RC2: Line 235**: Which values for μ and sigmaphi have been used for the fits in Figure 4? I assume you used the parameters presented in section 5.2? If so, I would suggest to change the order of sections 5.1 and 5.2.

**AC**: In response to other reviewer's comments, the parameters used to fit the data shown in Fig. 4 are now included in each plot, which will clarify what parameters were used to fit each dataset. We have not changed the order of Sections 5.1 and 5.2 as Section 5.1 presents the data and describe the luminescence depth profiles in their raw form, and then Section 5.2 then fits this raw data to determine the parameters, so it would not make sense to us to switch the two sections around.

**RC2: Lines 236-237**: The numbers given for the depths of the IR50 bleaching front do not agree with the modelled fits in figure 4. Please clarify.

**AC**: This is a fair comment and we thank the reviewer for highlighting the point. We originally presented the depth of the first rock slice that measured an $L_n/T_n$ value that was >50 % of the saturation plateau, which meant that it was much deeper than the model fit, and as highlighted here, misleading. We have rectified this by referring to the position of the bleaching front according to the fits show in Fig. 4 and corrected the text (see tracked changes document). We have also corrected the values presented in Fig. S4a.

**RC2: Lines 265-268**: Here dose rates are considered, although the term for dose accumulation is not part of equation (1). So either it was not equation (1) that was used for fitting, or the dose rate information is not needed. Please clarify.

**AC**: This is a fair comment as it was not clear exactly how the dose-rate information is used in the parameterisation of $\overline{\sigma\varphi_0}$ and $\mu$, and age/erosion rate determination. We have provided more clarity in this section and refer the reader to our new worked example of the inversion model (requested in the Editors Comments - ECs) for further explanation, but we summarise relevant amendments here:

$\overline{\sigma\varphi_0}$ and $\mu$ were calibrated using Eq. 1 and the known-age samples (ROAD01, ROAD02 and ROAD03) of similar, suitable rock composition as determined by the down-core profiles of RGB and grainsize (Section 4.2). Note that ($\dot{D}$) is not considered in Eq. 1 but is used to determine an apparent exposure or erosion rate and so needs to be measured for each sample.

**RC2: Line 282**: The values of Sohbati et al. (2012) are for quartz signals, thus for a different wave length spectrum. Since different wave lengths are attenuated differently, I would suggest to compare with other feldspar studies.

**AC**: A very good point, thank you. We have removed reference to Sohbati et al. (2012).

**RC2: Lines 282-286**: Please provide the numbers for $\mu$ and sigmaphi directly in the text.

**AC**: We had originally omitted this because it turns into a long list in the text that is much more easily read from the table. However, to respond to the reviewers comment here, we have added a few sentences listing the parameters in the text:

For ROAD01, the parameters determined using the $IR_{50}$ ($\mu = 3.2$ mm$^{-1}$, $\overline{\sigma\varphi_0} = 2.80e^{-4}$ s$^{-1}$), $pIRIR_{150}$ ($\mu = 3.1$ mm$^{-1}$, $\overline{\sigma\varphi_0} = 3.27e^{-5}$ s$^{-1}$) and $pIRIR_{225}$ ($\mu = 3.0$ mm$^{-1}$, $\overline{\sigma\varphi_0} = 2.88e^{-5}$ s$^{-1}$) signals were broadly consistent. For ROAD02, the parameters differed between the $IR_{50}$ ($\mu = 2.1$ mm$^{-1}$, $\overline{\sigma\varphi_0} = 6.67e^{-6}$ s$^{-1}$), $pIRIR_{150}$ ($\mu = 1.5$ mm$^{-1}$, $\overline{\sigma\varphi_0} = 1.73e^{-8}$ s$^{-1}$) and $pIRIR_{225}$ ($\mu = 2.8$ mm$^{-1}$, $\overline{\sigma\varphi_0} = 9.01e^{-8}$ s$^{-1}$) signals, but the values for each signal were broadly similar to the equivalent values determined for ROAD03 using the $IR_{50}$ ($\mu = 2.7$ mm$^{-1}$, $\overline{\sigma\varphi_0} = 1.56e^{-5}$ s$^{-1}$), $pIRIR_{150}$ ($\mu = 1.5$ mm$^{-1}$, $\overline{\sigma\varphi_0} = 3.80e^{-8}$ s$^{-1}$) and $pIRIR_{225}$ ($\mu = 1.4$ mm$^{-1}$, $\overline{\sigma\varphi_0} = 1.70e^{-8}$ s$^{-1}$) signals.

**RC2: Lines 288-292**: The unit of the ages should be a instead of a$^{-1}$. Also the numbers do not match those in table 3.

**AC**: We have corrected these mistakes in the text.

**RC2: Lines 306-309**: Please provide numbers for erosion rates in the text.

**AC**: Similar to above, we had originally omitted this because it turns into a long list in the text that is much more easily read from the table. However, to respond to the reviewers comment here, we have added the following text listing the parameters in the text:

However, the $pIRIR_{225}$ signal suggested a transient erosion state, where the luminescence signal could be derived from numerous pairs of erosion rates and initiation times from a maximum erosion rate of 310 mm/ka over a minimum time interval of 4 a to a minimum erosion rate of 12 mm/ka over time a minimum time interval of 90 a. All three IRSL signals from sample BALL03 consistently suggested a system undergoing a transient response to erosion, which was consistent with the $pIRIR_{225}$ signal of BALL02 (Fig. 7, Table 3). The $IR_{50}$ signal for BALL03 derived a maximum erosion rate of 460 mm/ka over a minimum time interval of 3 a and a minimum erosion rate of 6 mm/ka over a minimum time interval of 231 a. The $pIRIR_{150}$ signal for BALL03 derived a maximum erosion rate of 100 mm/ka over minimum time interval of 19 a and a minimum erosion rate of 14 mm/ka over a minimum time interval of 137 a. The $pIRIR_{225}$ signal for BALL03 derived a maximum erosion rate of 180 mm/ka over a minimum time interval of 4 a and a minimum erosion rate of 11 mm/ka over a minimum time interval of 73 a.

**RC2:** Line 317: Based on the modelled fits shown in figure 7 (red lines), the fit seems not to match the measured data for most of the signals, which indeed indicates that the parameters might be inaccurate. This should be explored in more detail.

**AC**: This was also identified by RC2 in their more general comments above and so we have provided a more detailed response there of the further discussion that we have included exploring the fit of the inferred erosion models to the experimental data. The same response applies to this comment.

**RC2: Lines 327-329**: This assumption does not really make sense in my opinion. Post-$IR_{225}$ signals need shorter wave lengths than $IR_{50}$ and post-$IR_{150}$ signals to be reset. But the attenuation of shorter wave lengths in rocks tends to be stronger than that of longer wave lengths (cf. Ou et al., 2018).

**AC**: In this statement we are suggesting that it is the Fe-coating that forms as a weathering product on the surface of the rock that may have preferentially attenuated the longer wavelengths that would reset the $IR_{50}$ and $pIRIR_{150}$ signals, rather than the referring to the rock itself. An Fe-coating and crystalline rock would have very different compositions. However, we appreciate that this is a speculative statement and to respond to the reviewer's comment here, we have re-phrased this sentence to better reflect the evidence we present. See below:

Interestingly, the similarity between BALL02 and BALL03 for the $pIRIR_{225}$ signal suggests that the presence of an Fe-coating altered the attenuation of the $IR_{50}$ and $pIRIR_{150}$ signals to a lesser extent than the $pIRIR_{225}$ signal, but the reasons for this requires further investigation.

**RC2: Line 343**: I think it should be "exploited" or similar instead of "inferred"?

**AC**: We have considered this comment at length and currently do not understand how the word exploited would replace inferred. If the authors were exploiting the fact that shorter-term erosion rates derived from luminescence measurements were higher than the long-term averages, then they would needed to have used them for some specific purpose after calculating them. From our understanding, the authors essentially reported that shorter-term erosion rates derived from luminescence measurements were higher than the long-term averages; thus, the word "inferred" is appropriate in this case. We would welcome further clarification if required.

**RC2:** Figure 1: Please also mark the locations of the road cuts and provide photographs of the road cut sampling sites.

AC: To allow the roadcut sample locations to be shown, we have added an extra panel into Fig. 1 (Fig. 1B) to include the sample locations in NW Scotland. We have included photographs of the roadcut sections in the Supplementary material.

**RC2: Figure 3**: Why are the datasets for grain size and RGB different? Were the analysis performed on different slices?

**AC**: The measurements were not performed on exactly the same cores, but example cores for each sample. We have added the text below into the figure caption to clarify this:

Note that the RGB values and grainsize measurements were not derived from exactly the same cores, but example cores for each sample.

**RC2:** Figure 4: I would suggest to mention in the figure caption that core 3 of Road 2 was not considered for fitting. Also, information regarding the inferred model (which μ, which sigmaphi, reference) should be provided.

**AC**: We have added the following text into the figure caption for Fig. 4 "Note that core 3 of ROAD02 was not considered for fitting". We have also included the μ and sigmaphi values in each plot as it was an excellent suggestion.

**RC2:** Figure 5: I recommend to add the inferred model fits that are shown in figure 7 also here. In figure 7 it seems that the model fits (red lines) do not match the measured data for most signals, but it is hard to judge since the panels are rather small. Figure 5 would allow for a much better evaluation.

**AC**: We have moved the inferred erosion models from Fig. 7 to Fig. 5 to better represent the data and fit. Fig. 7 is now limited to only the inversion model profiles of erosion rates.

**RC2:** Figure 6: The unit for the ages should be a instead of a$^{-1}$.

**AC**: You are absolutely correct, sorry for the mistakes. We have corrected this.

**RC2:** Figure 7: It seems that the model (red lines) in C, G, h and I does not really fit the data. Could you please comment why this might be the case? Please also provide more information on the likelihoods (where do these come from) and the forbidden zone (how do I recognize it in the figure).

**AC**: This was also identified by RC2 in their more general comments above and so we have provided a more detailed response there of the further discussion that we have included exploring the fit of the inferred erosion models to the experimental data. The same response applies to this comment. Also, the new worked examples in "Section 2 Theoretical background" also includes further details on the calculations of likelihoods and the forbidden zone. These additions were included in response to the ECs and so we refer you to our response to the ECs for full details.

**RC2:** Figure S4a: The signals are only identical for sample ROAD 01 (1 a reference sample).

**AC**: Correct, we have added the missing text "for ROAD01 (0.01 a known-age sample)" into the caption to clarify this.
* * *
**Associate Editor comments: Jim Feathers**

**EC**: Associate editor here.

The two reviews of Smedley et al. on erosion rates in NW Scotland have now been posted. I read them over and think they made many good points. The authors should address all their comments, but two in particular need attention. First, both reviewers found Figure 7 difficult to understand. I found the graphs with mostly blue shading to be fully incomprehensible, yet as one of the reviewers pointed out it is the main figure of the paper. Second, the authors rely heavily on Lehmann et al.'s approach for determining erosion rates that are punctuated rather than steady state. The authors need to better explain that approach, so that the reader does not have to consult Lehmann et al. to understand what the authors are doing. I would suggest giving a working example of how this approach works, using one of their samples as the example.

**AC**: We thank the EC for this suggestion and agree that this will add clarity to the approach used here. As such, we have added two paragraphs into "Section 2 Theoretical background" that provide worked examples of steady state and transient state erosion based on our data presented in Fig. 7 (revised). See the paragraphs below:

For determining erosion rates for rock surfaces of known exposure age, Sohbati et al. (2018) use a confluent hypergeometric function to provide an analytical solution, but assuming only steady-state erosion. Lehmann et al. (2019a) provide a numerical approach that exploits the differential sensitivities to erosion of the luminescence (shorter-term) and cosmogenic nuclide (longer-term) techniques to erosion to infer erosion histories (steady state and transient over time) for rock surfaces. This approach uses the experimental data from the luminescence depth profiles and the $^{10}$Be concentrations for the same sample. Modelling of the luminescence depth profiles accounts for the electron trapping dependent upon the environmental dose-rate and $D_0$ but does not consider athermal loss of the signal (i.e. anomalous fading) as it has been demonstrated to have a negligible impact upon the luminescence depth profiles (Lehmann et al. 2019a). Modelling of the $^{10}$Be concentrations assumes that no inheritance of the cosmogenic nuclides from prior exposure has occurred, and that the $^{10}$Be concentrations have been corrected for the sample depth and density, topographical shielding, local production rates, and the sample location (longitude, latitude and elevation). The combined experimental data for the luminescence depth profiles and cosmogenic nuclides are solved simultaneously for two unknowns: the exposure age and the erosion rate. Forward modelling is used to calculate all of the possible combinations of luminescence depth profiles and $^{10}$Be concentrations for synthetic erosion and exposure histories, which are then validated using inversion models against the experimental data to determine the combinations with the highest likelihood. A forbidden zone is determined where the range of possible solutions of erosion rates and durations are in excess of those that are feasible for the experimental $^{10}$Be concentrations provided for the sample; these solutions are excluded from the parameter ranges used for the inversion model. For example, the forbidden zone identified in the inversion model profile shown in Fig. 7A is restricted to ranges from ca. $10^4$ mm/ka for durations of ca. 100 a to ca. $10^3$ mm/ka for ca. >3000 a.

The approach of Lehmann et al. (2019a) models synthetic erosion histories in both steady and transient states. Steady state erosion assumes a constant erosion rate throughout the duration of surface exposure. Transient erosion is typical of shorter exposure histories where a steady state of erosion has not yet been reached. Transient erosion varies with time and was simulated here by assuming that the evolution of erosion in time follows a stepped function of fixed increases in erosion rates from zero for varying durations throughout the exposure history for each synthetic erosion history simulated. An illustration of this is provided by Fig. 7A where transient erosion rates of between ca. $10^4$ mm/ka were inferred for a minimum duration of ca. ≤1 a, extending up to ca. $10^3$ mm/ka for durations up to ca. 50 a. Beyond ca. 50 a, a steady state of erosion was reached at a constant erosion rate of ca. $10^3$ mm/ka, represented by the flattening of the profile with the highest likelihood. Alternatively, a profile indicative of a transient state of erosion where no steady state has been established is illustrated by Fig. 7D where transient erosion rates of between ca. $10^2$ mm/ka were inferred for a minimum duration of ca. ≤1 a, extending up to ca. $10^1$ mm/ka for durations beyond ca. 200 a. This numerical approach (Lehmann et al. 2019a) allows erosion history to be considered as non-constant in time (i.e. transient), in addition to steady-state, and so it is more indicative of the stochastic erosional processes (driven by temperature, precipitation, snow cover, wind) in nature.

**EC**: So I think the paper needs major changes, but I do concur with the reviewers that the paper has merit and deserves publication once the problems are corrected.

A few other minor comments that I have:

**EC**: **Lines 153-158** – Were whole rocks or just portions collected in the field?  Cores were drilled in the laboratory, but it is not clear how the rocks were collected.

**AC**: We have clarified this in the text in the Methods section. See below:

Portions of the original boulder or bedrock sample were collected in the field in daylight and immediately placed into opaque, black sample bags.

**EC**: **Line 196** – What do you mean by "similar".  You just mentioned fairly wide ranges in reduction.

**AC**: We have added further clarification to the text in response to this comment. See below:

This indicates that within our samples the minerals emitting the IRSL signals (i.e. K-feldspar) have similar inherent bleaching rates when exposed to longer durations of time (i.e. > 8 h in the solar simulator).

**EC**: **Line 578**. Figure 4 caption.  What do you mean by "replicate" core?   Replicates of what?

**AC**: We have added further clarification into the figure caption to help explain this.

Presented in age-order are the IRSL-depth profiles for each of the three replicate cores analysed per sample.

**EC**: **Lines 229-239** – I have seen surfaces of cores show complete saturation when nearby cores from the same rock did not.   I am not sure we understand fully why this should be.

**AC**: Interesting, it is certainly puzzling. We have added further clarification in this section in response to Benjamin Lehmann's comments, and hopefully our discussion here helps to fuel future research to address these issues.

[revised manuscript text omitted]

Fig. S5. (a) Depth of bleaching fronts for the known-age samples ROAD01 (0.01 a), ROAD02 (57 a) and ROAD03 (44 a). Note that the depth of the pIRIR$_{150}$ and pIRIR$_{225}$ bleaching fronts were identical for ROAD01 (0.01 a known-age sample). (b) Residual D$_e$ values determined for the surface slice (0-1 mm depth) of each sample. Note that sample BALL01 is not plotted on this figure as the residual D$_e$ values were large for all the IRSL signals: IR$_{50}$ (477.5 ± 20.7 Gy), pIRIR$_{150}$ (574.6 ± 36.5 Gy) and pIRIR$_{225}$ (could not be interpolated on to the dose-response curve).

[Figure]

Fig. S6. Photograph of the Beinn Alligin rock avalanche to illustrate the nature of the area surrounding the sampled boulders, which includes dense vegetation, the small-scale topography of the boulders and little evidence of shattered material.

Table S1. Multi-elevated temperature post- IR IRSL sequence used for analysis.

| Step | Procedure |
|------|-----------|
| 1 | Natural or regenerative dose |
| 2 | Preheat 250 °C for 100 s |
| 3 | IR LEDs 50 °C for 200 s |
| 4 | IR LEDs 150 °C for 200 s |
| 5 | IR LEDs 225 °C for 200 s |
| 6 | Test-dose 53 Gy |
| 7 | Preheat 250 °C for 100 s |
| 8 | IR LEDs 50 °C for 200 s |
| 9 | IR LEDs 150 °C for 200 s |
| 10 | IR LEDs 225 °C for 200 s |
| 11 | IR LEDs 290 °C for 200 s |

Table S2. Raw $L_n/T_n$ data used for fitting sample ROAD01 in Fig. 4A-C. Note that two rather than three cores were analysed for this sample.

| Depth (mm) | Core 1 | | | | | | Core 2 | | | | | |
|---|---|---|---|---|---|---|---|---|---|---|---|---|
| | $IR_{50}$ | | $pIRIR_{150}$ | | $pIRIR_{225}$ | | $IR_{50}$ | | $pIRIR_{150}$ | | $pIRIR_{225}$ | |
| | $L_n/T_n$ | Error | $L_n/T_n$ | Error | $L_n/T_n$ | Error | $L_n/T_n$ | Error | $L_n/T_n$ | Error | $L_n/T_n$ | Error |
| 0.5 | 0.01 | 0.00 | 0.06 | 0.00 | 0.11 | 0.00 | 0.05 | 0.00 | 0.16 | 0.01 | 0.25 | 0.01 |
| 1.5 | 0.20 | 0.01 | 0.68 | 0.02 | 0.79 | 0.03 | 0.64 | 0.02 | 0.98 | 0.03 | 1.02 | 0.03 |
| 2.5 | 1.02 | 0.03 | 1.09 | 0.03 | 1.04 | 0.03 | 0.98 | 0.03 | 0.99 | 0.03 | 0.99 | 0.03 |
| 3.5 | 1.02 | 0.03 | 1.03 | 0.03 | 1.05 | 0.03 | 1.15 | 0.04 | 1.01 | 0.03 | 0.95 | 0.03 |
| 4.5 | 0.97 | 0.03 | 1.01 | 0.03 | 1.02 | 0.03 | 0.94 | 0.03 | 0.93 | 0.03 | 0.97 | 0.03 |
| 5.5 | 1.01 | 0.03 | 1.01 | 0.03 | 1.04 | 0.03 | 1.04 | 0.03 | 1.03 | 0.03 | 1.03 | 0.03 |
| 6.5 | 1.02 | 0.03 | 1.06 | 0.03 | 1.05 | 0.03 | 1.03 | 0.03 | 0.99 | 0.03 | 0.98 | 0.03 |
| 7.5 | 1.00 | 0.03 | 0.98 | 0.03 | 0.98 | 0.03 | 1.04 | 0.03 | 1.02 | 0.03 | 1.03 | 0.03 |
| 8.5 | 1.03 | 0.03 | 0.99 | 0.03 | 0.98 | 0.03 | 1.06 | 0.03 | 0.98 | 0.03 | 1.00 | 0.03 |
| 9.5 | 1.09 | 0.03 | 1.04 | 0.03 | 1.03 | 0.03 | 1.00 | 0.03 | 1.00 | 0.03 | 1.00 | 0.03 |
| 10.5 | 1.10 | 0.03 | 1.07 | 0.03 | 1.04 | 0.03 | | | | | | |
| 11.5 | 1.00 | 0.03 | 1.00 | 0.03 | 1.00 | 0.03 | | | | | | |
| 12.5 | | | | | | | | | | | | |
| 13.5 | | | | | | | | | | | | |
| 14.5 | | | | | | | | | | | | |
| 15.5 | | | | | | | | | | | | |
| 16.5 | | | | | | | | | | | | |
| 17.5 | | | | | | | | | | | | |
| 18.5 | | | | | | | | | | | | |
| 19.5 | | | | | | | | | | | | |

Table S3. Raw $L_n/T_n$ data used for fitting sample ROAD02 in Fig. 4G-I.

| Depth (mm) | Core 1 | | | | | | Core 2 | | | | | | Core 3 | | | | | |
| --- | --- | --- | --- | --- | --- | --- | --- | --- | --- | --- | --- | --- | --- | --- | --- | --- | --- | --- |
| | $IR_{50}$ | | $pIRIR_{150}$ | | $pIRIR_{225}$ | | $IR_{50}$ | | $pIRIR_{150}$ | | $pIRIR_{225}$ | | $IR_{50}$ | | $pIRIR_{150}$ | | $pIRIR_{225}$ | |
| | $L_n/T_n$ | Error | $L_n/T_n$ | Error | $L_n/T_n$ | Error | $L_n/T_n$ | Error | $L_n/T_n$ | Error | $L_n/T_n$ | Error | $L_n/T_n$ | Error | $L_n/T_n$ | Error | $L_n/T_n$ | Error |
| 0.5 | 0.00 | 0.00 | 0.01 | 0.00 | 0.02 | 0.00 | | | | | | | 0.02 | 0.00 | 0.37 | 0.01 | 0.64 | 0.02 |
| 1.5 | 0.00 | 0.00 | 0.02 | 0.00 | 0.13 | 0.00 | | | | | | | 0.07 | 0.00 | 1.02 | 0.03 | 1.12 | 0.04 |
| 2.5 | 0.01 | 0.00 | 0.29 | 0.01 | 0.61 | 0.02 | 0.02 | 0.00 | 0.75 | 0.02 | 1.01 | 0.03 | 0.55 | 0.02 | 1.01 | 0.03 | 1.03 | 0.03 |
| 3.5 | 0.03 | 0.00 | 0.76 | 0.02 | 0.89 | 0.03 | | | | | | | 0.90 | 0.03 | 1.08 | 0.03 | 1.06 | 0.03 |
| 4.5 | 0.15 | 0.00 | 0.96 | 0.03 | 1.03 | 0.03 | 0.43 | 0.01 | 0.92 | 0.03 | 1.07 | 0.03 | 0.96 | 0.03 | 1.09 | 0.03 | 1.13 | 0.04 |
| 5.5 | 0.47 | 0.01 | 0.93 | 0.03 | 0.99 | 0.03 | 0.85 | 0.03 | 0.92 | 0.03 | 1.08 | 0.03 | 1.00 | 0.03 | 1.00 | 0.03 | 1.00 | 0.03 |
| 6.5 | 1.06 | 0.03 | 0.93 | 0.03 | 1.00 | 0.03 | 0.98 | 0.03 | 0.96 | 0.03 | 1.12 | 0.03 | | | | | | |
| 7.5 | 0.96 | 0.03 | 1.01 | 0.03 | 1.00 | 0.03 | 0.93 | 0.03 | 0.87 | 0.03 | 1.00 | 0.03 | | | | | | |
| 8.5 | 1.14 | 0.03 | 1.05 | 0.03 | 1.02 | 0.03 | 1.00 | 0.03 | 0.88 | 0.03 | 1.01 | 0.03 | | | | | | |
| 9.5 | 1.02 | 0.03 | 1.00 | 0.03 | 1.02 | 0.03 | 1.04 | 0.03 | 0.98 | 0.03 | 1.08 | 0.03 | | | | | | |
| 10.5 | | | | | | | 1.00 | 0.03 | 0.99 | 0.03 | 1.08 | 0.03 | | | | | | |
| 11.5 | 1.03 | 0.03 | 0.98 | 0.03 | 0.97 | 0.03 | 1.11 | 0.03 | 1.00 | 0.03 | 1.10 | 0.03 | | | | | | |
| 12.5 | 1.00 | 0.03 | 1.00 | 0.03 | 1.00 | 0.03 | 1.09 | 0.03 | 0.89 | 0.03 | 0.95 | 0.03 | | | | | | |
| 13.5 | | | | | | | 1.22 | 0.04 | 0.99 | 0.03 | 1.02 | 0.03 | | | | | | |
| 14.5 | | | | | | | 1.31 | 0.04 | 0.99 | 0.03 | 0.97 | 0.03 | | | | | | |
| 15.5 | | | | | | | 1.05 | 0.03 | 0.95 | 0.03 | 1.07 | 0.03 | | | | | | |
| 16.5 | | | | | | | 1.25 | 0.04 | 1.00 | 0.03 | 1.00 | 0.03 | | | | | | |
| 17.5 | | | | | | | | | | | | | | | | | | |
| 18.5 | | | | | | | | | | | | | | | | | | |
| 19.5 | | | | | | | | | | | | | | | | | | |

Table S4. Raw $L_n/T_n$ data used for fitting sample ROAD03 in Fig. 4D-F.

| Depth (mm) | Core 1 | | | | | | Core 2 | | | | | | Core 3 | | | | | |
|---|---|---|---|---|---|---|---|---|---|---|---|---|---|---|---|---|---|---|
| | $IR_{50}$ | | $pIRIR_{150}$ | | $pIRIR_{225}$ | | $IR_{50}$ | | $pIRIR_{150}$ | | $pIRIR_{225}$ | | $IR_{50}$ | | $pIRIR_{150}$ | | $pIRIR_{225}$ | |
| | $L_n/T_n$ | Error | $L_n/T_n$ | Error | $L_n/T_n$ | Error | $L_n/T_n$ | Error | $L_n/T_n$ | Error | $L_n/T_n$ | Error | $L_n/T_n$ | Error | $L_n/T_n$ | Error | $L_n/T_n$ | Error |
| 0.5 | 0.00 | 0.00 | 0.01 | 0.00 | 0.02 | 0.00 | 0.00 | 0.00 | 0.02 | 0.00 | 0.07 | 0.00 | 0.01 | 0.00 | 0.01 | 0.00 | 0.02 | 0.00 |
| 1.5 | 0.01 | 0.00 | 0.03 | 0.00 | 0.11 | 0.00 | 0.01 | 0.00 | 0.15 | 0.01 | 0.32 | 0.01 | 0.01 | 0.00 | 0.04 | 0.00 | 0.12 | 0.00 |
| 2.5 | 0.09 | 0.00 | 0.66 | 0.02 | 0.70 | 0.02 | 0.07 | 0.00 | 0.44 | 0.01 | 0.59 | 0.02 | 0.04 | 0.00 | 0.43 | 0.01 | 0.56 | 0.02 |
| 3.5 | | | | | | | 0.10 | 0.00 | 0.40 | 0.01 | 0.41 | 0.01 | 0.12 | 0.00 | 0.72 | 0.02 | 0.71 | 0.02 |
| 4.5 | 0.65 | 0.02 | 1.03 | 0.03 | 1.05 | 0.03 | 0.74 | 0.02 | 0.93 | 0.03 | 1.03 | 0.03 | 0.93 | 0.03 | 0.96 | 0.03 | 0.71 | 0.02 |
| 5.5 | 1.00 | 0.03 | 1.00 | 0.03 | 1.00 | 0.03 | 0.95 | 0.03 | 0.99 | 0.03 | 1.05 | 0.03 | 1.00 | 0.03 | 0.92 | 0.03 | 0.88 | 0.03 |
| 6.5 | | | | | | | 1.32 | 0.04 | 1.26 | 0.04 | 1.23 | 0.04 | 0.77 | 0.02 | 0.81 | 0.02 | 0.90 | 0.03 |
| 7.5 | | | | | | | 1.16 | 0.04 | 1.06 | 0.03 | 1.02 | 0.03 | 0.90 | 0.03 | 0.95 | 0.03 | 0.96 | 0.03 |
| 8.5 | | | | | | | 1.05 | 0.03 | 1.05 | 0.03 | 1.07 | 0.03 | 0.85 | 0.03 | 0.89 | 0.03 | 0.87 | 0.03 |
| 9.5 | | | | | | | 0.85 | 0.03 | 0.91 | 0.03 | 0.90 | 0.03 | 0.82 | 0.03 | 0.87 | 0.03 | 0.92 | 0.03 |
| 10.5 | | | | | | | 1.01 | 0.03 | 0.99 | 0.03 | 1.08 | 0.03 | 1.00 | 0.03 | 0.86 | 0.03 | 0.86 | 0.03 |
| 11.5 | | | | | | | 1.02 | 0.03 | 1.08 | 0.03 | 1.09 | 0.03 | 0.99 | 0.03 | 1.00 | 0.03 | 1.00 | 0.03 |
| 12.5 | | | | | | | 1.04 | 0.03 | 0.97 | 0.03 | 0.98 | 0.03 | 0.85 | 0.03 | 0.80 | 0.02 | 1.00 | 0.03 |
| 13.5 | | | | | | | 1.04 | 0.03 | 1.07 | 0.03 | 1.09 | 0.03 | 0.99 | 0.03 | 0.92 | 0.03 | 0.90 | 0.03 |
| 14.5 | | | | | | | 1.03 | 0.03 | 1.02 | 0.03 | 1.05 | 0.03 | | | | | | |
| 15.5 | | | | | | | 0.96 | 0.03 | 1.07 | 0.03 | 1.14 | 0.04 | | | | | | |
| 16.5 | | | | | | | 0.93 | 0.03 | 0.98 | 0.03 | 1.04 | 0.03 | | | | | | |
| 17.5 | | | | | | | 1.08 | 0.03 | 1.05 | 0.03 | 1.07 | 0.03 | | | | | | |
| 18.5 | | | | | | | 1.09 | 0.03 | 1.17 | 0.04 | 1.14 | 0.04 | | | | | | |
| 19.5 | | | | | | | 1.00 | 0.03 | 1.00 | 0.03 | 1.00 | 0.03 | | | | | | |

Table S5. Raw $L_n/T_n$ data used for fitting sample BALL01 in Fig. 5A-C.

| Depth (mm) | Core 1 | | | | | | Core 2 | | | | | | Core 3 | | | | | | Core 4 | | | | | |
|---|---|---|---|---|---|---|---|---|---|---|---|---|---|---|---|---|---|---|---|---|---|---|---|---|
| | $IR_{50}$ | | $pIRIR_{150}$ | | $pIRIR_{225}$ | | $IR_{50}$ | | $pIRIR_{150}$ | | $pIRIR_{225}$ | | $IR_{50}$ | | $pIRIR_{150}$ | | $pIRIR_{225}$ | | $IR_{50}$ | | $pIRIR_{150}$ | | $pIRIR_{225}$ | |
| | $L_n/T_n$ | Error | $L_n/T_n$ | Error | $L_n/T_n$ | Error | $L_n/T_n$ | Error | $L_n/T_n$ | Error | $L_n/T_n$ | Error | $L_n/T_n$ | Error | $L_n/T_n$ | Error | $L_n/T_n$ | Error | $L_n/T_n$ | Error | $L_n/T_n$ | Error | $L_n/T_n$ | Error |
| 0.5 | 0.70 | 0.02 | 0.93 | 0.03 | 0.99 | 0.03 | 1.25 | 0.04 | 1.28 | 0.04 | 1.19 | 0.04 | | | | | | | 1.99 | 0.06 | 1.50 | 0.05 | 1.35 | 0.04 |
| 1.5 | 0.93 | 0.03 | 1.06 | 0.03 | 1.02 | 0.03 | 1.17 | 0.04 | 1.30 | 0.04 | 1.18 | 0.04 | 1.28 | 0.04 | 0.82 | 0.03 | 0.96 | 0.03 | 1.21 | 0.04 | 1.13 | 0.03 | 1.13 | 0.04 |
| 2.5 | 1.21 | 0.04 | 1.14 | 0.03 | 1.07 | 0.03 | 0.71 | 0.02 | 0.79 | 0.02 | 0.80 | 0.02 | 0.72 | 0.02 | 0.75 | 0.02 | 0.79 | 0.02 | 1.12 | 0.03 | 1.15 | 0.04 | 1.15 | 0.04 |
| 3.5 | 1.00 | 0.03 | 1.08 | 0.03 | 1.10 | 0.03 | 0.84 | 0.03 | 1.06 | 0.03 | 1.00 | 0.03 | | | | | | | 1.55 | 0.05 | 1.28 | 0.04 | 1.18 | 0.04 |
| 4.5 | 0.86 | 0.03 | 1.02 | 0.03 | 1.05 | 0.03 | 1.00 | 0.03 | 1.10 | 0.03 | 1.00 | 0.03 | 1.18 | 0.04 | 1.03 | 0.03 | 0.92 | 0.03 | 1.19 | 0.04 | 1.05 | 0.03 | 1.08 | 0.03 |
| 5.5 | 0.73 | 0.02 | 0.80 | 0.02 | 0.89 | 0.03 | 1.14 | 0.03 | 1.26 | 0.04 | 1.14 | 0.04 | | | | | | | 1.26 | 0.04 | 0.96 | 0.03 | 0.94 | 0.03 |
| 6.5 | 0.79 | 0.02 | 1.00 | 0.03 | 1.00 | 0.03 | 0.71 | 0.02 | 1.00 | 0.03 | 1.00 | 0.03 | 1.12 | 0.03 | 0.92 | 0.03 | 0.83 | 0.03 | 1.00 | 0.03 | 1.00 | 0.03 | 1.00 | 0.03 |
| 7.5 | 1.02 | 0.03 | 1.15 | 0.04 | 1.10 | 0.03 | | | | | | | 0.82 | 0.03 | 0.79 | 0.02 | 0.83 | 0.03 | 1.10 | 0.03 | 0.90 | 0.03 | 0.88 | 0.03 |
| 8.5 | 0.72 | 0.02 | 0.78 | 0.02 | 0.84 | 0.03 | | | | | | | 1.35 | 0.04 | 1.10 | 0.03 | 0.96 | 0.03 | 1.01 | 0.03 | 0.92 | 0.03 | 0.91 | 0.03 |
| 9.5 | 1.27 | 0.04 | 1.15 | 0.04 | 1.09 | 0.03 | | | | | | | | | | | | | 1.18 | 0.04 | 0.99 | 0.03 | 0.90 | 0.03 |
| 10.5 | 0.70 | 0.02 | 1.03 | 0.03 | 1.08 | 0.03 | | | | | | | 0.84 | 0.03 | 0.81 | 0.02 | 0.82 | 0.03 | 1.31 | 0.04 | 1.00 | 0.03 | 0.92 | 0.03 |
| 11.5 | 0.49 | 0.02 | 0.57 | 0.02 | 0.64 | 0.02 | | | | | | | | | | | | | 1.30 | 0.04 | 1.06 | 0.03 | 0.97 | 0.03 |
| 12.5 | 0.40 | 0.01 | 0.63 | 0.02 | 0.70 | 0.02 | | | | | | | 0.90 | 0.03 | 0.86 | 0.03 | 0.86 | 0.03 | 0.98 | 0.03 | 0.93 | 0.03 | 0.93 | 0.03 |
| 13.5 | | | | | | | | | | | | | 1.00 | 0.03 | 1.00 | 0.03 | 1.00 | 0.03 | | | | | | |
| 14.5 | | | | | | | | | | | | | | | | | | | | | | | | |
| 15.5 | | | | | | | | | | | | | | | | | | | | | | | | |
| 16.5 | | | | | | | | | | | | | | | | | | | | | | | | |
| 17.5 | | | | | | | | | | | | | | | | | | | | | | | | |
| 18.5 | | | | | | | | | | | | | | | | | | | | | | | | |
| 19.5 | | | | | | | | | | | | | | | | | | | | | | | | |

Table S6. Raw $L_n/T_n$ data used for fitting sample BALL02 in Fig. 5D-F.

| Depth (mm) | Core 1 IR$_{50}$ $L_n/T_n$ | Error | Core 1 pIRIR$_{150}$ $L_n/T_n$ | Error | Core 1 pIRIR$_{225}$ $L_n/T_n$ | Error | Core 2 IR$_{50}$ $L_n/T_n$ | Error | Core 2 pIRIR$_{150}$ $L_n/T_n$ | Error | Core 2 pIRIR$_{225}$ $L_n/T_n$ | Error | Core 3 IR$_{50}$ $L_n/T_n$ | Error | Core 3 pIRIR$_{150}$ $L_n/T_n$ | Error | Core 3 pIRIR$_{225}$ $L_n/T_n$ | Error |
|---|---|---|---|---|---|---|---|---|---|---|---|---|---|---|---|---|---|---|
| 0.5 | | | | | | | | | | | | | | | | | | |
| 1.5 | | | | | | | 0.00 | 0.00 | 0.04 | 0.00 | 0.16 | 0.01 | 0.01 | 0.00 | 0.02 | 0.00 | 0.04 | 0.00 |
| 2.5 | | | | | | | 0.01 | 0.00 | 0.15 | 0.01 | 0.40 | 0.01 | 0.03 | 0.00 | 0.46 | 0.01 | 0.61 | 0.02 |
| 3.5 | 0.39 | 0.01 | 0.89 | 0.03 | 0.91 | 0.03 | 0.35 | 0.01 | 0.93 | 0.03 | 0.91 | 0.03 | 0.16 | 0.01 | 1.08 | 0.03 | 1.03 | 0.03 |
| 4.5 | 1.21 | 0.04 | 1.05 | 0.03 | 1.03 | 0.03 | 0.80 | 0.02 | 0.98 | 0.03 | 1.03 | 0.03 | 0.87 | 0.03 | 1.29 | 0.04 | 1.17 | 0.04 |
| 5.5 | 1.05 | 0.03 | 0.96 | 0.03 | 1.04 | 0.03 | 1.16 | 0.04 | 1.24 | 0.04 | 1.19 | 0.04 | 0.84 | 0.03 | 1.12 | 0.03 | 1.09 | 0.03 |
| 6.5 | 1.02 | 0.03 | 0.95 | 0.03 | 0.97 | 0.03 | | | | | | | 0.99 | 0.03 | 1.20 | 0.04 | 1.12 | 0.04 |
| 7.5 | 0.98 | 0.03 | 0.83 | 0.03 | 0.87 | 0.03 | 1.06 | 0.03 | 1.04 | 0.03 | 1.01 | 0.03 | 0.96 | 0.03 | 1.09 | 0.03 | 1.01 | 0.03 |
| 8.5 | 1.04 | 0.03 | 1.02 | 0.03 | 1.03 | 0.03 | 1.15 | 0.03 | 1.18 | 0.04 | 1.15 | 0.04 | 0.90 | 0.03 | 1.16 | 0.04 | 1.08 | 0.03 |
| 9.5 | 0.94 | 0.03 | 0.93 | 0.03 | 0.94 | 0.03 | 1.23 | 0.04 | 1.06 | 0.03 | 0.98 | 0.03 | 0.92 | 0.03 | 1.10 | 0.03 | 1.07 | 0.03 |
| 10.5 | 0.92 | 0.03 | 0.98 | 0.03 | 0.99 | 0.03 | 1.40 | 0.04 | 1.22 | 0.04 | 1.14 | 0.04 | 1.00 | 0.03 | 1.00 | 0.03 | 1.00 | 0.03 |
| 11.5 | | | | | | | 1.03 | 0.03 | 1.12 | 0.03 | 1.17 | 0.04 | | | | | | |
| 12.5 | 1.13 | 0.03 | 1.02 | 0.03 | 1.01 | 0.03 | 1.01 | 0.03 | 1.09 | 0.03 | 1.05 | 0.03 | | | | | | |
| 13.5 | 1.15 | 0.04 | 0.93 | 0.03 | 0.93 | 0.03 | | | | | | | | | | | | |
| 14.5 | 1.00 | 0.03 | 1.00 | 0.03 | 1.00 | 0.03 | 1.02 | 0.03 | 1.07 | 0.03 | 1.08 | 0.03 | | | | | | |
| 15.5 | 1.05 | 0.03 | 0.92 | 0.03 | 0.94 | 0.03 | 1.00 | 0.03 | 1.00 | 0.03 | 1.00 | 0.03 | | | | | | |
| 16.5 | 1.14 | 0.03 | 1.06 | 0.03 | 1.04 | 0.03 | | | | | | | | | | | | |
| 17.5 | | | | | | | | | | | | | | | | | | |
| 18.5 | | | | | | | | | | | | | | | | | | |
| 19.5 | | | | | | | | | | | | | | | | | | |

Table S7. Raw $L_n/T_n$ data used for fitting sample BALL03 in Fig. 5G-I.

[revised manuscript text omitted]

Fig. S5. (a) Depth of bleaching fronts for the known-age samples ROAD01 (0.01 a), ROAD02 (57 a) and ROAD03 (44 a). Note that the depth of the pIRIR$_{150}$ and pIRIR$_{225}$ bleaching fronts were identical for ROAD01 (0.01 a known-age sample). (b) Residual D$_e$ values determined for the surface slice (0-1 mm depth) of each sample. Note that sample BALL01 is not plotted on this figure as the residual D$_e$ values were large for all the IRSL signals: IR$_{50}$ (477.5 ± 20.7 Gy), pIRIR$_{150}$ (574.6 ± 36.5 Gy) and pIRIR$_{225}$ (could not be interpolated on to the dose-response curve).

[Figure]

Fig. S6. Photograph of the Beinn Alligin rock avalanche to illustrate the nature of the area surrounding the sampled boulders, which includes dense vegetation, the small-scale topography of the boulders and little evidence of shattered material.

Table S1. Multi-elevated temperature post- IR IRSL sequence used for analysis.

| Step | Procedure |
|------|-----------|
| 1 | Natural or regenerative dose |
| 2 | Preheat 250 °C for 100 s |
| 3 | IR LEDs 50 °C for 200 s |
| 4 | IR LEDs 150 °C for 200 s |
| 5 | IR LEDs 225 °C for 200 s |
| 6 | Test-dose 53 Gy |
| 7 | Preheat 250 °C for 100 s |
| 8 | IR LEDs 50 °C for 200 s |
| 9 | IR LEDs 150 °C for 200 s |
| 10 | IR LEDs 225 °C for 200 s |
| 11 | IR LEDs 290 °C for 200 s |

Table S2. Raw $L_n/T_n$ data used for fitting sample ROAD01 in Fig. 4A-C. Note that two rather than three cores were analysed for this sample.

| Depth (mm) | Core 1 | | | | | | Core 2 | | | | | |
|---|---|---|---|---|---|---|---|---|---|---|---|---|
| | $IR_{50}$ | | $pIRIR_{150}$ | | $pIRIR_{225}$ | | $IR_{50}$ | | $pIRIR_{150}$ | | $pIRIR_{225}$ | |
| | $L_n/T_n$ | Error | $L_n/T_n$ | Error | $L_n/T_n$ | Error | $L_n/T_n$ | Error | $L_n/T_n$ | Error | $L_n/T_n$ | Error |
| 0.5 | 0.01 | 0.00 | 0.06 | 0.00 | 0.11 | 0.00 | 0.05 | 0.00 | 0.16 | 0.01 | 0.25 | 0.01 |
| 1.5 | 0.20 | 0.01 | 0.68 | 0.02 | 0.79 | 0.03 | 0.64 | 0.02 | 0.98 | 0.03 | 1.02 | 0.03 |
| 2.5 | 1.02 | 0.03 | 1.09 | 0.03 | 1.04 | 0.03 | 0.98 | 0.03 | 0.99 | 0.03 | 0.99 | 0.03 |
| 3.5 | 1.02 | 0.03 | 1.03 | 0.03 | 1.05 | 0.03 | 1.15 | 0.04 | 1.01 | 0.03 | 0.95 | 0.03 |
| 4.5 | 0.97 | 0.03 | 1.01 | 0.03 | 1.02 | 0.03 | 0.94 | 0.03 | 0.93 | 0.03 | 0.97 | 0.03 |
| 5.5 | 1.01 | 0.03 | 1.01 | 0.03 | 1.04 | 0.03 | 1.04 | 0.03 | 1.03 | 0.03 | 1.03 | 0.03 |
| 6.5 | 1.02 | 0.03 | 1.06 | 0.03 | 1.05 | 0.03 | 1.03 | 0.03 | 0.99 | 0.03 | 0.98 | 0.03 |
| 7.5 | 1.00 | 0.03 | 0.98 | 0.03 | 0.98 | 0.03 | 1.04 | 0.03 | 1.02 | 0.03 | 1.03 | 0.03 |
| 8.5 | 1.03 | 0.03 | 0.99 | 0.03 | 0.98 | 0.03 | 1.06 | 0.03 | 0.98 | 0.03 | 1.00 | 0.03 |
| 9.5 | 1.09 | 0.03 | 1.04 | 0.03 | 1.03 | 0.03 | 1.00 | 0.03 | 1.00 | 0.03 | 1.00 | 0.03 |
| 10.5 | 1.10 | 0.03 | 1.07 | 0.03 | 1.04 | 0.03 | | | | | | |
| 11.5 | 1.00 | 0.03 | 1.00 | 0.03 | 1.00 | 0.03 | | | | | | |
| 12.5 | | | | | | | | | | | | |
| 13.5 | | | | | | | | | | | | |
| 14.5 | | | | | | | | | | | | |
| 15.5 | | | | | | | | | | | | |
| 16.5 | | | | | | | | | | | | |
| 17.5 | | | | | | | | | | | | |
| 18.5 | | | | | | | | | | | | |
| 19.5 | | | | | | | | | | | | |

Table S3. Raw $L_n/T_n$ data used for fitting sample ROAD02 in Fig. 4G-I.

| Depth (mm) | Core 1 | | | | | | Core 2 | | | | | | Core 3 | | | | | |
|---|---|---|---|---|---|---|---|---|---|---|---|---|---|---|---|---|---|---|
| | $IR_{50}$ | | $pIRIR_{150}$ | | $pIRIR_{225}$ | | $IR_{50}$ | | $pIRIR_{150}$ | | $pIRIR_{225}$ | | $IR_{50}$ | | $pIRIR_{150}$ | | $pIRIR_{225}$ | |
| | $L_n/T_n$ | Error | $L_n/T_n$ | Error | $L_n/T_n$ | Error | $L_n/T_n$ | Error | $L_n/T_n$ | Error | $L_n/T_n$ | Error | $L_n/T_n$ | Error | $L_n/T_n$ | Error | $L_n/T_n$ | Error |
| 0.5 | 0.00 | 0.00 | 0.01 | 0.00 | 0.02 | 0.00 | | | | | | | 0.02 | 0.00 | 0.37 | 0.01 | 0.64 | 0.02 |
| 1.5 | 0.00 | 0.00 | 0.02 | 0.00 | 0.13 | 0.00 | | | | | | | 0.07 | 0.00 | 1.02 | 0.03 | 1.12 | 0.04 |
| 2.5 | 0.01 | 0.00 | 0.29 | 0.01 | 0.61 | 0.02 | 0.02 | 0.00 | 0.75 | 0.02 | 1.01 | 0.03 | 0.55 | 0.02 | 1.01 | 0.03 | 1.03 | 0.03 |
| 3.5 | 0.03 | 0.00 | 0.76 | 0.02 | 0.89 | 0.03 | | | | | | | 0.90 | 0.03 | 1.08 | 0.03 | 1.06 | 0.03 |
| 4.5 | 0.15 | 0.00 | 0.96 | 0.03 | 1.03 | 0.03 | 0.43 | 0.01 | 0.92 | 0.03 | 1.07 | 0.03 | 0.96 | 0.03 | 1.09 | 0.03 | 1.13 | 0.04 |
| 5.5 | 0.47 | 0.01 | 0.93 | 0.03 | 0.99 | 0.03 | 0.85 | 0.03 | 0.92 | 0.03 | 1.08 | 0.03 | 1.00 | 0.03 | 1.00 | 0.03 | 1.00 | 0.03 |
| 6.5 | 1.06 | 0.03 | 0.93 | 0.03 | 1.00 | 0.03 | 0.98 | 0.03 | 0.96 | 0.03 | 1.12 | 0.03 | | | | | | |
| 7.5 | 0.96 | 0.03 | 1.01 | 0.03 | 1.00 | 0.03 | 0.93 | 0.03 | 0.87 | 0.03 | 1.00 | 0.03 | | | | | | |
| 8.5 | 1.14 | 0.03 | 1.05 | 0.03 | 1.02 | 0.03 | 1.00 | 0.03 | 0.88 | 0.03 | 1.01 | 0.03 | | | | | | |
| 9.5 | 1.02 | 0.03 | 1.00 | 0.03 | 1.02 | 0.03 | 1.04 | 0.03 | 0.98 | 0.03 | 1.08 | 0.03 | | | | | | |
| 10.5 | | | | | | | 1.00 | 0.03 | 0.99 | 0.03 | 1.08 | 0.03 | | | | | | |
| 11.5 | 1.03 | 0.03 | 0.98 | 0.03 | 0.97 | 0.03 | 1.11 | 0.03 | 1.00 | 0.03 | 1.10 | 0.03 | | | | | | |
| 12.5 | 1.00 | 0.03 | 1.00 | 0.03 | 1.00 | 0.03 | 1.09 | 0.03 | 0.89 | 0.03 | 0.95 | 0.03 | | | | | | |
| 13.5 | | | | | | | 1.22 | 0.04 | 0.99 | 0.03 | 1.02 | 0.03 | | | | | | |
| 14.5 | | | | | | | 1.31 | 0.04 | 0.99 | 0.03 | 0.97 | 0.03 | | | | | | |
| 15.5 | | | | | | | 1.05 | 0.03 | 0.95 | 0.03 | 1.07 | 0.03 | | | | | | |
| 16.5 | | | | | | | 1.25 | 0.04 | 1.00 | 0.03 | 1.00 | 0.03 | | | | | | |
| 17.5 | | | | | | | | | | | | | | | | | | |
| 18.5 | | | | | | | | | | | | | | | | | | |
| 19.5 | | | | | | | | | | | | | | | | | | |

Table S4. Raw $L_n/T_n$ data used for fitting sample ROAD03 in Fig. 4D-F.

| Depth (mm) | Core 1 | | | | | | Core 2 | | | | | | Core 3 | | | | | |
|---|---|---|---|---|---|---|---|---|---|---|---|---|---|---|---|---|---|---|
| | $IR_{50}$ | | $pIRIR_{150}$ | | $pIRIR_{225}$ | | $IR_{50}$ | | $pIRIR_{150}$ | | $pIRIR_{225}$ | | $IR_{50}$ | | $pIRIR_{150}$ | | $pIRIR_{225}$ | |
| | $L_n/T_n$ | Error | $L_n/T_n$ | Error | $L_n/T_n$ | Error | $L_n/T_n$ | Error | $L_n/T_n$ | Error | $L_n/T_n$ | Error | $L_n/T_n$ | Error | $L_n/T_n$ | Error | $L_n/T_n$ | Error |
| 0.5 | 0.00 | 0.00 | 0.01 | 0.00 | 0.02 | 0.00 | 0.00 | 0.00 | 0.02 | 0.00 | 0.07 | 0.00 | 0.01 | 0.00 | 0.01 | 0.00 | 0.02 | 0.00 |
| 1.5 | 0.01 | 0.00 | 0.03 | 0.00 | 0.11 | 0.00 | 0.01 | 0.00 | 0.15 | 0.01 | 0.32 | 0.01 | 0.01 | 0.00 | 0.04 | 0.00 | 0.12 | 0.00 |
| 2.5 | 0.09 | 0.00 | 0.66 | 0.02 | 0.70 | 0.02 | 0.07 | 0.00 | 0.44 | 0.01 | 0.59 | 0.02 | 0.04 | 0.00 | 0.43 | 0.01 | 0.56 | 0.02 |
| 3.5 | | | | | | | 0.10 | 0.00 | 0.40 | 0.01 | 0.41 | 0.01 | 0.12 | 0.00 | 0.72 | 0.02 | 0.71 | 0.02 |
| 4.5 | 0.65 | 0.02 | 1.03 | 0.03 | 1.05 | 0.03 | 0.74 | 0.02 | 0.93 | 0.03 | 1.03 | 0.03 | 0.93 | 0.03 | 0.96 | 0.03 | 0.71 | 0.02 |
| 5.5 | 1.00 | 0.03 | 1.00 | 0.03 | 1.00 | 0.03 | 0.95 | 0.03 | 0.99 | 0.03 | 1.05 | 0.03 | 1.00 | 0.03 | 0.92 | 0.03 | 0.88 | 0.03 |
| 6.5 | | | | | | | 1.32 | 0.04 | 1.26 | 0.04 | 1.23 | 0.04 | 0.77 | 0.02 | 0.81 | 0.02 | 0.90 | 0.03 |
| 7.5 | | | | | | | 1.16 | 0.04 | 1.06 | 0.03 | 1.02 | 0.03 | 0.90 | 0.03 | 0.95 | 0.03 | 0.96 | 0.03 |
| 8.5 | | | | | | | 1.05 | 0.03 | 1.05 | 0.03 | 1.07 | 0.03 | 0.85 | 0.03 | 0.89 | 0.03 | 0.87 | 0.03 |
| 9.5 | | | | | | | 0.85 | 0.03 | 0.91 | 0.03 | 0.90 | 0.03 | 0.82 | 0.03 | 0.87 | 0.03 | 0.92 | 0.03 |
| 10.5 | | | | | | | 1.01 | 0.03 | 0.99 | 0.03 | 1.08 | 0.03 | 1.00 | 0.03 | 0.86 | 0.03 | 0.86 | 0.03 |
| 11.5 | | | | | | | 1.02 | 0.03 | 1.08 | 0.03 | 1.09 | 0.03 | 0.99 | 0.03 | 1.00 | 0.03 | 1.00 | 0.03 |
| 12.5 | | | | | | | 1.04 | 0.03 | 0.97 | 0.03 | 0.98 | 0.03 | 0.85 | 0.03 | 0.80 | 0.02 | 1.00 | 0.03 |
| 13.5 | | | | | | | 1.04 | 0.03 | 1.07 | 0.03 | 1.09 | 0.03 | 0.99 | 0.03 | 0.92 | 0.03 | 0.90 | 0.03 |
| 14.5 | | | | | | | 1.03 | 0.03 | 1.02 | 0.03 | 1.05 | 0.03 | | | | | | |
| 15.5 | | | | | | | 0.96 | 0.03 | 1.07 | 0.03 | 1.14 | 0.04 | | | | | | |
| 16.5 | | | | | | | 0.93 | 0.03 | 0.98 | 0.03 | 1.04 | 0.03 | | | | | | |
| 17.5 | | | | | | | 1.08 | 0.03 | 1.05 | 0.03 | 1.07 | 0.03 | | | | | | |
| 18.5 | | | | | | | 1.09 | 0.03 | 1.17 | 0.04 | 1.14 | 0.04 | | | | | | |
| 19.5 | | | | | | | 1.00 | 0.03 | 1.00 | 0.03 | 1.00 | 0.03 | | | | | | |

Table S5. Raw $L_n/T_n$ data used for fitting sample BALL01 in Fig. 5A-C.

| Depth (mm) | Core 1 | | | | | | Core 2 | | | | | | Core 3 | | | | | | Core 4 | | | | | |
|---|---|---|---|---|---|---|---|---|---|---|---|---|---|---|---|---|---|---|---|---|---|---|---|---|
| | $IR_{50}$ | | $pIRIR_{150}$ | | $pIRIR_{225}$ | | $IR_{50}$ | | $pIRIR_{150}$ | | $pIRIR_{225}$ | | $IR_{50}$ | | $pIRIR_{150}$ | | $pIRIR_{225}$ | | $IR_{50}$ | | $pIRIR_{150}$ | | $pIRIR_{225}$ | |
| | $L_n/T_n$ | Error | $L_n/T_n$ | Error | $L_n/T_n$ | Error | $L_n/T_n$ | Error | $L_n/T_n$ | Error | $L_n/T_n$ | Error | $L_n/T_n$ | Error | $L_n/T_n$ | Error | $L_n/T_n$ | Error | $L_n/T_n$ | Error | $L_n/T_n$ | Error | $L_n/T_n$ | Error |
| 0.5 | 0.70 | 0.02 | 0.93 | 0.03 | 0.99 | 0.03 | 1.25 | 0.04 | 1.28 | 0.04 | 1.19 | 0.04 | | | | | | | 1.99 | 0.06 | 1.50 | 0.05 | 1.35 | 0.04 |
| 1.5 | 0.93 | 0.03 | 1.06 | 0.03 | 1.02 | 0.03 | 1.17 | 0.04 | 1.30 | 0.04 | 1.18 | 0.04 | 1.28 | 0.04 | 0.82 | 0.03 | 0.96 | 0.03 | 1.21 | 0.04 | 1.13 | 0.03 | 1.13 | 0.04 |
| 2.5 | 1.21 | 0.04 | 1.14 | 0.03 | 1.07 | 0.03 | 0.71 | 0.02 | 0.79 | 0.02 | 0.80 | 0.02 | 0.72 | 0.02 | 0.75 | 0.02 | 0.79 | 0.02 | 1.12 | 0.03 | 1.15 | 0.04 | 1.15 | 0.04 |
| 3.5 | 1.00 | 0.03 | 1.08 | 0.03 | 1.10 | 0.03 | 0.84 | 0.03 | 1.06 | 0.03 | 1.00 | 0.03 | | | | | | | 1.55 | 0.05 | 1.28 | 0.04 | 1.18 | 0.04 |
| 4.5 | 0.86 | 0.03 | 1.02 | 0.03 | 1.05 | 0.03 | 1.00 | 0.03 | 1.10 | 0.03 | 1.00 | 0.03 | 1.18 | 0.04 | 1.03 | 0.03 | 0.92 | 0.03 | 1.19 | 0.04 | 1.05 | 0.03 | 1.08 | 0.03 |
| 5.5 | 0.73 | 0.02 | 0.80 | 0.02 | 0.89 | 0.03 | 1.14 | 0.03 | 1.26 | 0.04 | 1.14 | 0.04 | | | | | | | 1.26 | 0.04 | 0.96 | 0.03 | 0.94 | 0.03 |
| 6.5 | 0.79 | 0.02 | 1.00 | 0.03 | 1.00 | 0.03 | 0.71 | 0.02 | 1.00 | 0.03 | 1.00 | 0.03 | 1.12 | 0.03 | 0.92 | 0.03 | 0.83 | 0.03 | 1.00 | 0.03 | 1.00 | 0.03 | 1.00 | 0.03 |
| 7.5 | 1.02 | 0.03 | 1.15 | 0.04 | 1.10 | 0.03 | | | | | | | 0.82 | 0.03 | 0.79 | 0.02 | 0.83 | 0.03 | 1.10 | 0.03 | 0.90 | 0.03 | 0.88 | 0.03 |
| 8.5 | 0.72 | 0.02 | 0.78 | 0.02 | 0.84 | 0.03 | | | | | | | 1.35 | 0.04 | 1.10 | 0.03 | 0.96 | 0.03 | 1.01 | 0.03 | 0.92 | 0.03 | 0.91 | 0.03 |
| 9.5 | 1.27 | 0.04 | 1.15 | 0.04 | 1.09 | 0.03 | | | | | | | | | | | | | 1.18 | 0.04 | 0.99 | 0.03 | 0.90 | 0.03 |
| 10.5 | 0.70 | 0.02 | 1.03 | 0.03 | 1.08 | 0.03 | | | | | | | 0.84 | 0.03 | 0.81 | 0.02 | 0.82 | 0.03 | 1.31 | 0.04 | 1.00 | 0.03 | 0.92 | 0.03 |
| 11.5 | 0.49 | 0.02 | 0.57 | 0.02 | 0.64 | 0.02 | | | | | | | | | | | | | 1.30 | 0.04 | 1.06 | 0.03 | 0.97 | 0.03 |
| 12.5 | 0.40 | 0.01 | 0.63 | 0.02 | 0.70 | 0.02 | | | | | | | 0.90 | 0.03 | 0.86 | 0.03 | 0.86 | 0.03 | 0.98 | 0.03 | 0.93 | 0.03 | 0.93 | 0.03 |
| 13.5 | | | | | | | | | | | | | 1.00 | 0.03 | 1.00 | 0.03 | 1.00 | 0.03 | | | | | | |
| 14.5 | | | | | | | | | | | | | | | | | | | | | | | | |
| 15.5 | | | | | | | | | | | | | | | | | | | | | | | | |
| 16.5 | | | | | | | | | | | | | | | | | | | | | | | | |
| 17.5 | | | | | | | | | | | | | | | | | | | | | | | | |
| 18.5 | | | | | | | | | | | | | | | | | | | | | | | | |
| 19.5 | | | | | | | | | | | | | | | | | | | | | | | | |

Table S6. Raw $L_n/T_n$ data used for fitting sample BALL02 in Fig. 5D-F.

| Depth (mm) | Core 1 IR$_{50}$ $L_n/T_n$ | Error | Core 1 pIRIR$_{150}$ $L_n/T_n$ | Error | Core 1 pIRIR$_{225}$ $L_n/T_n$ | Error | Core 2 IR$_{50}$ $L_n/T_n$ | Error | Core 2 pIRIR$_{150}$ $L_n/T_n$ | Error | Core 2 pIRIR$_{225}$ $L_n/T_n$ | Error | Core 3 IR$_{50}$ $L_n/T_n$ | Error | Core 3 pIRIR$_{150}$ $L_n/T_n$ | Error | Core 3 pIRIR$_{225}$ $L_n/T_n$ | Error |
|---|---|---|---|---|---|---|---|---|---|---|---|---|---|---|---|---|---|---|
| 0.5 | | | | | | | | | | | | | | | | | | |
| 1.5 | | | | | | | 0.00 | 0.00 | 0.04 | 0.00 | 0.16 | 0.01 | 0.01 | 0.00 | 0.02 | 0.00 | 0.04 | 0.00 |
| 2.5 | | | | | | | 0.01 | 0.00 | 0.15 | 0.01 | 0.40 | 0.01 | 0.03 | 0.00 | 0.46 | 0.01 | 0.61 | 0.02 |
| 3.5 | 0.39 | 0.01 | 0.89 | 0.03 | 0.91 | 0.03 | 0.35 | 0.01 | 0.93 | 0.03 | 0.91 | 0.03 | 0.16 | 0.01 | 1.08 | 0.03 | 1.03 | 0.03 |
| 4.5 | 1.21 | 0.04 | 1.05 | 0.03 | 1.03 | 0.03 | 0.80 | 0.02 | 0.98 | 0.03 | 1.03 | 0.03 | 0.87 | 0.03 | 1.29 | 0.04 | 1.17 | 0.04 |
| 5.5 | 1.05 | 0.03 | 0.96 | 0.03 | 1.04 | 0.03 | 1.16 | 0.04 | 1.24 | 0.04 | 1.19 | 0.04 | 0.84 | 0.03 | 1.12 | 0.03 | 1.09 | 0.03 |
| 6.5 | 1.02 | 0.03 | 0.95 | 0.03 | 0.97 | 0.03 | | | | | | | 0.99 | 0.03 | 1.20 | 0.04 | 1.12 | 0.04 |
| 7.5 | 0.98 | 0.03 | 0.83 | 0.03 | 0.87 | 0.03 | 1.06 | 0.03 | 1.04 | 0.03 | 1.01 | 0.03 | 0.96 | 0.03 | 1.09 | 0.03 | 1.01 | 0.03 |
| 8.5 | 1.04 | 0.03 | 1.02 | 0.03 | 1.03 | 0.03 | 1.15 | 0.03 | 1.18 | 0.04 | 1.15 | 0.04 | 0.90 | 0.03 | 1.16 | 0.04 | 1.08 | 0.03 |
| 9.5 | 0.94 | 0.03 | 0.93 | 0.03 | 0.94 | 0.03 | 1.23 | 0.04 | 1.06 | 0.03 | 0.98 | 0.03 | 0.92 | 0.03 | 1.10 | 0.03 | 1.07 | 0.03 |
| 10.5 | 0.92 | 0.03 | 0.98 | 0.03 | 0.99 | 0.03 | 1.40 | 0.04 | 1.22 | 0.04 | 1.14 | 0.04 | 1.00 | 0.03 | 1.00 | 0.03 | 1.00 | 0.03 |
| 11.5 | | | | | | | 1.03 | 0.03 | 1.12 | 0.03 | 1.17 | 0.04 | | | | | | |
| 12.5 | 1.13 | 0.03 | 1.02 | 0.03 | 1.01 | 0.03 | 1.01 | 0.03 | 1.09 | 0.03 | 1.05 | 0.03 | | | | | | |
| 13.5 | 1.15 | 0.04 | 0.93 | 0.03 | 0.93 | 0.03 | | | | | | | | | | | | |
| 14.5 | 1.00 | 0.03 | 1.00 | 0.03 | 1.00 | 0.03 | 1.02 | 0.03 | 1.07 | 0.03 | 1.08 | 0.03 | | | | | | |
| 15.5 | 1.05 | 0.03 | 0.92 | 0.03 | 0.94 | 0.03 | 1.00 | 0.03 | 1.00 | 0.03 | 1.00 | 0.03 | | | | | | |
| 16.5 | 1.14 | 0.03 | 1.06 | 0.03 | 1.04 | 0.03 | | | | | | | | | | | | |
| 17.5 | | | | | | | | | | | | | | | | | | |
| 18.5 | | | | | | | | | | | | | | | | | | |
| 19.5 | | | | | | | | | | | | | | | | | | |

Table S7. Raw $L_n/T_n$ data used for fitting sample BALL03 in Fig. 5G-I.

| Depth (mm) | Core 1 | | | | | | Core 2 | | | | | | Core 3 | | | | | |
|---|---|---|---|---|---|---|---|---|---|---|---|---|---|---|---|---|---|---|
| | $IR_{50}$ | | $pIRIR_{150}$ | | $pIRIR_{225}$ | | $IR_{50}$ | | $pIRIR_{150}$ | | $pIRIR_{225}$ | | $IR_{50}$ | | $pIRIR_{150}$ | | $pIRIR_{225}$ | |
| | $L_n/T_n$ | Error | $L_n/T_n$ | Error | $L_n/T_n$ | Error | $L_n/T_n$ | Error | $L_n/T_n$ | Error | $L_n/T_n$ | Error | $L_n/T_n$ | Error | $L_n/T_n$ | Error | $L_n/T_n$ | Error |
| 0.5 | | | | | | | | | | | | | 0.00 | 0.00 | 0.01 | 0.00 | 0.03 | 0.00 |
| 1.5 | | | | | | | | | | | | | 0.01 | 0.00 | 0.04 | 0.00 | 0.11 | 0.00 |
| 2.5 | 0.00 | 0.00 | 0.07 | 0.00 | 0.26 | 0.01 | 0.01 | 0.00 | 0.26 | 0.01 | 0.64 | 0.02 | 0.01 | 0.00 | 0.15 | 0.00 | 0.34 | 0.01 |
| 3.5 | | | | | | | 0.01 | 0.00 | 0.33 | 0.01 | 0.62 | 0.02 | 0.04 | 0.00 | 0.52 | 0.02 | 0.76 | 0.02 |
| 4.5 | 0.06 | 0.00 | 0.72 | 0.02 | 0.76 | 0.02 | 0.07 | 0.00 | 0.80 | 0.02 | 0.98 | 0.03 | 0.33 | 0.01 | 0.85 | 0.03 | 0.85 | 0.03 |
| 5.5 | 0.25 | 0.01 | 0.86 | 0.03 | 0.78 | 0.02 | 0.39 | 0.01 | 0.88 | 0.03 | 0.95 | 0.03 | 0.78 | 0.02 | 0.91 | 0.03 | 0.90 | 0.03 |
| 6.5 | 1.11 | 0.03 | 1.03 | 0.03 | 0.92 | 0.03 | 0.78 | 0.02 | 1.12 | 0.03 | 1.16 | 0.04 | 0.75 | 0.02 | 0.75 | 0.02 | 0.80 | 0.02 |
| 7.5 | 1.29 | 0.04 | 1.17 | 0.04 | 1.04 | 0.03 | 0.88 | 0.03 | 0.98 | 0.03 | 1.01 | 0.03 | 0.88 | 0.03 | 0.86 | 0.03 | 0.92 | 0.03 |
| 8.5 | 1.06 | 0.03 | 1.03 | 0.03 | 0.97 | 0.03 | 0.74 | 0.02 | 1.00 | 0.03 | 1.11 | 0.03 | 0.99 | 0.03 | 0.93 | 0.03 | 0.95 | 0.03 |
| 9.5 | 1.07 | 0.03 | 1.12 | 0.03 | 1.05 | 0.03 | 1.00 | 0.03 | 1.00 | 0.03 | 1.00 | 0.03 | 1.15 | 0.04 | 1.09 | 0.03 | 1.06 | 0.03 |
| 10.5 | 1.15 | 0.03 | 1.04 | 0.03 | 1.00 | 0.03 | | | | | | | 1.16 | 0.04 | 1.07 | 0.03 | 1.04 | 0.03 |
| 11.5 | 1.07 | 0.03 | 1.00 | 0.03 | 0.92 | 0.03 | | | | | | | 1.01 | 0.03 | 0.95 | 0.03 | 0.97 | 0.03 |
| 12.5 | 1.12 | 0.03 | 1.01 | 0.03 | 0.97 | 0.03 | | | | | | | 1.00 | 0.03 | 1.00 | 0.03 | 1.00 | 0.03 |
| 13.5 | 1.29 | 0.04 | 1.03 | 0.03 | 0.97 | 0.03 | | | | | | | | | | | | |
| 14.5 | 1.13 | 0.03 | 1.15 | 0.04 | 1.09 | 0.03 | | | | | | | | | | | | |
| 15.5 | 0.99 | 0.03 | 0.95 | 0.03 | 0.93 | 0.03 | | | | | | | | | | | | |
| 16.5 | 1.00 | 0.03 | 1.00 | 0.03 | 1.00 | 0.03 | | | | | | | | | | | | |
| 17.5 | | | | | | | | | | | | | | | | | | |
| 18.5 | | | | | | | | | | | | | | | | | | |
| 19.5 | | | | | | | | | | | | | | | | | | |

---

## Author Response (AR4)

**Responses to compiled reviewer and editor comments for manuscript titled:** *Erosion rates in a wet, temperate climate derived from rock luminescence techniques*.

In this document, we detail changes to the manuscript in response to each of the reviewer and editor comments for the **rounds of revisions** and the **Editors follow-up comments**. You will find the second round of revisions at the end of this document. We use the following abbreviations. Please note that the ACs are our responses to the (unedited) reviewer and editor comments.

Reviewer 1 (Benjamin Lehmann) comments: RC1

Reviewer 2 (Anonymous) comments: RC2

Editor comments: EC

Author comments: AC

**First round of revisions**

**Reviewer 1: Benjamin Lehmann (RC1)**

**General comments**

**RC1:** This paper presents a study on the erosion rate history of rock avalanche deposit using surface exposure datings from optically stimulated luminescence signals. The authors present luminescence signals of calibration sites and from rock boulder surfaces. The setting of the rock avalanche deposits is framed by terrestrial cosmogenic nuclide (TCN) dating from the literature. The study is well organized and takes advantage of the previous work on this OSL application by using rock color and texture observations in order to choose the most appropriated calibration samples. The main innovation of this study is the use of a multi-elevated temperature, post-infra-red, infra-red stimulated luminescence (MET-pIRIR) protocol (50, 150 and 225°C) allowing the identification of samples complexities and bringing more constrain to calculate exposure datings and erosion rate histories.

Overall the paper is well written, easy to follow but the figures lake in clarity and bring confusion to the reader. Indeed, Figure 7 is supposed to convince of the good quality of the inversion of the erosion history (erosion rate and time at which the erosion is switch on) from the experimental luminescence signals, but in its current form, the figure does not allow any visual inspection and validation of the results of erosion rate history.

Also, the authors use an approach developed by Lehmann et al., 2019a, in which the use of OSL signals from bedrock surface allows to calculate an erosion correction over TCN dating. Here, this approach in not fully exploited. Erosion rate histories are determined but are not used to discuss the possible erosion correction of the TCN exposure dating.

**AC**: This is a very good suggestion that we have included in the paper, please see a full explanation in response to your comment on Lines 271-281, which is directly related to this comment.

**RC1:** Finally, this study brings important observations on the differences of bleaching depth according of the energy traps targeted by the OSL stimulation. The IR50 signals are bleached deeper than pIR150 which is bleached deeper than pIR225, in a way that the higher the temperature of stimulation is, the longer it takes to the light exposure to affect the OSL signal in depth. However, the discussion on the difference of bleaching rate of the different signal could have be brought further.

Does the difference in bleaching depth of the different stimulations of a same sample could give us information about complex burial/exposure histories? Do the signals from different stimulations would have the same bleaching difference in a steady state or with a transient state with erosion?

**AC**: Although, these are very interesting ideas to explore in the future and we thank the reviewer for sharing them, we do not currently consider that our data provides any evidence to explore them to any depth within the discussion of this paper as our samples have very simple burial/exposure histories. As such, we have included a statement to highlight these future research avenues where we discuss the value of using the MET protocol in Section 6.1 as we consider the ideas to be valuable for future research avenues in light of our findings. See below:

Beyond this, it is possible that the MET-pIRIR protocol may be useful in identifying complex burial or exposure histories of rocks, similar to those that have been reported in previous studies but solely using the $IR_{50}$ signal (e.g. Freiesleben et al. 2015; Brill et al. 2021). There is also potential to explore whether the different temperature IRSL signals of the MET protocol record different states of erosion (i.e. steady or transient states) within the same rock surface, whereby the pIRIR signals that are attenuated greater are more susceptible to transient states of erosion in comparison to the lower temperature signals, which measure luminescence depth profiles to greater depths within the rock surface.

**Specific comments and technical corrections**

**RC1: Line 53:** The authors should mentioned the work of Brown and Moon, 2019* and Brown, 2020**.

**AC**: We have added mention of these works in Section 1. It includes the following sentences at the appropriate positions within the paragraph:

Brown (2020) even combine these phenomena within model simulations to explore different sample histories of exposure and burial to inform geomorphological interpretations of luminescence depth profiles measured for samples collected from the natural environment.

Recent findings from erosion simulations compared with measured data have shown that the erosion rates derived from luminescence depth profiles can be accurate even with stochastic erosion as experience in nature (Brown and Moon, 2019).

**RC1: Line 90:** The authors should mentioned the work of Brill et al., 2021***.

**AC**: We agree and have added the work of Brill et al. (2021) on carbonate limestones into our text:

Studies have applied rock luminescence techniques (mostly exposure dating) to a variety of lithologies including granites, gneisses (Lehmann et al. 2018 2019a,b; Meyer et al. 2019), sandstones (Sohbati et al. 2012; Chapot et al. 2012; Pederson et al. 2014), quartzites (Gliganic et al. 2019) and carbonate limestone (Brill et al. 2021).

**RC1: Lines 208-213:** How do the raw data Lx/Tx were normalised (L0 determination)? Is it done for each core individually or for each sample (considering the same L0 for every core of a same sample)? Core 3 for IR50 signal of sample ROAD3 (Fig. 4D) for example, seems to be normalised too low. I would recommend to normalise each independently. The normalisation approach should also be mentioned the Fig. 4 caption.

**AC**: The raw data $L_x/T_x$ for each core was normalised individually as you suggest. There is a large amount of scatter in the saturated plateau of the cores (as is typical in rock luminescence depth profiles) which makes Core 3 for $IR_{50}$ signal of sample ROAD03 (Fig. 4D) look low, but it is not inconsistent with the rest of the samples. Based upon your suggestion, we have added an explanation of the normalisation approach in the caption of Fig. 4:

All of the raw $L_n/T_n$ data presented in this figure were normalised individually for each core and subsequent analysis uses the data in this format.

**RC1: Lines 219-221:** The difference in depth of the bleaching front regarding the difference of ages of sample ROAD02 and ROAD03 could be explain also with the noise of the signal, the orientation of the sampled faces, difference at the rock surface.

**AC**: We have added this as an example within the text to illustrate factors that may influence the light penetration in our calibration samples as suggested:

This suggests that either another factor is influencing light penetration with depth in these rocks (e.g. small differences in the orientation of the sampled rock faces)…

**RC1: Lines 214-221:** How does the fit (black lines) in Fig. 4 are produced? This should be mentionned in the main text and in the caption of Fig. 4.

**AC**: In the text we have added the following sentence in response to this comment:

Note that the inferred models shown in Fig. 4 were fit using the $\overline{\sigma\varphi_0}$ and μ values included in each figure. See Section 5.2 for further explanation of the estimation of the model parameters.

While in the caption of Fig. 4 we have included an explanation of how the fit was produced and also provided the sigmaphi and mu values in each figure to provide further clarification (this is in response to other reviewer comments):

The black line shown is the inferred model that was fitted to derive the corresponding $\overline{\sigma\varphi_0}$ and μ values included in each figure.

**RC1: Line 229:** "sample BALL01 was coarser grained than BALL02 and BALL03" this affirmation is not supported by the results shown in Fig. 3B where only the two first discs of sample BALL01 are coarser than the other samples BALLs but deeper into the signal BALL03 seems to has the coarsest grained texture.

**AC**: We have edited the text so that it specifies that the surface is coarser:

… the surface of sample BALL01 was coarser grained than BALL02 and BALL03 (Fig. 2; Fig. 3b).

**RC1: Lines 232-233:** "[…] lost during sample and/or sample preparation […]". Are there any ways to invalidate this experimental bias? During the sampling, did you mark the exposed surface with ink? Did you measure the core before and after slicing? Does the surface of the first disc look alike the surface of other 1$^{st}$ disc of other cores?

**AC**: Yes, we did all those suggestions during sampling and preparation as we were very careful. Prior to writing the paper, we revisited all information on this sample, visually inspecting all of the discs, cores and core holes in the rocks, and could not find any other explanation. We have added a clarification of this into the text, but cannot invalidate the statement definitively so retain the narrative. See below:

Thus, although care was taken when sampling to mark the surface of rock and to measure the rock cores before and after slicing, it is possible that the luminescence depth profile (likely <10 mm based on BALL02 and BALL03) was lost during sampling and/or sample preparation due to the presence of a fragile weathering crust, potentially with a sub-surface zone of weakness (e.g. Robinson and Williams, 1987).

**RC1: Lines 244-245:** "sigmaphi0 and μ were calibrated using the known-age sample […]" Reading this sentence I was confused that ROADs samples are the calibration samples. "ROAD samples" could be explicitly mentioned in this sentence to improve clarity.

**AC**: We have added this to the sentence:

$\overline{\sigma\varphi_0}$ and μ were calibrated using Eq (1) and the known-age samples (ROAD01, ROAD02 and ROAD03) of similar, suitable rock composition as determined by the down-core profiles of RGB and grainsize (Section 4.2).

**RC1: Lines 253-258:** There is no mention of the results for the calibration of sigmaphi0 and μ parameters. They should be explicitly written in the main text.

**AC**: We had original omitted this because it turns into a long list in the text that is much more easily read from the table. However, to respond to the reviewers comment here, we have added a few sentences listing the parameters in the text:

For ROAD01, the parameters determined using the $IR_{50}$ (μ = 3.2 mm$^{-1}$, $\overline{\sigma\varphi_0}$ = 2.80e$^{-4}$ s$^{-1}$), pIRIR$_{150}$ (μ = 3.1 mm$^{-1}$, $\overline{\sigma\varphi_0}$ = 3.27e$^{-5}$ s$^{-1}$) and pIRIR$_{225}$ (μ = 3.0 mm$^{-1}$, $\overline{\sigma\varphi_0}$ = 2.88e$^{-5}$ s$^{-1}$) signals were broadly consistent. For ROAD02, the parameters differed between the $IR_{50}$ (μ = 2.1 mm$^{-1}$, $\overline{\sigma\varphi_0}$ = 6.67e$^{-6}$ s$^{-1}$), pIRIR$_{150}$ (μ = 1.5 mm$^{-1}$, $\overline{\sigma\varphi_0}$ = 1.73e$^{-8}$ s$^{-1}$) and pIRIR$_{225}$ (μ = 2.8 mm$^{-1}$, $\overline{\sigma\varphi_0}$ = 9.01e$^{-8}$ s$^{-1}$) signals, but the values for each signal were broadly similar to the equivalent values determined for ROAD03 using the $IR_{50}$ (μ = 2.7 mm$^{-1}$, $\overline{\sigma\varphi_0}$ = 1.56e$^{-5}$ s$^{-1}$), pIRIR$_{150}$ (μ = 1.5 mm$^{-1}$, $\overline{\sigma\varphi_0}$ = 3.80e$^{-8}$ s$^{-1}$) and pIRIR$_{225}$ (μ = 1.4 mm$^{-1}$, $\overline{\sigma\varphi_0}$ = 1.70e$^{-8}$ s$^{-1}$) signals.

**RC1: Lines 258-260:** Note that the μ value for ROAD03 are not so different than the ones for ROAD02 even if the grain size are very different. Could you comment on that?

**AC**: Yes, we have added a comment as suggested:

Given the similarity of $\overline{\sigma\varphi_0}$ and μ determined using all three IRSL signals for ROAD02 and ROAD03 and the difference in grainsizes (Fig. 3B), it suggests that grainsize has a minimal impact upon the attenuation of light into a rock surface in comparison to other factors (e.g. mineralogy, surficial coatings).

**RC1: Lines 262-266:** All the values mentioned in this section are different than the values in Table 3!

**AC**: We thank the reviewer for noticing these mistakes and have corrected the text accordingly.

**RC1: Line 305:** The "(2018)" reference is wrong and it should "(2019a)"

**AC**: We have corrected this in text.

**RC1: Lines 271-281:** Now that you inferred erosion history is determined for the two boulders, what are the consequences on the cosmo age of the deposit? Does the sampled boulders are the same sampled for TCN dating? If not, it would be interested to discuss the potential exposure age correction. Lehmann et al., 2019a approach does correct the TCN age with the inferred erosion history. It seems that the approach is not fully exploited here.

**AC:** This is a very good suggestion that we have incorporated into the paper as a paragraph within Section 6.2, which is included below:

The modelled erosion histories that we have calculated here using the luminescence erosion-meter for samples BALL02 and BALL03 would have had a minimal effect upon the cosmogenic nuclide exposure age (4.54 ± 0.27 ka; Ballantyne and Stone, 2004). Only the steady-state erosion rate of 66 mm/ka inferred for BALL02 using the $IR_{50}$ signal, when applied for durations exceeding 1 ka, would have increased the exposure age to any great degree. For example, when the steady-state erosion rate of 66 mm/ka was applied for 0.1 ka, the corrected cosmogenic nuclide exposure age would have been 4.58 ka and, when the same erosion rate was applied for 1 ka it would have been 4.99 ka; these corrected ages are consistent within ± 2 σ uncertainties of the uncorrected age of 4.54 ± 0.27 ka (Ballantyne and Stone, 2004). The higher, transient erosion rates inferred for BALL03 were all applied for such a short period of time (e.g. Table 3) that they had a minimal effect on the cosmogenic nuclide exposure age.

Based on the long-term erosion rates derived here, the boulder sampled for BALL02 would have lost a total of 300 mm ($IR_{50}$), 41 mm ($pIRIR_{150}$) and 54 mm ($pIRIR_{225}$) from the surface over 4.54 ka, while the long-term erosion rates determined for BALL03 suggested that the boulder surface would have lost 27 mm ($IR_{50}$), 64 mm ($pIRIR_{150}$) and 50 mm ($pIRIR_{225}$). All of these values (except for the $IR_{50}$ signal of BALL02) were broadly consistent with field observations of quartz protrusions on the surface of boulders >2 x 2 x 2 m that were densely distributed within the rock avalanche feature (Fig. 1). Alternatively, the maximum (shorter-term) erosion rate end members of the transient erosion histories would have removed 1407 mm (BALL02, $pIRIR_{225}$), 2088 mm (BALL03, $IR_{50}$), 454 mm (BALL03, $pIRIR_{150}$) and 817 mm (BALL03, $pIRIR_{225}$) from the boulder surface over the 4.54 ka. These large values were inconsistent with field evidence and so indicative of the transient state of erosion where high erosion rates were only sustained over short periods of time.

**RC1: Lines 355-365:** Do field observations of the deposit of weathered material on/in the ground/soil below the blocks have been done and would validate the hypothesis raised in this paragraph?

**AC**: We have included the sentence below in Section 6.3 to discuss field observations of weathered material and also included a photograph of the wider area in Fig. S6 newly included in the supplementary material to illustrate our observations:

This is also supported by a lack of shattered material surrounding the large sampled boulders (and in fact on much of the Beinn Alligin rock avalanche deposit), despite the presence of dense vegetation surrounding the boulders (e.g. Fig. S6.

**RC1: Figure 1:** This figure could be highly improved by adding in Panel B, the outline of the rock avalanche deposit, the elevation isoline or two elevation points and the coordinates. In Panel C, the north or flow direction of the rock avalanche deposit.

**AC**: We have added the outline of the rock avalanche into Fig. 1C in addition to the elevations. We have also added the rock avalanche flow direction into each image of Fig. 1D.

**RC1: Figure 2:** The scale could be directly placed on the figure.

**AC**: The scale has now been added to the figure.

**RC1: Figure 3:** The direction of the y-axis label should be turned 180° to be consistent with other figures.

**AC**: The axes have been turned by 180° so that the plot is consistent with other figures.

**RC1: Figure 4:** The uncertainties of the inversion could be plot as an envelope using ±1sigma of the µ and sigmaphi0.

**AC**: Thank you for this comment, it was very enlightening. We have produced inferred models for each of the ROAD samples using the $\pm$ 1 $\sigma$ $\overline{\sigma\varphi_0}$ and µ values as suggested here. These lines are included in Fig. 4 and we have added a sentence into the figure caption to describe how they were produced.

**RC1: Figure 6:** In every sub figure, an age is written in white, for example (0.01 a$^{-1}$) in Fig. 6A. Does unit [a$^{-1}$] is the correct unit? Also, the units are mentioned with "[ ]" but should be "( )" for consistency with the rest of the paper.

**AC**: You are absolutely correct, sorry for the mistake. We have corrected both the units and the brackets in Fig. 6.

**RC1:** Finally, the inversions for each stimulation of the ROAD01 sample appear to explore a truncated range of µ values, that is, the probabilities of 1 (yellow) reach the side of the inspection box. The µ values obtained will surely be much higher if the inversion will allow to explore the values of µ up to 5 or 6 mm$^{-1}$.

**AC**: We have not extended the axes of these inversions as ROAD01 had such a short exposure history of only 0.01 a. Thus, there is large uncertainty in these inversions. We use it as a means of demonstrating that even very short durations of exposure can lead to the development of a luminescence depth profile, but do not use them for calibrating any of our samples and so retain the

use of the original parameter space. If we were using them for calibration, we would of course provide a wider parameter space; however, it is unlikely that the data would then have such large uncertainties and so would likely not need this.

**RC1: Figure 7:** The formatting of this figure should be considerably improved. The panels A, B, C, G, H, I do not allow any visual inspection of the data and inversion qualities (for ex: the x-axis boundaries should be set between 0 and 20 mm). The inversions in C, G, H and I do not seem to fit to the experimental value. This figure should be THE figure of the paper, but in the current formatting, it removes persuasive force of the results on the erosion rate history and confuses the conveyed message by the study.

**AC**: To improve the data visualisation, we have removed the luminescence depth profiles from Fig. 7 and included the inferred erosion models in the luminescence depth profiles of Fig. 5. Fig. 7 now only includes the inversion model profiles of likelihoods. We have provided additional comments on the fit of the inferred erosion models to the experimental data in "Section 5.3 Apparent exposure ages and erosion rates".

**RC1: Figures S1, S4:** These figures are too pixelised and should be improved.

**AC**: Both figures have been re-exported, which has improved the resolution.

**RC1: Supplementary material:** Raw data of Lx/Tx luminescence with depth for every core/sample could be shared in the supplementary material.

**AC**: We have provided the raw data of the luminescence depth profiles in the supplementary material from Table S2-S7. We have also included a note explaining the availability of this data in the supplementary material in the figure captions.

**RC1: Formatting:** In general, there is a lack in consistency between the labelling of figures (i.e. Fig. 4 A, B) in uppercase letters and its mention in the main text (i.e. Fig 4 a, b) in lowercase letters.

**AC**: Apologies, I think some confusion arose around journal formatting regulations. We have corrected all of these formatting issues.

**RC1:** *Brown, N. D., & Moon, S. (2019). Revisiting erosion rate estimates from luminescence profiles in exposed bedrock surfaces using stochastic erosion simulations. Earth and Planetary Science Letters, 528, 115842.

**Brown, N.D. (2020) Which geomorphic processes can be informed by luminescence measurements?, Geomorphology, https://doi.org/10.1016/ j.geomorph.2020.107296

***Brill, D., May, S. M., Mhammdi, N., King, G., Lehmann, B., Burow, C., ... & Brückner, H. (2021). Evaluating optically stimulated luminescence rock surface exposure dating as a novel approach for reconstructing coastal boulder movement on decadal to centennial timescales. Earth Surface Dynamics, 9(2), 205-234.

**AC**: Thank you for the reference suggestions here, we have added them into the text and reference list.

**Reviewer 2: anonymous (RC2)**

**General remarks**

**RC2:** The paper "Erosion rates in a wet, temperate climate derived from rock luminescence techniques" presents new data for the application of luminescence rock surface techniques. The approach is applied to rock avalanche deposits from Scotland that have previously been dated by terrestrial cosmogenic nuclides to infer regional erosion rates for the last millennia. The study is well structured, well written and generally easy to follow. It presents valuable new results for the emerging topic of luminescence rock surface dating and erosion rate modelling that are highly needed to better understand the limitations and the potential of the technique. Innovative methodological aspects of the study are the use of a MET-post-IR-IRSL protocol to provide internal quality criteria for the selection of samples with appropriate lithology to adequately record light penetration into the rock surface.

There are several inconsistencies in the paper with regard to the presentation of the data. In particular numbers presented in the main text, the tables and figures do not always match (I will provide details below). This must be revised prior to publication.

**AC**: We thank the reviewer for bringing this to our attention and have corrected all of the inconsistencies and formatting issues within the paper. Please see the tracked changes document for details.

**RC2:** Another irritating aspect that needs clarification is the model fit for the unknown age samples in Figure 7. It seems that the inferred model does not really fit the measured data for most of the measured signals. This might indicate inadequate values for μ and/or sigmaphi and the authors should comment on that.

**AC**: This is a sound observation and so we have further explored this. Firstly, we have moved the inferred erosion model from Fig. 7 to Fig. 5, so that Fig. 7 now only includes the inversion profiles. We have also included further discussion into "Section 5.3 Apparent exposure ages and erosion rates" of the fits in response to the RCs:

At face value, the fit of the inferred erosion model to the experimental data for BALL02 using the $IR_{50}$ (Fig. 5D) and $pIRIR_{150}$ (Fig. 5E) signals is better than the equivalent fits for BALL02 using the $pIRIR_{225}$ signal (Fig. 5F) and BALL03 using the $IR_{50}$ (Fig. 5G), $pIRIR_{150}$ (Fig. 5H) and $pIRIR_{225}$ (Fig. 5I) signals. In the latter cases, the inferred erosion model is shallower than the experimental data. This could suggest that the $\overline{\sigma\varphi_0}$ and μ values were inaccurate, i.e. the attenuation of light with depth into the rock surface is lower in BALL02 and BALL03 than estimated by ROAD02. A possible explanation for this is that the surface of the roadcut sampled by ROAD02 (Fig. S1a) was orientated differently to the Beinn Alligin rock avalanche boulders sampled by BALL02 and BALL03 (Fig. 1), relative to the incoming sunlight (e.g. Gliganic et al. 2019). If the orientation of the known-age roadcut samples was inconsistent with the unknown samples, we would expect these inconsistencies to manifest similarly in all three MET signals for BALL02 and BALL03, which was not observed here. A factor that is common to all the profiles that are less well fit by the inferred erosion model is that they determined transient erosion rates. This suggests that these surfaces experienced complex erosional histories over time whereby the erosion was time-varying. Consequently, it is possible that surficial weathering

products may have changed in thickness and composition over time, which in turn could slightly vary the attenuation of light (Meyer et al. 2018; Luo et al. 2018), meaning that the calibration of $\overline{\sigma\varphi_0}$ and µ from ROAD02 would introduce more uncertainty into the inferred erosion model. It is possible that sample-specific measurements of $\overline{\sigma\varphi_0}$ and µ (e.g. Ou et al. 2018), rather than calibration from known-age samples, could reduce the uncertainty introduced by time-varying light attenuation. However, further investigation is required into the physical mechanisms of time-varying light attenuation in the context of surficial weathering and subsequent erosion, and the impacts upon inferred transient erosion rates.

**Specific comments**

**RC2: Lines 79-80**: Please add a reference for the insights on the exposure history.

**AC**: We have added the citation for the exposure history from Ballantyne and Stone (2004).

**RC2:** Lines 85-93: The shape and position of the bleaching front is also influenced by dose accumulation during exposure. Although this term is irrelevant for most samples, the authors should include a short explanation why they think it is not necessary to address dosing in their setting. It is also confusing that dose rates are considered in the methods section, while they are not part of equation (1). This is confusing and needs clarification.

**AC**: We have now added worked examples into "Section 2 Theoretical background" in response to the ECs comments, which provides improved clarification on how the dose accumulated over the exposure time is incorporated into the modelling. Please see comments below where we respond to the ECs comments for further information.

**RC2: Lines 104-107**: How do you consider temporal variability of µ? With your approach you rather account for spatial variability of light attenuation between different surfaces and in different depth levels of a surface.

**AC**: It is not possible to determine the temporal variability of µ as we cannot monitor this in real-time over thousands of years. As such, none of the existing studies consider the temporal variability of µ. Similarly, we cannot measure it here. However, we do include some consideration of the possibility that µ is time-dependent when exploring the potential explanations for the fit of the inferred erosion models to the experimental data in Section 5.2 in response to a number of comments from RC1 and RC2.

**RC2: Lines 176-177**: Please provide details regarding internal dose rate assessment here. What internal potassium contents were used? How exactly has sample grain size been determined?

**AC**: This information was provided in the caption for Table 1, however, we have also included the information in the text in response to the reviewers comment:

Internal dose-rates were calculated assuming an internal K-content of 10 ± 2 % (Smedley et al. 2012) and internal U and Th concentrations of 0.3 ± 0.1 ppm and 1.7 ± 0.4 ppm (Smedley and Pearce, 2016), in addition to measured average grain sizes for each sample.

**RC2: Line 190**: This should be "successively" instead of "simultaneously".

**AC**: We have exchanged these words in the text where suggested and in Section 4.1 where it also applies.

**RC2: Line 196**: Please provide the number of cores that were used per sample.

**AC**: We had already included this information in the original manuscript. See below:

De values were determined for the shallowest disc and the deepest disc from one core of each sample to quantify the natural residual dose and saturation limit (L0, Eq. 1), respectively.

**RC2: Line 202**: "…were in line with previous measurements of IRSL signals"

**AC**: We have corrected this in the text. See the tracked changes document.

**RC2: Lines 219-220**: How exactly was grain size determined with the microscope? Did you use a software or were grains measured manually? For the latter, how many grains were measured per slice?

**AC**: We have added additional information into the text to explain the methods of measuring grainsize. See below:

The average down-core grainsize of each sample was measured under an optical microscope using Infinity Analyze. For each rock slice of an example core per sample, ten randomly-selected grains were measured and the mean and standard deviation grainsize were calculated per core and plotted against the core depths (Fig. 3B).

**RC2: Line 235**: Which values for $\mu$ and sigmaphi have been used for the fits in Figure 4? I assume you used the parameters presented in section 5.2? If so, I would suggest to change the order of sections 5.1 and 5.2.

**AC**: In response to other reviewer's comments, the parameters used to fit the data shown in Fig. 4 are now included in each plot, which will clarify what parameters were used to fit each dataset. We have not changed the order of Sections 5.1 and 5.2 as Section 5.1 presents the data and describe the luminescence depth profiles in their raw form, and then Section 5.2 then fits this raw data to determine the parameters, so it would not make sense to us to switch the two sections around.

**RC2: Lines 236-237**: The numbers given for the depths of the IR50 bleaching front do not agree with the modelled fits in figure 4. Please clarify.

**AC**: This is a fair comment and we thank the reviewer for highlighting the point. We originally presented the depth of the first rock slice that measured an $L_n/T_n$ value that was >50 % of the

saturation plateau, which meant that it was much deeper than the model fit, and as highlighted here, misleading. We have rectified this by referring to the position of the bleaching front according to the fits show in Fig. 4 and corrected the text (see tracked changes document). We have also corrected the values presented in Fig. S4a.

**RC2: Lines 265-268**: Here dose rates are considered, although the term for dose accumulation is not part of equation (1). So either it was not equation (1) that was used for fitting, or the dose rate information is not needed. Please clarify.

**AC**: This is a fair comment as it was not clear exactly how the dose-rate information is used in the parameterisation of $\overline{\sigma\varphi_0}$ and μ, and age/erosion rate determination. We have provided more clarity in this section and refer the reader to our new worked example of the inversion model (requested in the Editors Comments - ECs) for further explanation, but we summarise relevant amendments here:

$\overline{\sigma\varphi_0}$ and μ were calibrated using Eq. 1 and the known-age samples (ROAD01, ROAD02 and ROAD03) of similar, suitable rock composition as determined by the down-core profiles of RGB and grainsize (Section 4.2). Note that ($\dot{D}$) is not considered in Eq. 1 but is used to determine an apparent exposure or erosion rate and so needs to be measured for each sample.

**RC2: Line 282**: The values of Sohbati et al. (2012) are for quartz signals, thus for a different wave length spectrum. Since different wave lengths are attenuated differently, I would suggest to compare with other feldspar studies.

**AC**: A very good point, thank you. We have removed reference to Sohbati et al. (2012).

**RC2: Lines 282-286**: Please provide the numbers for μ and sigmaphi directly in the text.

**AC**: We had originally omitted this because it turns into a long list in the text that is much more easily read from the table. However, to respond to the reviewers comment here, we have added a few sentences listing the parameters in the text:

For ROAD01, the parameters determined using the $IR_{50}$ (μ = 3.2 mm$^{-1}$, $\overline{\sigma\varphi_0}$ = 2.80e$^{-4}$ s$^{-1}$), $pIRIR_{150}$ (μ = 3.1 mm$^{-1}$, $\overline{\sigma\varphi_0}$ = 3.27e$^{-5}$ s$^{-1}$) and $pIRIR_{225}$ (μ = 3.0 mm$^{-1}$, $\overline{\sigma\varphi_0}$ = 2.88e$^{-5}$ s$^{-1}$) signals were broadly consistent. For ROAD02, the parameters differed between the $IR_{50}$ (μ = 2.1 mm$^{-1}$, $\overline{\sigma\varphi_0}$ = 6.67e$^{-6}$ s$^{-1}$), $pIRIR_{150}$ (μ = 1.5 mm$^{-1}$, $\overline{\sigma\varphi_0}$ = 1.73e$^{-8}$ s$^{-1}$) and $pIRIR_{225}$ (μ = 2.8 mm$^{-1}$, $\overline{\sigma\varphi_0}$ = 9.01e$^{-8}$ s$^{-1}$) signals, but the values for each signal were broadly similar to the equivalent values determined for ROAD03 using the $IR_{50}$ (μ = 2.7 mm$^{-1}$, $\overline{\sigma\varphi_0}$ = 1.56e$^{-5}$ s$^{-1}$), $pIRIR_{150}$ (μ = 1.5 mm$^{-1}$, $\overline{\sigma\varphi_0}$ = 3.80e$^{-8}$ s$^{-1}$) and $pIRIR_{225}$ (μ = 1.4 mm$^{-1}$, $\overline{\sigma\varphi_0}$ = 1.70e$^{-8}$ s$^{-1}$) signals.

**RC2: Lines 288-292**: The unit of the ages should be a instead of a$^{-1}$. Also the numbers do not match those in table 3.

**AC**: We have corrected these mistakes in the text.

**RC2: Lines 306-309**: Please provide numbers for erosion rates in the text.

**AC**: Similar to above, we had originally omitted this because it turns into a long list in the text that is much more easily read from the table. However, to respond to the reviewers comment here, we have added the following text listing the parameters in the text:

However, the pIRIR$_{225}$ signal suggested a transient erosion state, where the luminescence signal could be derived from numerous pairs of erosion rates and initiation times from a maximum erosion rate of 310 mm/ka over a minimum time interval of 4 a to a minimum erosion rate of 12 mm/ka over time a minimum time interval of 90 a. All three IRSL signals from sample BALL03 consistently suggested a system undergoing a transient response to erosion, which was consistent with the pIRIR$_{225}$ signal of BALL02 (Fig. 7, Table 3). The IR$_{50}$ signal for BALL03 derived a maximum erosion rate of 460 mm/ka over a minimum time interval of 3 a and a minimum erosion rate of 6 mm/ka over a minimum time interval of 231 a. The pIRIR$_{150}$ signal for BALL03 derived a maximum erosion rate of 100 mm/ka over minimum time interval of 19 a and a minimum erosion rate of 14 mm/ka over a minimum time interval of 137 a. The pIRIR$_{225}$ signal for BALL03 derived a maximum erosion rate of 180 mm/ka over a minimum time interval of 4 a and a minimum erosion rate of 11 mm/ka over a minimum time interval of 73 a.

**RC2:** Line 317: Based on the modelled fits shown in figure 7 (red lines), the fit seems not to match the measured data for most of the signals, which indeed indicates that the parameters might be inaccurate. This should be explored in more detail.

**AC**: This was also identified by RC2 in their more general comments above and so we have provided a more detailed response there of the further discussion that we have included exploring the fit of the inferred erosion models to the experimental data. The same response applies to this comment.

**RC2: Lines 327-329**: This assumption does not really make sense in my opinion. Post-IR$_{225}$ signals need shorter wave lengths than IR$_{50}$ and post-IR$_{150}$ signals to be reset. But the attenuation of shorter wave lengths in rocks tends to be stronger than that of longer wave lengths (cf. Ou et al., 2018).

**AC**: In this statement we are suggesting that it is the Fe-coating that forms as a weathering product on the surface of the rock that may have preferentially attenuated the longer wavelengths that would reset the IR$_{50}$ and pIRIR$_{150}$ signals, rather than the referring to the rock itself. An Fe-coating and crystalline rock would have very different compositions. However, we appreciate that this is a speculative statement and to respond to the reviewer's comment here, we have re-phrased this sentence to better reflect the evidence we present. See below:

Interestingly, the similarity between BALL02 and BALL03 for the pIRIR$_{225}$ signal suggests that the presence of an Fe-coating altered the attenuation of the IR$_{50}$ and pIRIR$_{150}$ signals to a lesser extent than the pIRIR$_{225}$ signal, but the reasons for this requires further investigation.

**RC2: Line 343**: I think it should be "exploited" or similar instead of "inferred"?

**AC**: We have considered this comment at length and currently do not understand how the word exploited would replace inferred. If the authors were exploiting the fact that shorter-term erosion rates derived from luminescence measurements were higher than the long-term averages, then they would needed to have used them for some specific purpose after calculating them. From our understanding, the authors essentially reported that shorter-term erosion rates derived from luminescence measurements were higher than the long-term averages; thus, the word "inferred" is appropriate in this case. We would welcome further clarification if required.

**RC2:** Figure 1: Please also mark the locations of the road cuts and provide photographs of the road cut sampling sites.

AC: To allow the roadcut sample locations to be shown, we have added an extra panel into Fig. 1 (Fig. 1B) to include the sample locations in NW Scotland. We have included photographs of the roadcut sections in the Supplementary material.

**RC2: Figure 3**: Why are the datasets for grain size and RGB different? Were the analysis performed on different slices?

**AC**: The measurements were not performed on exactly the same cores, but example cores for each sample. We have added the text below into the figure caption to clarify this:

Note that the RGB values and grainsize measurements were not derived from exactly the same cores, but example cores for each sample.

**RC2:** Figure 4: I would suggest to mention in the figure caption that core 3 of Road 2 was not considered for fitting. Also, information regarding the inferred model (which µ, which sigmaphi, reference) should be provided.

**AC**: We have added the following text into the figure caption for Fig. 4 "Note that core 3 of ROAD02 was not considered for fitting". We have also included the µ and sigmaphi values in each plot as it was an excellent suggestion.

**RC2:** Figure 5: I recommend to add the inferred model fits that are shown in figure 7 also here. In figure 7 it seems that the model fits (red lines) do not match the measured data for most signals, but it is hard to judge since the panels are rather small. Figure 5 would allow for a much better evaluation.

**AC**: We have moved the inferred erosion models from Fig. 7 to Fig. 5 to better represent the data and fit. Fig. 7 is now limited to only the inversion model profiles of erosion rates.

**RC2:** Figure 6: The unit for the ages should be a instead of $a^{-1}$.

**AC**: You are absolutely correct, sorry for the mistakes. We have corrected this.

**RC2:** Figure 7: It seems that the model (red lines) in C, G, h and I does not really fit the data. Could you please comment why this might be the case? Please also provide more information on the likelihoods (where do these come from) and the forbidden zone (how do I recognize it in the figure).

**AC**: This was also identified by RC2 in their more general comments above and so we have provided a more detailed response there of the further discussion that we have included exploring the fit of the inferred erosion models to the experimental data. The same response applies to this comment. Also, the new worked examples in "Section 2 Theoretical background" also includes further details on the calculations of likelihoods and the forbidden zone. These additions were included in response to the ECs and so we refer you to our response to the ECs for full details.

**RC2:** Figure S4a: The signals are only identical for sample ROAD 01 (1 a reference sample).

**AC**: Correct, we have added the missing text "for ROAD01 (0.01 a known-age sample)" into the caption to clarify this.
* * *
**Associate Editor comments: Jim Feathers**

**EC**: Associate editor here.

The two reviews of Smedley et al. on erosion rates in NW Scotland have now been posted. I read them over and think they made many good points. The authors should address all their comments, but two in particular need attention. First, both reviewers found Figure 7 difficult to understand. I found the graphs with mostly blue shading to be fully incomprehensible, yet as one of the reviewers pointed out it is the main figure of the paper. Second, the authors rely heavily on Lehmann et al.'s approach for determining erosion rates that are punctuated rather than steady state. The authors need to better explain that approach, so that the reader does not have to consult Lehmann et al. to understand what the authors are doing. I would suggest giving a working example of how this approach works, using one of their samples as the example.

**AC**: We thank the EC for this suggestion and agree that this will add clarity to the approach used here. As such, we have added two paragraphs into "Section 2 Theoretical background" that provide worked examples of steady state and transient state erosion based on our data presented in Fig. 7 (revised). See the paragraphs below:

For determining erosion rates for rock surfaces of known exposure age, Sohbati et al. (2018) use a confluent hypergeometric function to provide an analytical solution, but assuming only steady-state erosion. Lehmann et al. (2019a) provide a numerical approach that exploits the differential sensitivities to erosion of the luminescence (shorter-term) and cosmogenic nuclide (longer-term) techniques to erosion to infer erosion histories (steady state and transient over time) for rock surfaces. This approach uses the experimental data from the luminescence depth profiles and the $^{10}$Be concentrations for the same sample. Modelling of the luminescence depth profiles accounts for the electron trapping dependent upon the environmental dose-rate and $D_0$ but does not consider athermal loss of the signal (i.e. anomalous fading) as it has been demonstrated to have a negligible impact upon the luminescence depth profiles (Lehmann et al. 2019a). Modelling of the $^{10}$Be concentrations assumes that no inheritance of the cosmogenic nuclides from prior exposure has occurred, and that the $^{10}$Be concentrations have been corrected for the sample depth and density, topographical shielding, local production rates, and the sample location (longitude, latitude and elevation). The combined experimental data for the luminescence depth profiles and cosmogenic nuclides are solved simultaneously for two unknowns: the exposure age and the erosion rate. Forward modelling is used to calculate all of the possible combinations of luminescence depth profiles and $^{10}$Be concentrations for synthetic erosion and exposure histories, which are then validated using inversion models against the experimental data to determine the combinations with the highest likelihood. A forbidden zone is determined where the range of possible solutions of erosion rates and durations are in excess of those that are feasible for the experimental $^{10}$Be concentrations provided for the sample; these solutions are excluded from the parameter ranges used for the inversion model. For example, the forbidden zone identified in the inversion model profile shown in Fig. 7A is restricted to ranges from ca. $10^4$ mm/ka for durations of ca. 100 a to ca. $10^3$ mm/ka for ca. >3000 a.

The approach of Lehmann et al. (2019a) models synthetic erosion histories in both steady and transient states. Steady state erosion assumes a constant erosion rate throughout the duration of surface exposure. Transient erosion is typical of shorter exposure histories where a steady state of erosion has not yet been reached. Transient erosion varies with time and was simulated here by assuming that the evolution of erosion in time follows a stepped function of fixed increases in erosion rates from zero for varying durations throughout the exposure history for each synthetic erosion history simulated. An illustration of this is provided by Fig. 7A where transient erosion rates of between ca. $10^4$ mm/ka were inferred for a minimum duration of ca. ≤1 a, extending up to ca. $10^3$ mm/ka for durations up to ca. 50 a. Beyond ca. 50 a, a steady state of erosion was reached at a constant erosion rate of ca. $10^3$ mm/ka, represented by the flattening of the profile with the highest likelihood. Alternatively, a profile indicative of a transient state of erosion where no steady state has been established is illustrated by Fig. 7D where transient erosion rates of between ca. $10^2$ mm/ka were inferred for a minimum duration of ca. ≤1 a, extending up to ca. $10^1$ mm/ka for durations beyond ca. 200 a. This numerical approach (Lehmann et al. 2019a) allows erosion history to be considered as non-constant in time (i.e. transient), in addition to steady-state, and so it is more indicative of the stochastic erosional processes (driven by temperature, precipitation, snow cover, wind) in nature.

**EC**: So I think the paper needs major changes, but I do concur with the reviewers that the paper has merit and deserves publication once the problems are corrected.

A few other minor comments that I have:

**EC**: **Lines 153-158** – Were whole rocks or just portions collected in the field? Cores were drilled in the laboratory, but it is not clear how the rocks were collected.

**AC**: We have clarified this in the text in the Methods section. See below:

Portions of the original boulder or bedrock sample were collected in the field in daylight and immediately placed into opaque, black sample bags.

**EC**: **Line 196** – What do you mean by "similar". You just mentioned fairly wide ranges in reduction.

**AC**: We have added further clarification to the text in response to this comment. See below:

This indicates that within our samples the minerals emitting the IRSL signals (i.e. K-feldspar) have similar inherent bleaching rates when exposed to longer durations of time (i.e. > 8 h in the solar simulator).

**EC**: **Line 578**. Figure 4 caption. What do you mean by "replicate" core? Replicates of what?

**AC**: We have added further clarification into the figure caption to help explain this.

Presented in age-order are the IRSL-depth profiles for each of the three replicate cores analysed per sample.

**EC**: **Lines 229-239** – I have seen surfaces of cores show complete saturation when nearby cores from the same rock did not. I am not sure we understand fully why this should be.

**AC**: Interesting, it is certainly puzzling. We have added further clarification in this section in response to Benjamin Lehmann's comments, and hopefully our discussion here helps to fuel future research to address these issues.

**Editors follow-up comments**

**Associate Editor comments: Jim Feathers**

**EC: 1.** Maybe most other readers understand this, but I do not understand what forward vs inversion modeling is. Maybe a line of explanation will help readers like me who have little modeling experience.

**AC:** We have expanded the explanation of forward and inversion modelling in Paragraph 4 of Section 2 Theoretical background. The new text is shown below:

Forward modelling is used to simply simulate a projected outcome and here it is used to calculate all of the possible luminescence depth profiles for these synthetic erosion and exposure histories. Inversion modelling matches measured data with the outcome of simulations to determine best fit of the raw data. Here, inversion modelling was used to validate the luminescence depth profile and cosmogenic nuclide concentration data against the synthetic erosion and exposure histories to determine the combinations with the highest likelihood.

**EC: 2.** On lines 147-148, I think the definition of transient erosion is rather opaque. I am not sure what you mean.

**AC:** We have expanded this description to improve the explanation of transient erosion and how it is defined in the approach of Lehmann et al. (2019a). The text in Paragraph 5 of Section 2 Theoretical background has been edited as below:

Transient erosion is typical of shorter exposure histories where a steady state of erosion has not yet been reached and the erosion rate varies over time. In the approach of Lehmann et al. (2019a), transient erosion is defined by erosion rates that decrease linearly with increased timing of erosion onset within the parameter space, ultimately reaching steady state (i.e. a constant erosion rate).

**EC: 3.** In lines 151-154, you have a "between" but no "and" to go with it. Is it supposed to be between 0 and 10^4 mm/ka?

**AC:** We have removed the "between".

**EC: 4.** I found it interesting that on one core one of the MET-pIRIR signals detected transient erosion while the other two did not. I was not entirely clear why this should be. You say this is the advantage of using MET-pIRIR, because it allows complications like surface coatings to be detected. I am wondering if the use of MET-pIRIR is mainly to spot problems that might not be detected with just one signal, or if it can lead to more positive interpretations. (You do not have to address this comment, partly because I am not sure I completely understand my own question.)

**AC:** We will respond as we think this is a very valid point and exactly the reasoning for using the MET-pIRIR signal. We therefore assume that we have not quite conveyed this outcome as effectively as possible and so have edited the following text to provide better clarity:

**Abstract:** The use of a multi-elevated temperature, post-infra-red, infra-red stimulated luminescence (MET-pIRIR) protocol (50, 150 and 225°C) was advantageous as it identified samples with complexities that would not have been observed using only the standard IRSL signal measured at 50 °C, such as that introduced by within-sample variability (e.g. surficial coatings).

**Section 7. Conclusion**: Larger sample populations and careful sampling of rock surfaces (avoiding the potential for rock pools and trickle paths) will likely be key for accurate measurements of landscape-scale erosion, and the use of a MET-pIRIR protocol (50, 150 and 225 °C) is advantageous as it can identify samples suffering from complexities that would not have been observed using only the standard IRSL signal measured at 50 °C, such as that introduced by within-sample variability (e.g. surficial coatings).

**EC:** 5) Your Figure 7 is much better than it was, or at least explained better. I am not quite sure I understand the x-axis term, "erosion initiation" I understand, in 7A for example, that erosion begins in year 1, but does not reach a steady rate until almost 100 years. Is "initiation" the right word? By the time 100 years is reached, erosion has already been initiated.

**AC:** We have added further clarification into the figure caption for Fig. 7 to better explain the data plotted in the x-axes. See the text below:

Figure 7. Probability distributions inverted from the respective plots of luminescence depth profiles derived from the inversion results (using the approach of Lehmann et al. 2019a) for samples BALL02 (A-C) and BALL03 (D-F) using the $IR_{50}$, (A,D), $pIRIR_{150}$ (B,E) and $pIRIR_{225}$ (C,F) signals. The x-axis plots the time interval of the erosion rate initiation. Forbidden zones define the range of solutions with high erosion rates and durations that are not feasible within the bounds of the experimental $^{10}$Be and luminescence data.

**Final round of revisions**. Please note that the ACs are our responses to the (unedited) reviewer and editor comments.

Reviewer 1 (Benjamin Lehmann) comments: RC1

Author comments: AC

**Reviewer 1: Benjamin Lehmann (RC1)**

**RC:** I thank the authors to address all my comments, to answer all my questions and produce the change in the manuscript. In its revised version, the manuscript gains in quality with a better explanation of methlolodogy, a deepening of the discussions on the results and a clearer retranscription of the results in the different figures.

About the reconstruction of the the erosion history using Lehmann et al., 2019a approach, I acknowledge the efforts of the authors in discussing the difference between the experimental values and the inferred model (presented in Figs. 5G, 5F, 5F and 5I). All the points made by the authors

are accurate and could explain the difference mentioned above. I still want to raise few questions on this part.

First of all, to which values of the inversion correspond the dashed black line in Figure 5, the maximum likelihood, median value, values above a certain threshold? In Lehmann et al., 2019a, the inversion results show the best-fitting profiles inverted for all numerical solutions with likelihood > 5 %.

**AC:** The data in Fig. 4 and Fig. 5 are the best fit of the inversion results and we have added this information into the captions of both figures.

**RC:** Secondly, I still think the model should be able to fit the experimental values. One explanation of the poor fit quality could be that the model gives too much importance of the in the plateau and too little importance of the values in the bleaching front. A solution to test this potential modelling problem would be to run the inversion using only experimental values at depths from 0 to 10 mm. Being an important part of this study, I think the luminescence as an erosion-meter should show more convincing inversion results.

**AC:** We have explored the effect of modelling just the data from 0 to 10 mm depths as requested by the reviewer to assess the importance of the plateau on the fit, and whether this may account for the poor fit currently calculated. As a sensitivity test, we performed this for the sample BALL03 pIRIR$_{150}$ signal and the output is shown below. Evidently, only using the data from 0 to 10 mm made little difference to the fit of the data.

[Figure]

As currently stated in our discussion, specifically the last paragraph of Section 5.3, "it is possible that surficial weathering products may have changed in thickness and composition over time, which in turn could slightly vary the attenuation of light (Meyer et al. 2018; Luo et al. 2018), meaning that the calibration of $\overline{\sigma\varphi_0}$ and μ from ROAD02 here introduced uncertainty into the inferred erosion model as it was not time-varying".

Luminescence as an erosion meter is still a new application of the method. No studies have yet derived transient erosion ages that can be compared to independent erosion estimates. As such, statements such as "luminescence as an erosion-meter **should** (our emphasis) show more

convincing inversion results" are currently untested with real-world examples. We suggest a physical explanation for the lack of fit that is plausible and accords with circumstantial evidence from field observations. We would argue that a central aim of science is to take hypotheses (e.g. the erosion model) and test them against evidence (e.g. our measured profiles). Our findings thus set up important research questions that can refine the hypothesis. While beyond the scope of this study, such refinements can now be addressed in future studies thus improving the applicability of the luminescence erosion meter.